# Beyond Shallow Behavior: Task-Efficient Value-Based Multi-Task Offline MARL via Skill Discovery

## Abstract

As a data-driven approach, offline MARL learns superior policies solely from offline datasets, ideal for domains rich in historical data but with high interaction costs and risks. However, most existing methods are task-specific, requiring retraining for new tasks, leading to redundancy and inefficiency. To address this issue, we propose a task-efficient value-based multi-task offline MARL algorithm, Skill-Discovery Conservative Q-Learning (SD-CQL). Unlike existing methods decoding actions from skills via behavior cloning, SD-CQL discovers skills in a latent space by reconstructing the next observation, evaluates fixed and variable actions separately, and uses conservative Q-learning with local value calibration to select the optimal action for each skill. It eliminates the need for local-global alignment and enables strong multi-task generalization from limited, small-scale source tasks. Substantial experiments on StarCraft II demonstrate the superior generalization performance and task-efficiency of SD-CQL. It achieves the best performance on **13** out of 14 task sets, with up to **68.9%** improvement on individual task sets.

## 1 Introduction

Multi-agent reinforcement learning (MARL), as a cornerstone of artificial intelligence, provides advanced methodologies to tackle complex challenges requiring coordinated, task-driven decision-making among multiple agents through interaction (Shoham & Leyton-Brown, 2008; Gronauer & Diepold, 2022). Integrated with deep neural networks, MARL has demonstrated exceptional success across a diverse range of critical applications, e.g., video games (Mathieu et al., 2021), autonomous systems (Shalev-Shwartz et al., 2016), and finance (Lee et al., 2007). As data's centrality to machine learning grows, offline MARL has drawn increased attention from researchers (Li et al., 2023; Shao et al., 2023; Liu et al., 2024b). It learns policies solely from offline datasets, particularly suitable for domains where historical data is abundant but real-time interaction is costly or risky.

However, most existing offline MARL methods (Li et al., 2023; Shao et al., 2023; Liu et al., 2024b) are task-specific. They tend to focus solely on the source task of the dataset and train a tailored policy. Consequently, although they may achieve impressive performance, any alterations to the deployment environment or the task, such as a small change in the controllable agents count, necessitate reacquiring data and training an entirely new policy, leading to significant inefficiency.

Hence, unlike existing approaches, we seek to build a method capable of learning versatile policies offline on a small set of known tasks and generalizing well to unseen but similar scenarios. By doing so, one can eliminate the redundancy and resource overhead of repeatedly retraining from scratch for each new task. Furthermore, in practice, accessible task scenarios are typically limited, making it infeasible to train policies across all potential scenarios. Hence, we aim for this method to exhibit high ***task-efficiency***—handling more unseen tasks more effectively with fewer known tasks. This will help broaden its applicability, especially in scenarios where known tasks are scarce.

Therefore, we need to confront the following two key challenges: (i) The distributional shift between limited offline data and the real environmental dynamics within each task. (ii) Generalization across tasks, that is, maintaining policy performance on unseen tasks without access to their offline data or any dynamic information. Although we can overcome the first challenge via conservative policy

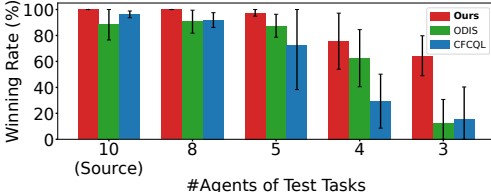 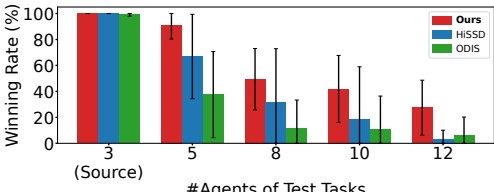

(a) Skill-Discovery enhances the multi-task generalization. Trained on *10m* dataset, and tested on all tasks.

(b) Recent Skill-Discovery methods struggle due to the limited source task. Trained on *3m* dataset, and tested on all tasks.

Figure 1: Skill-Discovery enhances cross-task generalization for offline MARL, while existing methods have limitations. The number with *Source* behind it indicates the source task used for training. The error bars represent $\pm 1$ standard deviation (truncated outside the $0 \sim 100\%$ range).

optimization, increased conservatism on the source task usually leads to poorer generalization to unseen tasks, highlighting the importance of solving both challenges simultaneously.

As shown in Figure 1a, we train the advanced offline MARL algorithm, CFCQL (Shao et al., 2023), on a task with 10 agents (source task) and test it on tasks with fewer agents. We find that it performs poorly when the number of agents in the test tasks is reduced, even though these tasks appear easier. This suggests that the agents need to learn and utilize generalizable decision-making structures across tasks, making skill-discovery a promising solution. In this approach, agents identify high-level decision patterns, i.e., "skills"[1], from source tasks and select the appropriate one to perform specific actions in unseen tasks, achieving cross-task generalization. However, most existing skill-discovery methods require online interactions to discover skills (Liu et al., 2022; Tian et al., 2023), or learn the utilization of offline discovered skills (Chen et al., 2024), with limited attention to the entirely offline setting.

Recently, ODIS (Zhang et al., 2023a) and HiSSD (Liu et al., 2025) propose methods for multi-task offline MARL. Although they exhibit better generalization to unseen tasks (e.g., ODIS in Figure 1a), as shown in Figure 1b, they both struggle to discover effective skills in small-scale source tasks. Moreover, both methods rely on behavior cloning (BC) to decode actions from a presumed optimal skill representation directly. However, this often results in a ***Skill-Action Optimality Mismatch***. In other words, as a skill usually spans a subset of offline actions, BC biases agents toward likely, not optimal, choices. While optimizing actions via value-based methods can mitigate this likelihood bias, challenges in the scalability and generalization of Q-values leave its feasibility for offline multi-task MARL still to be established. This motivates us to investigate the following important question:

***Can we combine the strengths of skill discovery and value-based methods to enable strong cross-task performance with high task-efficiency?***

We answer this question affirmatively by proposing the Skill-Discovery Conservative Q-Learning algorithm (**SD-CQL**). It achieves effective skill discovery by reconstructing the next local observation and training a set of scalable skill-conditioned policies via local value calibration, showing that value-based method in this setting is not only viable but also superior. As shown in Figure 2, the offline agents acquire two skills, *retreat* and *attack*, from the 3-marines task (Figure 2a). When deployed in the 12-marines task (Figure 2b), they exhibit the ability to *retreat* when health is low and to *attack* when health is sufficient, with *attack* encompassing two specific actions: firing and advancing. It is consistent with the visualization of the skills adopted by the agent, as shown in Figure 2c. More details can be found in Section 4.3 and Appendix E.

Different from existing methods (Zhang et al., 2023a; Liu et al., 2025) that rely on global information or predefine the skill count, SD-CQL discovers continuous skill vectors entirely based on local observations. This prompts agents to extract expressive, task-agnostic features from limited data, and eliminates the local-global alignment. Then, SD-CQL conservatively trains separate skill-conditioned Q-networks for fixed and variable actions, and applies local value calibration to mitigate agent-scaling estimation error. Extensive experiments demonstrate its SOTA task efficiency.

---

[1]Following prior work (Zhang et al., 2023a; Liu et al., 2025), the term *skill* refers to a broader concept of distinguishable behavioral patterns, and is not necessarily limited to an explicit action sequence or a sub-policy.

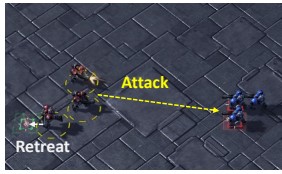
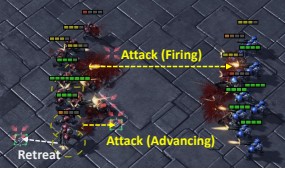
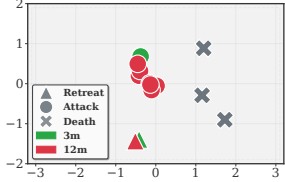

(a) Training in *3m*.   (b) Testing in *12m*.   (c) Visualization of skills.

Figure 2: SD-CQL effectively learns two skills, ***retreat*** and ***attack***, from the source task (a) and successfully transfers them to the target task (b), each with distinct actions. This is consistent with the visualization in (c), where each marker corresponds to a skill adopted by the agent.

This paper is organized as follows. Section 2 introduces the necessary background, followed by Section 3, which elaborates on the proposed algorithm. Section 4 presents the main empirical results, and Section 5 concludes the paper with key takeaways. Beyond the main text, the appendices provide the LLM usage description (Appendix A), a comprehensive review of related work (Appendix B), implementation and experimental details (Appendices C and D), additional results (Appendices E, F, G, I, and J), and extended discussions (Appendix H).

In summary, the key contributions of this work are:

- We investigate a rarely explored yet important field: multi-task offline MARL. We analyze current limitations and propose a novel method to demonstrate that value-based method is not only viable but also superior in this setting, offering a new perspective for research in this area.

- We propose SD-CQL for multi-task offline MARL. It discovers skills in latent space by reconstructing the next observation, then evaluates fixed and variable actions separately. Finally, it executes the optimal action for each skill via conservative Q-learning with local value calibration, enabling better generalization and task efficiency than existing methods.

- Through comprehensive experiments, we show that SD-CQL achieves the best performance in **13** out of 14 task sets, with an average performance improvement of up to **68.9%** on individual task sets over existing baselines.

## 2 BACKGROUND

### 2.1 MULTI-TASK MARL AND MULTI-TASK OFFLINE MARL

A cooperative MARL task, indexed by $m$, can be formulated as a decentralized partially observable Markov decision process (Dec-POMDP) (Oliehoek et al., 2016). A Dec-POMDP is represented by a tuple $(\mathcal{N}_m, \mathcal{S}_m, \mathcal{O}_m, \mathcal{A}_m, P_m, R_m, \gamma)$, where $\mathcal{N}_m$ is the set of agents, $\mathcal{S}_m$ is the set of states, $\mathcal{O}_m$ is the joint observation space, $\mathcal{A}_m$ is the joint action space, $P_m$ is the state transition probability (defining the probability of transitioning to the next state given the current state and joint action), $R_m$ is the immediate reward shared by all agents, and $\gamma$ is the discount factor.

The multi-task MARL problem is a collection of such MARL tasks, which can be represented by the tuple $(\mathcal{N}, \mathcal{S}, \mathcal{O}, \mathcal{A}, P, R, \gamma, \mathbb{T})$. Here, $\mathbb{T}$ is the set of tasks, $\gamma$ is the discount factor shared by all tasks, and the remaining elements are the union of corresponding elements across all tasks. For example, $\mathcal{N} = \bigcup_{m \in \mathbb{T}} \mathcal{N}_m$ represents the union of the sets of agents across all tasks.

For the task $m$, each agent maintains its observation-action history $\tau^i \in \mathcal{T}_m^i$, and the corresponding joint history is denoted by $\tau \in \mathcal{T}_m$. $\mathcal{T} = \bigcup_{m \in \mathbb{T}} \mathcal{T}_m$ represents the collection of observation-action histories across all tasks, and the goal is to find a joint policy $\pi : \mathcal{T} \to \mathcal{A}$ that maximizes the expected discounted return average over all tasks: $\mathcal{J} = \mathbb{E}_{\pi, \mathbb{T}} \left[ \sum_{t=0}^{T} \gamma^t \cdot r_m(s_t^m, \boldsymbol{a}_t^m) \right]$, where $T$ is the time horizon and $r_m(s_t^m, \boldsymbol{a}_t^m)$ is the reward for taking joint action $\boldsymbol{a}_t^m$ at state $s_t^m$ in task $m$.

As for multi-task offline MARL, agents can only access the decision dataset $\mathcal{D} = \{\mathcal{D}_m\}_{m \in \mathbb{T}_s}$, where $\mathbb{T}_s \subset \mathbb{T}$ is referred to as *source tasks* (the complement referred to as *unseen tasks*), without

any interaction with the environment. The goal is still to maximize $\mathcal{J}$. In Appendix H.3, we further elaborate on the multi-task learning in the contexts of MARL and conventional RL.

## 2.2 CTDE Framework, QMIX and CQL

A common framework for cooperative MARL is Centralized Training for Decentralized Execution (CTDE). In this framework, agents can leverage centralized information during training but must rely solely on their local observations during execution. In this work, we adopt QMIX (Rashid et al., 2020) as our backbone algorithm, which is one of the most popular discrete-action CTDE algorithms. In principle, our method can be applied to any value-based algorithm.

In QMIX, each agent maintains an individual Q-function $Q_i(\tau^i, a^i)$, which is conditioned on its own observation-action history $\tau^i$ and action $a^i$. Then, it calculates a joint Q-function $Q_{tot}(\tau, \boldsymbol{a})$ from individual Q-functions through a mixing network $f_s$, such that:

$$Q_{\text{tot}}(\tau, \boldsymbol{a}) = f_s(Q_1(\tau^1, a^1), \ldots, Q_n(\tau^n, a^n)). \tag{1}$$

The mixing network is designed to satisfy the monotonicity constraint, ensuring that the partial derivative of $Q_{\text{tot}}$ with respect to each $Q_i$ is non-negative. To train the Q-function, QMIX minimizes the temporal-difference error on $Q_{\text{tot}}$.

In offline MARL, the lack of interaction may cause Q-value overestimation on out-of-distribution (OOD) samples. Thus, Conservative Q-learning (CQL) (Kumar et al., 2020) uses a regularization term to enhance the Q-value of in-dataset samples while suppressing that of OOD ones:

$$\mathcal{L}_{\text{CQL}} = \mathbb{E}_{\tau \sim \mathcal{D}, \boldsymbol{a} \sim \mu} \left[ Q_{tot}(\tau, \boldsymbol{a}) \right] - \mathbb{E}_{(\tau, \boldsymbol{a}) \sim \mathcal{D}} \left[ Q_{tot}(\tau, \boldsymbol{a}) \right] \tag{2}$$

where $\mu$ denotes the sampling distribution (e.g., sampling from a uniform distribution).

## 3 Skill-Discovery Conservative Q-Learning

To develop scalable policies that can generalize to varying unseen tasks through data from only a few source tasks, we propose Skill-Discovery Conservative Q-Learning (SD-CQL), a task-efficient algorithm for multi-task offline MARL. As illustrated in Figure 3, SD-CQL discovers cross-task generalizable skills in the latent space through observation reconstruction, and then trains scalable policies via multi-task CQL and local value calibration. This approach mitigates distributional shifts and error accumulation in offline MARL, as well as the generalization challenges inherent in multi-task learning. We present a detailed introduction of each component below, and offer an in-depth architectural comparison with existing methods in Appendix H.2. The full algorithm is presented in Algorithm 1, located in Appendix C.

### 3.1 Observation Reconstruction

To improve policy generalization across tasks, a natural idea is to make agents learn task-shared decision features, i.e., "skills", from offline data. Hence, we choose to achieve this by reconstructing the next local observations without global state or task-specific rewards. This not only helps agents to learn more transferable, task-agnostic temporal features, but also reduces skills' dependence on task scale and enables direct distributed execution without local-global alignment.

#### 3.1.1 Observation Encoder

As the number of agents may vary across tasks, leading to changes in state or observation size, Following the standard practice (Liu et al., 2020; Zhang et al., 2023a; Wu et al., 2024), we decompose the state or observation into entity units and map them into a shared embedding space. We then adopt a transformer (Vaswani, 2017), following Zhang et al. (2023a), for further processing.

Specifically, we first decompose the $i$-th agent's observation at $t$-th time step $o_t^i$ into $o_t^{i,0}$, which is related to the agent itself, and $\{o_t^{i,k}\}_{k=1}^{K_i}$ corresponding to the entities it can observe. Among these $K_i$ entities, there are $K_i^a$ entities that agent $i$ can interact with by taking actions. We further encode these heterogeneous entity information into embeddings $\{e_{i,k}\}_{k=0}^{K_i}$ with same dimensions. Finally, we use a single-layer transformer to capture the relationships between entities and obtain the final

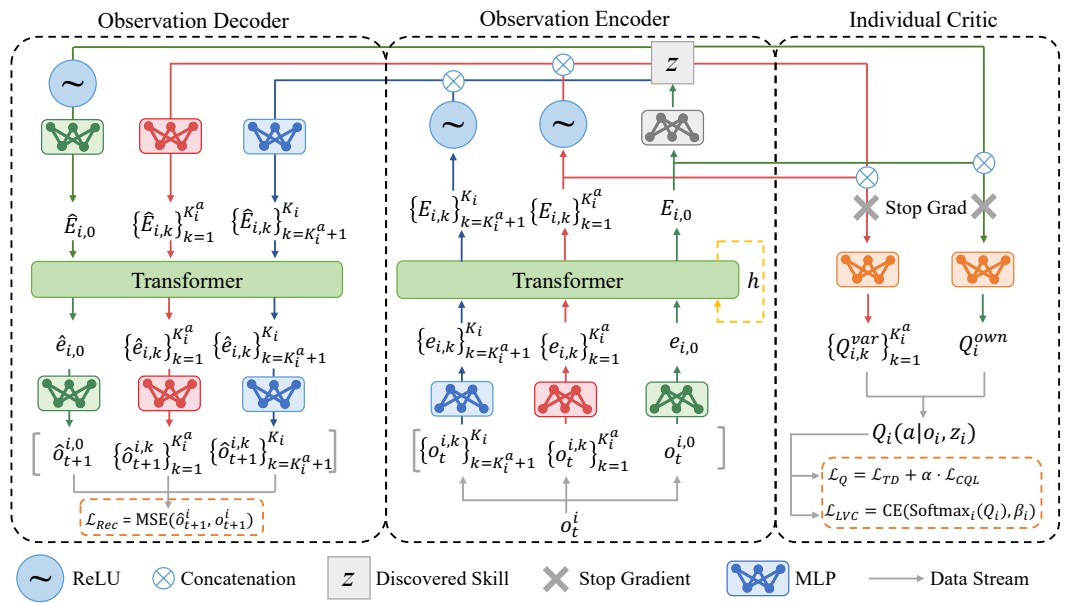

Figure 3: SD-CQL framework. For the $i$-th agent, the encoder splits the current observation $o_t$ into $K_i + 1$ entities, embedding them into consistent dimensions, with the 0-th representing the agent itself. Then, the skill $z$ is extracted from $E_{i,0}$, the embedding associated with the agent itself. Finally, SD-CQL injects local information into $z$ by reconstructing the next observation $o_{t+1}$. Meanwhile, it computes the $z$-conditioned Q-values for fixed and variable actions separately, and optimizes scalable polices via CQL and local value calibration.

encoding vector $\{E_{i,k}\}_{k=0}^{K_i}$ for each entity. More details of the decomposer and architecture are provided in Appendix D.3.

Since the agents' local observations are relatively limited, we add an additional "pseudo-entity" $h = e_{i,K_i+1}$ as hidden variable. It is represented in the same embedding space as other entity embeddings and recurrently involved in the encoder transformer but is excluded from the decoder, enabling the skill to capture temporal information. We also incorporate the previous actions $a_{t-1}$ in $o_t$ in line with the standard practices, but exclude them from the reconstruction target $o_{t+1}$.

### 3.1.2 OBSERVATION DECODER

To enable situation-appropriate skill selection, we propose a decoder to learn expressive representations in a continuous latent space via reconstructing the next local observation. By focusing on action-agnostic local transitions, these representations capture higher-level spatiotemporal patterns while reducing task-specific information, thereby improving cross-task generalization.

Concretely, for the embeddings $\{E_{i,k}\}_{k=0}^{K_i}$ from the observation encoder, the decoder first extracts a $d$-dimension latent skill vector $z \in \mathbb{R}^d$. Then, it reconstructs the embeddings $\{\widehat{E}_{i,k}\}_{k=0}^{K_i}$ with $z$. Finally, similar to the encoder, the reconstructed embeddings are passed through a single-layer transformer, restored into entity vectors $\{\hat{o}_{i,k}\}_{k=0}^{K_i}$, and reassembled into the next observation $\hat{o}_{t+1}^i$. Detailed architecture can be checked in Appendix D.3. Then, the final reconstruction loss is:

$$\mathcal{L}_{\text{Rec}} = \frac{1}{N} \sum_{i=1}^{N} \text{MSE}(\hat{o}_{t+1}^i, o_{t+1}^i) \tag{3}$$

where $N$ is the number of controllable agents in the task, and $\text{MSE}(\cdot, \cdot)$ is the Mean Square Error.

***Remark:*** *Although our reconstruction only use local observations, each agent's observation includes the teammates and opponents in its field of view. Therefore, the transformer can encode local relations among the agent, its teammates and its opponents, and then the latent $z$ can still captures short-horizon cooperative cues.*

## 3.2 SKILL-CONDITIONED POLICY OPTIMIZATION

After identifying and selecting the appropriate skill, existing multi-task offline MARL algorithms primarily execute the corresponding actions through behavior cloning (BC). However, since a single skill often encompasses multiple specific actions, selecting the optimal skill does not necessarily ensure that all associated actions are optimal.

Therefore, we leverage conservative Q-learning to optimize the skill-conditioned policy $\pi(a \mid s, z)$, enabling the execution of optimal actions associated with the selected skill. To mitigate the accumulated estimation errors inherent in Q-learning and reduce distributional shift, SD-CQL separately evaluates fixed and variable actions with local value calibration, thereby enhancing training stability for larger-scale tasks.

### 3.2.1 CONSERVATIVE SKILL-CONDITIONED Q-VALUE

To handle the variation in the number of actions across different tasks, we utilize two separate Q-networks. One network, called $Q^{own}$, is responsible for a fixed set of actions related to the agent itself, while the other, called $Q^{var}$, handles a variable set of actions associated with other entities in its observation. Both networks receive the corresponding entity embeddings $E_{i,k}$ and skill vectors $z$ as inputs and output the Q-values for their respective actions. The parameters of Q-value networks are shared by all the agents. Therefore, the individual Q-values for the $i$-th agent with skill $z_i$ are:

$$Q_i(o_i, z_i, a) = \begin{cases} Q^{own}(E_{i,0}, \ z_i, a) & \text{if } a \in \mathcal{A}_i^{own} \\ Q^{var}(E_{i,k}, \ z_i, a) & \text{if } a \in \mathcal{A}_i^k \end{cases} \tag{4}$$

where $\mathcal{A}_i^{own}$ is the set of actions that only related to the $i$-th agent itself, and $\mathcal{A}_i^k$ is the set of actions that related to the $i$-th agent and the $k$-th other entity. While similar techniques have been adopted in previous work (Lin & Lee, 2024) to enhance Q-value estimation, it is important to note that our use of separate networks is primarily motivated by the need to accommodate the varying dimensions of states, observations, and actions inherent in multi-task offline MARL.

To avoid interference between observation reconstruction and policy optimization, we truncate the gradient propagation of the embeddings and skill vectors before feeding them into the Q-network.

Finally, we employ a mixing network to aggregate the individual Q-values into a global Q-value $Q_{\text{tot}}$ according to equation 1. Then, we can optimize $Q_{\text{tot}}$ through the temporal difference loss outlined below:

$$\mathcal{L}_{\text{TD}} = \left[ Q_{\text{tot}}(\tau, \boldsymbol{z}, \boldsymbol{a}) - \left( r + \gamma \max_{\boldsymbol{a}'} \bar{Q}_{\text{tot}}(\tau', \boldsymbol{z}', \boldsymbol{a}') \right) \right]^2 \tag{5}$$

where $\tau'$, $\boldsymbol{a}'$ and $\boldsymbol{z}'$ denote the next observation-action history, joint action, and joint skill, respectively, and $\bar{Q}_{\text{tot}}$ represents the target joint action-value function.

In the offline setting, the scarcity of data and the lack of online interaction may cause agents to make detrimental decisions by overestimating OOD state–action pairs. To address this issue, we employ CQL (Kumar et al., 2020) to regularize the skill-conditioned Q-values:

$$\mathcal{L}_{\text{CQL}} = \mathbb{E}_{\tau \sim \mathcal{D}, \boldsymbol{a} \sim \mu} \left[ Q(\tau, \boldsymbol{z}, \boldsymbol{a}) \right] - \mathbb{E}_{(\tau, \boldsymbol{a}) \sim \mathcal{D}} \left[ Q(\tau, \boldsymbol{z}, \boldsymbol{a}) \right] \tag{6}$$

where $\mu$ denotes the sampling distribution (e.g., sampling from a uniform distribution). And the total loss function for Q-learning is:

$$\mathcal{L}_{\text{Q}} = \mathcal{L}_{\text{TD}} + \alpha \mathcal{L}_{\text{CQL}} \tag{7}$$

where $\alpha > 0$ is a hyperparameter to control conservatism.

### 3.2.2 LOCAL VALUE CALIBRATION

In addition to distributional shifts, the accumulation of estimation errors caused by multiple agents is another challenging issue. In multi-agent environments, relying solely on Q-learning remains inadequate to mitigate the rapidly escalating Q-value estimation errors (Pan et al., 2021).

Therefore, inspired by previous studies (Fujimoto & Gu, 2021; Pan et al., 2022), we propose *Local Value Calibration (LVC)* to calibrate each agent's local Q-value distribution over all available actions

against its corresponding behavior policy. This term enhances the stability of SD-CQL across tasks with different scales:

$$\mathcal{L}_{\text{LVC}} = \frac{1}{N} \sum_{i=1}^{N} \text{CE}\left(\text{Softmax}(Q_i),\ \beta_i\right) \tag{8}$$

where $\text{CE}\left(\cdot, \cdot\right)$ is the cross entropy loss, $N$ is the number of agents, $Q_i$ is the individual Q-values of all the actions available for the $i$-th agent, and $\beta_i$ is the corresponding one-hot action in the offline datasets.

Then, the total loss function for SD-CQL is:

$$\mathcal{L} = (1 - \eta) \cdot \mathcal{L}_{\text{Q}} + \eta \cdot \mathcal{L}_{LVC} + \mathcal{L}_{\text{Rec}} \tag{9}$$

where $\eta \in [0, 1]$ is the hyperparameter to control the strength of value calibration.

## 4 EXPERIMENTS

To evaluate the performance of SD-CQL in multi-task offline MARL scenarios, we establish multiple transfer training task sets based on the StarCraft Multi-Agent Challenge (SMAC) (Samvelyan et al., 2019) and the Multi-Agent Particles Environment (MPE) (Lowe et al., 2017). Through these experimental task sets, we aim to address the following questions: (i) Compared to existing algorithms, does SD-CQL demonstrate better performance in multi-task offline MARL (Section 4.2), (ii) Do skill vectors indeed characterize the decision-making features of agents in different contexts (Section 4.3), (iii) Are components within SD-CQL critical to its performance (Appendix F), and (iv) Is SD-CQL still effective when extended to continuous-action environments (Appendix G).

### 4.1 SETUP

**Datasets**   Following the definition by D4RL (Fu et al., 2020), our experiments utilize datasets of four quality: *Expert*, *Medium*, *Medium-Replay*, and *Medium-Expert*. For fair comparisons, we use the datasets provided by ODIS (Zhang et al., 2023a). Details are provided in Appendix D.1.

**Task Sets**   To comprehensively simulate different transfer scenarios, we construct 14 representative offline-training, zero-shot transfer task sets across five scenarios: *Marine-Easy*, *Marine-Hard*, *Stalker-Zealot*, *Marine-Single*, and *Marine-Single-Inv*, where the *Stalker-Zealot* scenario involves heterogeneous agents. They can be categorized into two groups: (i) Training on multiple tasks and test on multiple tasks (Multi-to-Multi) and (ii) Training on one task and test on multiple tasks (One-to-Multi). The first three scenarios, which fall under the Multi-to-Multi group, are consistent with those in ODIS (Zhang et al., 2023a), while the last two, classified as One-to-Multi, are additionally designed by us. More details are included in Appendix D.2.

**Baselines**   We primarily select four offline multi-task MARL algorithms as baselines: BC-t, BC-r, CQL (Kumar et al., 2020), ODIS (Zhang et al., 2023a), and HiSSD (Liu et al., 2025). Among them, BC-t is a behavior cloning approach based on a multi-task transformer, BC-r incorporates return-to-go information into the input of BC-t, CQL is a representative offline value-based method, which we implement with a multi-task transformer, ODIS is the first algorithm tailored for multi-task offline MARL, and HiSSD is the state-of-the-art algorithm for multi-task offline MARL. Implementation details for the baselines are provided in Appendix D.3.

**Experiment Setup**   We implement SD-CQL based on the PYMARL2 (Hu et al., 2023) and the ODIS codebase, while the baselines are directly utilized from the codes implemented by ODIS (Zhang et al., 2023a) and HiSSD (Liu et al., 2025). To ensure fairness, each algorithm runs $35,000$ steps of multi-task offline training. The exception is ODIS, which additionally requires an initial pre-training of $15,000$ steps. During evaluation, each test environment runs 32 episodes, and the average results are recorded. Unless otherwise specified, we report the average performance of the final policy across five random seeds, with the best performance for each task highlighted in bold. The learning curves are presented in Appendix J.

**Hyperparameters** For SD-CQL, the primary hyperparameter adjusted is the LVC weight $\eta$, while the CQL weight $\alpha$ is fixed at 1.0 (the same value as in CQL). As to ODIS and HiSSD, we use the hyperparameter configurations provided in their original papers (Zhang et al., 2023a; Liu et al., 2025) for Multi-to-Multi task sets. For One-to-Multi task sets we specifically designed, they do not offer hyperparameter configurations. Thus, we adjust the primary hyperparameters involved in their RL losses and report the best performance achieved. All other hyperparameters remain consistent with the default settings. For implementation details and main hyperparameter settings, please refer to Appendix D.4.

Table 1: Win rates of Multi-to-Multi task sets. The reported results are the average performance over all source tasks and unseen tasks, and are averaged over 5 random seeds. Bold numbers indicate the best performance, and $\pm$ denotes one standard deviation.

| Task Set | BC-best* | CQL | ODIS | HiSSD | SD-CQL (Ours) |
|---|---|---|---|---|---|
| *marine-easy-e* | **99.38** $\pm$ 0.40 | 31.13 $\pm$ 6.36 | 70.50 $\pm$ 30.14 | 97.94 $\pm$ 1.23 | 98.06 $\pm$ 1.00 |
| *marine-hard-e* | 64.84 $\pm$ 3.24 | 24.84 $\pm$ 1.53 | 21.09 $\pm$ 14.72 | 68.02 $\pm$ 4.47 | **70.68** $\pm$ 4.80 |
| *stalker-zealot-e* | 68.61 $\pm$ 3.93 | 10.62 $\pm$ 5.88 | 55.05 $\pm$ 9.11 | 66.59 $\pm$ 7.77 | **75.72** $\pm$ 3.73 |
| *marine-easy-m* | 71.81 $\pm$ 3.74 | 17.38 $\pm$ 7.31 | 68.50 $\pm$ 5.74 | 71.44 $\pm$ 2.77 | **75.81** $\pm$ 1.73 |
| *marine-hard-m* | 42.08 $\pm$ 2.63 | 9.17 $\pm$ 1.78 | 32.55 $\pm$ 4.31 | 47.60 $\pm$ 2.66 | **48.12** $\pm$ 5.66 |
| *stalker-zealot-m* | 23.94 $\pm$ 2.72 | 15.29 $\pm$ 4.89 | 11.97 $\pm$ 7.53 | 23.51 $\pm$ 3.76 | **40.43** $\pm$ 7.06 |
| *marine-easy-mr* | 49.62 $\pm$ 9.17 | 29.56 $\pm$ 11.73 | 11.06 $\pm$ 8.94 | 72.88 $\pm$ 3.16 | **75.81** $\pm$ 6.83 |
| *marine-hard-mr* | 46.98 $\pm$ 1.95 | 21.67 $\pm$ 2.83 | 37.29 $\pm$ 5.48 | 47.55 $\pm$ 3.56 | **49.84** $\pm$ 3.36 |
| *stalker-zealot-mr* | 17.64 $\pm$ 4.40 | 14.57 $\pm$ 2.02 | 8.17 $\pm$ 6.08 | 15.00 $\pm$ 4.02 | **23.08** $\pm$ 3.38 |
| *marine-easy-me* | 75.25 $\pm$ 7.41 | 33.56 $\pm$ 6.47 | 66.00 $\pm$ 9.21 | 78.31 $\pm$ 4.99 | **86.88** $\pm$ 5.68 |
| *marine-hard-me* | 51.93 $\pm$ 7.21 | 20.31 $\pm$ 1.97 | 19.58 $\pm$ 17.63 | 53.28 $\pm$ 1.88 | **56.20** $\pm$ 3.21 |
| *stalker-zealot-me* | 39.62 $\pm$ 2.98 | 13.41 $\pm$ 2.53 | 30.72 $\pm$ 14.65 | 40.29 $\pm$ 4.64 | **66.97** $\pm$ 7.01 |

*-e*, *-m*, *-mr*, and *-mr* represent *-expert*, *-medium*, *-medium-replay*, and *-medium-expert*, respectively.
* BC-best represents the better average performance between BC-r and BC-t.

## 4.2 EVALUATION

### 4.2.1 MULTI-TO-MULTI

The Multi-to-Multi task sets include *Marine-Easy*, *Marine-Hard*, and *Stalker-Zealot*, each comprising three source tasks of varying scales and several unseen tasks. Agents are trained offline using datasets from the source tasks and are tested across all tasks, with zero-shot testing conducted on the unseen tasks. For each task set, we set up four datasets of different qualities. We report the average performance of the algorithms across all tasks for each task set and dataset in Table 1, where BC-best denotes the better result between BC-t and BC-r. Detailed results can be found in Appendix I.1.

The results show that SD-CQL achieves the best performance for 11 out of 12 task sets, with the remaining one being very close to the best baseline algorithms (within 1.2%). In particular, in *Stalker-Zealot-Medium* and *Stalker-Zealot-Medium-Expert* task sets, SD-CQL significantly outperforms other algorithms, with performance improvements of 68.9% and 66.2%, respectively, compared to the best-performing baseline. This demonstrates that SD-CQL is capable of maintaining superior multi-task offline generalization performance across various dataset qualities.

### 4.2.2 ONE-TO-MULTI

The One-to-Multi task sets include *Marine-Single* and *Marine-Single-Inv*. Specifically, *Marine-Single* requires agents to train offline using only the dataset from the *3m* task and to test on tasks with scales up to *12m* to assess the agents' generalization ability to larger-scale tasks. (*3m* refers to each side having 3 Marine units, and similarly for the others). Conversely, *Marine-Single-Inv* requires agents to train offline only from the *10m* task and to test on tasks with a minimum scale of *3m* to evaluate whether the agents can truly learn more general decision-making skills. It is unlikely to expect a well-generalized policy from a small set of tasks with poor datasets. Hence, we use the expert dataset to simulate mastering simpler tasks before extending to more complex ones. We

Table 2: Win rates of One-to-Multi task sets. The results are averaged over 5 random seeds. Bold numbers indicate the best performance, and $\pm$ denotes one standard deviation.

| Task Set | | BC-best[*] | CQL | ODIS | HiSSD | SD-CQL (Ours) |
|---|---|---|---|---|---|---|
| *Marine Single* | 3m $^\diamond$ | **100.00** $\pm$ 0.00 | **100.00** $\pm$ 0.00 | 99.38 $\pm$ 1.4 | **100.00** $\pm$ 0.00 | **100.00** $\pm$ 0.00 |
| | 5m | 81.88 $\pm$ 9.22 | 88.12 $\pm$ 7.46 | 37.5 $\pm$ 33.15 | 66.88 $\pm$ 32.52 | **91.25** $\pm$ 10.92 |
| | 8m | 38.75 $\pm$ 22.38 | 40.00 $\pm$ 26.00 | 11.88 $\pm$ 21.47 | 31.87 $\pm$ 41.01 | **49.38** $\pm$ 23.63 |
| | 10m | 20.62 $\pm$ 22.38 | 34.38 $\pm$ 25.39 | 11.25 $\pm$ 25.16 | 18.75 $\pm$ 40.20 | **41.88** $\pm$ 25.83 |
| | 12m | 9.38 $\pm$ 13.98 | 17.50 $\pm$ 10.27 | 6.25 $\pm$ 13.98 | 3.12 $\pm$ 6.99 | **27.50** $\pm$ 21.13 |
| **Average** | | 50.12 $\pm$ 9.65 | 56.00 $\pm$ 11.07 | 33.25 $\pm$ 16.50 | 44.12 $\pm$ 18.89 | **62.00** $\pm$ 13.80 |
| *Marine Single-Inv* | 10m $^\diamond$ | **100.00** $\pm$ 0.00 | 0.62 $\pm$ 1.4 | 88.75 $\pm$ 12.22 | **100.00** $\pm$ 0.00 | **100.00** $\pm$ 0.00 |
| | 8m | 98.75 $\pm$ 1.71 | 3.75 $\pm$ 8.39 | 90.62 $\pm$ 8.84 | **100.00** $\pm$ 0.00 | **100.00** $\pm$ 0.00 |
| | 5m | 33.12 $\pm$ 33.63 | 6.25 $\pm$ 13.98 | 87.50 $\pm$ 8.84 | 91.88 $\pm$ 6.09 | **97.50** $\pm$ 2.61 |
| | 4m | 20.62 $\pm$ 33.04 | 0.62 $\pm$ 1.4 | 62.50 $\pm$ 21.99 | 73.75 $\pm$ 15.72 | **75.62** $\pm$ 21.58 |
| | 3m | 6.88 $\pm$ 13.69 | 2.5 $\pm$ 4.07 | 12.50 $\pm$ 18.22 | **68.75** $\pm$ 17.4 | 64.38 $\pm$ 15.4 |
| **Average** | | 48.50 $\pm$ 17.09 | 2.75 $\pm$ 5.19 | 68.38 $\pm$ 11.47 | 86.88 $\pm$ 4.49 | **87.50** $\pm$ 6.54 |

$\diamond$ denotes the source task.
[*] BC-best represents the better per-task and average performance between BC-r and BC-t.

report the per-task and average results in Table 2, where BC-best denotes the better result between BC-t and BC-r. Full results are presented in Appendix I.2.

Since the *Marine-Single* task set only provides *3m-expert* data, all algorithms perform better on tasks with smaller agent scales. However, as the task scale increases, the performance of BC-based methods sharply degrades, whereas CQL-based methods exhibit much less degradation. SD-CQL even maintains an average win rate of 27% on *12m*, which is four times the scale of the source task. Moreover, SD-CQL exhibits the best performance across all test tasks, with average performance far surpassing other baseline algorithms. This indicates that SD-CQL acquires general decision-making skills via skill discovery, achieving higher task efficiency and better scalability to unseen tasks.

For the *Marine-Single-Inv* task set, SD-CQL also demonstrates superior generalization performance on inverse generalization tasks. In contrast, CQL completely collapses due to error accumulation in tasks with large numbers of agents. The BC methods fail on smaller-scale tasks despite good performance on source tasks, as they do not learn generalizable skills. ODIS's skill-learning mechanism alleviates the generalization difficulties inherent in pure BC. However, since its final actions still rely on BC-generated outputs, its performance remains considerably below that of SD-CQL. The average performance of HiSSD trails only SD-CQL. However, the total number of trainable parameters in HiSSD is more than four times that of ODIS and SD-CQL, further highlighting the simplicity and efficiency of SD-CQL.

## 4.3 SKILL-DISCOVERY VISUALIZATION

To provide a more intuitive demonstration of the multi-task generalization capability of SD-CQL, we visualize part of the decision scenarios of the source task *3m* and the target task *12m* from the *Marine-Single* task set in Figure 2. Specifically, we record the battle replays (2a and 2b) and project the corresponding skill vectors $z$ at each decision point onto a two-dimensional plane using t-SNE (Van der Maaten & Hinton, 2008) (2c, where each marker represents the skill vector $z$ chosen by an agent at the moment recorded in 2a and 2b). More details are available in Appendix E.

It can be observed that SD-CQL successfully learns two skills, ***retreat*** and ***attack***, in the *3m* task and applies them effectively in the unseen *12m* task. When the healthy level is low, the agents choose to ***retreat***, while when health is sufficient, they opt to ***attack***, which is consistent with the visualization in 2c. This demonstrates that SD-CQL indeed discovers effective skills. Notably, in *12m*, agents adopting the ***attack*** skill exhibit two specific actions: *firing* and *advancing*, further supporting that SD-CQL extracts cross-task generalizable decision structures instead of mimicking specific actions. We provide evidence in Appendix E that the *advancing* action can be classified as an ***attack*** skill.

Additionally, we demonstrate the more complex behavioral patterns exhibited by SD-CQL on the unseen task *4s3z* from the heterogeneous task set *Stalker-Zealot-Expert* in Appendix E.

## 5 CONCLUSION AND FUTURE DIRECTIONS

In this paper, we address the challenging and underexplored problem of multi-task offline MARL by proposing a novel task-efficient algorithm, Skill-Discovery Conservative Q-Learning (SD-CQL). It discovers skills offline via next-observation reconstruction and trains scalable policies via conservative Q-learning with local value calibration. We conduct substantial experiments on StarCraft II to present the superior generalization and task-efficiency of SD-CQL: It achieves the best performance on **13** out of 14 task sets, with up to **68.9%** improvement on individual task sets. The results demonstrate value-based method is not only viable but superior in this setting.

Our work suggests several directions for future research, including deriving theoretical insights into the role of skill-discovery in generalization and task efficiency; integrating with alternative skill discovery methods; enabling offline-to-online fine-tuning on target tasks. We hope this study encourages further exploration of multi-task offline MARL to advance multi-agent decision-making.

## REPRODUCIBILITY STATEMENT

To ensure reproducibility, we detail our experimental setup in Section 4 and Appendix D, covering data collection, task configuration, implementation details, hyperparameters, and computational resources. The source code will be made publicly available upon publication.

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

# Technical Appendices

# A    LLM USAGE

In this paper, we use the LLM to polish writing and check grammar issues.

# B    RELATED WORK

**MARL**    Multi-Agent Reinforcement Learning (MARL) has seen substantial progress in recent years, with numerous approaches developed under different paradigms. The centralized training with decentralized execution (CTDE) paradigm (Oliehoek et al., 2008; Matignon et al., 2021; Amato, 2024) has been particularly influential, with methods such as HASAC (Liu et al., 2024a), MAPPO (Yu et al., 2022), QMIX (Rashid et al., 2020), VDN (Sunehag et al., 2018), and MADDPG (Lowe et al., 2017). These approaches use centralized training for better coordination and decentralized execution for real-time decision-making. On the other hand, fully decentralized training and execution schemes have also been explored (Tampuu et al., 2017; de Witt et al., 2020). However, the performance of such methods is often constrained by the absence of information sharing. In this paper, we mainly focus on the CTDE paradigm with QMIX as the backbone.

**Offline MARL**    Due to the absence of online interaction with the environment, offline training faces a fundamental challenge—distributional shift. To address this issue, several techniques for single-agent RL have been proposed, many of which leverage conservatism to regularize either the policy (Fujimoto & Gu, 2021; Kostrikov et al., 2021) or the Q-value function (Kumar et al., 2020; Kostrikov et al., 2022; Rezaeifar et al., 2022). These methods mitigate the risks of overestimating the value of unseen state-action pairs. However, specific challenges caused by multiple agents, such as the exponential explosion of complexity, hinder these techniques from directly extending to multi-agent scenarios. Therefore, several tailored approaches (Jiang & Lu, 2021; Yang et al., 2021; Pan et al., 2022; Li et al., 2023; Shao et al., 2023; Liu et al., 2024b) have been proposed to address offline MARL. However, these methods often focus excessively on source tasks, inevitably compromising their multi-task generalization ability.

**Multi-task MARL**    Current research on multi-task MARL can be broadly categorized into two types. The first, often referred to as Ad-Hoc Teamwork (Stone et al., 2010; Zhang et al., 2023b), focuses on exploring how to effectively collaborate with unknown and uncontrollable teammates within a given task. The second type, which is the primary focus of this paper, involves scenarios where the algorithm controls the entire team, but the agents are trained on some tasks and tested on unseen tasks with similar structures (Omidshafiei et al., 2017; Wang et al., 2020; Wu et al., 2024). This requires agents to learn and utilize generalizable decision-making structures across tasks from a limited set of source tasks, positioning skill discovery as a promising solution. However, most existing approaches discover skills typically through sample reconstruction (Wang et al., 2021; Liu et al., 2022; Yang et al., 2023), clustering (Li et al., 2025), or sub-task decomposition (Tian et al., 2023), primarily relying on online interactions, with limited attention to offline settings. VO-MASD (Chen et al., 2024) explores offline skill discovery with a codebook. However, its emphasis lies in training high-level policies online using the discovered skills, rather than in a fully offline setting. Recently, ODIS (Zhang et al., 2023a) proposed a method for multi-task offline MARL, but it suffers from the limitations of behavior cloning when the scale of source tasks is small. Inspired by ODIS, HiSSD (Liu et al., 2025) leverages value functions to separately extract common and task-specific skills, improving generalization performance. However, its task efficiency remains constrained by the BC action decoder, which hinders the discovery of effective skills from a limited number of source tasks.

## C  ALGORITHM PSEUDOCODE

---

**Algorithm 1** Skill-Discovery Conservative Q-Learning

---

**Input:** Datasets of source tasks $\{\mathcal{D}_m\}_{m \in \mathbb{T}_s}$.
Initialize the parameters $\phi$ for the encoder and decoder, $\theta$ for $Q_{tot}$ and $\bar{\theta}$ for target $Q_{tot}$.
**for** $i = 1$ **to** $T_{\max}$ **do**
    **for** $j = 1$ **to** $|\mathbb{T}_s|$ **do**
        Sample trajectories $\hat{\tau} = \{(s_t, o_t, a_t, r_t, o'_t)\}$ from $\mathcal{D}_j$.
        Calculate the total loss $\mathcal{L}_j$ by $\hat{\tau}$ for task $j$, according to equation 9.
    **end for**
    Calculate the multi-task loss $\mathcal{L} = \sum_{j=1}^{|\mathbb{T}_s|} \mathcal{L}_j$.
    Update $\phi$ with $\nabla_\phi \mathcal{L}$ and update $\theta$ with $\nabla_\theta \mathcal{L}$
    **if** $i \equiv 0 \pmod{T_{\text{update}}}$ **then**
        $\bar{\theta} \leftarrow \theta$
    **end if**
**end for**

---

## D  EXPERIMENT DETAILS

### D.1  DATASETS

Following the definition by (Fu et al., 2020), we conduct experiments primarily on datasets of four qualities. The collection procedure for each quality is as follows:

- Expert: Trajectories collected from policies trained to an expert level using QMIX.
- Medium: Trajectories collected from policies trained to a medium level using QMIX.
- Medium-Replay: The replay buffer generated while training QMIX policies to a medium level.
- Medium-Expert: A 50-50 mixture of Medium and Expert trajectories.

To ensure fair comparisons, we use the datasets provided by ODIS (Zhang et al., 2023a), where only up to $2,000$ trajectories are sampled for each dataset. The key information of the datasets used in our experiments is summarized in Table 3.

### D.2  TASKS CONFIGURATION

In this section, we present the key information of all tasks involved in our experiments in Table 4 and the task composition of each task set in Table 5.

### D.3  IMPLEMENTATION DETAILS

#### D.3.1  DECOMPOSER

To handle variable-sized inputs across different tasks, we follow prior works (Hu et al., 2021; Tian et al., 2023; Zhang et al., 2023a; Liu et al., 2025) and adopt a rule-based decomposer to split the input observations or states into individual entity information units. This is feasible because SMAC provides a well-defined and consistent structure for states and observations across tasks.

Specifically, taking observations as an example, each agent's observation is a concatenation of its own features, the features of all allies, and the features of all enemies. While the feature schema for each entity is fixed and standardized, the number of allies and enemies depends on the task configuration. For entities that are not observable (e.g., outside the agent's field of view), their corresponding feature fields are zero-masked. This allows a deterministic rule-based decomposer to segment the observation into individual entity features based on the known observation layout and the number of agents involved in the current task.

Table 3: The primary information of the offline datasets used in our experiments.

| Task | #Trajectories | Average Return |
|------|---------------|----------------|
| 3m-Expert | 2,000 | 19.87 |
| 3m-Medium | 2,000 | 14.00 |
| 3m-Medium-Replay | 2,000 | 10.71 |
| 3m-Medium-Expert | 2,000 | 16.94 |
| 5m-Expert | 2,000 | 19.93 |
| 5m-Medium | 2,000 | 17.35 |
| 5m-Medium-Replay | 1,266 | 3.21 |
| 5m-Medium-Expert | 2,000 | 18.66 |
| 10m-Expert | 2,000 | 19.89 |
| 10m-Medium | 2,000 | 16.63 |
| 10m-Medium-Replay | 331 | 2.30 |
| 10m-Medium-Expert | 2,000 | 18.27 |
| 5m_vs_6m-Expert | 2,000 | 17.34 |
| 5m_vs_6m-Medium | 2,000 | 12.63 |
| 5m_vs_6m-Medium-Replay | 2,000 | 9.41 |
| 5m_vs_6m-Medium-Expert | 2,000 | 14.98 |
| 9m_vs_10m-Expert | 2,000 | 19.59 |
| 9m_vs_10m-Medium | 2,000 | 15.52 |
| 9m_vs_10m-Medium-Replay | 2,000 | 11.76 |
| 9m_vs_10m-Medium-Expert | 2,000 | 17.49 |
| 2s3z-Expert | 2,000 | 19.78 |
| 2s3z-Medium | 2,000 | 16.61 |
| 2s3z-Medium-Replay | 2,000 | 14.25 |
| 2s3z-Medium-Expert | 2,000 | 18.26 |
| 2s4z-Expert | 2,000 | 19.73 |
| 2s4z-Medium | 2,000 | 16.85 |
| 2s4z-Medium-Replay | 2,000 | 14.38 |
| 2s4z-Medium-Expert | 2,000 | 18.26 |
| 3s5z-Expert | 2,000 | 19.78 |
| 3s5z-Medium | 2,000 | 16.31 |
| 3s5z-Medium-Replay | 2,000 | 15.29 |
| 3s5z-Medium-Expert | 2,000 | 18.05 |

Importantly, the decomposer operates solely based on environment-level specifications rather than task-level ones. And, as discussed in the main text, our study focuses on a series of structurally similar tasks that differ in scale, difficulty, and objectives. Therefore, although SMAC explicitly provides the number of interactable entities and the total number of other entities, similar information can be obtained in other environments using techniques such as entity detection or masking mechanisms. Hence, this component does not limit the generality or contribution of SD-CQL.

### D.3.2 SD-CQL

**Skill Discovery.** In the skill discovery part, we use single-layer Transformers for both the encoder and the decoder, and fully connected layers for embedding and skill extraction. To make the skill vector extract more information, the decoder reconstructs the embedding associated with the agent itself directly with $z$:

$$\widehat{E}_{i,0} = W_0 \cdot \text{ReLU}(z) + b_0.$$

While for embeddings of other entities, it first masks parts of the information by a ReLU function, and then concatenates them with $z$ for reconstruction:

$$\widehat{E}_{i,k} = W_k \cdot [\text{ReLU}(E_{i,k}),\ z] + b_k, \quad k = 1, 2, \ldots, K_i \tag{10}$$

where $[\cdot,\ \cdot]$ represents the concatenation operation.

**Skill-conditioned Policy Optimization.** In the skill-conditioned policy optimization part, we use MLPs as the individual Q network, and similar to ODIS, we employ an attention-based mixing

Table 4: The key information about the tasks used in our experiments.

| Task Name | Allied Units | Enemy Units |
|-----------|--------------|-------------|
| 3m | 3 Marines | 3 Marines |
| 4m | 4 Marines | 4 Marines |
| 5m | 5 Marines | 5 Marines |
| 6m | 6 Marines | 6 Marines |
| 7m | 7 Marines | 7 Marines |
| 8m | 8 Marines | 8 Marines |
| 9m | 9 Marines | 9 Marines |
| 10m | 10 Marines | 10 Marines |
| 11m | 11 Marines | 11 Marines |
| 12m | 12 Marines | 12 Marines |
| 5m_vs_6m | 5 Marines | 6 Marines |
| 6m_vs_7m | 6 Marines | 7 Marines |
| 7m_vs_8m | 7 Marines | 8 Marines |
| 8m_vs_9m | 8 Marines | 9 Marines |
| 9m_vs_10m | 9 Marines | 10 Marines |
| 10m_vs_11m | 10 Marines | 11 Marines |
| 10m_vs_12m | 10 Marines | 12 Marines |
| 13m_vs_15m | 13 Marines | 15 Marines |
| 1s3z | 1 Stalkers, 3 Zealots | 1 Stalkers, 3 Zealots |
| 1s4z | 1 Stalkers, 4 Zealots | 1 Stalkers, 4 Zealots |
| 1s5z | 1 Stalkers, 5 Zealots | 1 Stalkers, 5 Zealots |
| 2s3z | 2 Stalkers, 3 Zealots | 2 Stalkers, 3 Zealots |
| 2s4z | 2 Stalkers, 4 Zealots | 2 Stalkers, 4 Zealots |
| 2s5z | 2 Stalkers, 5 Zealots | 2 Stalkers, 5 Zealots |
| 3s3z | 3 Stalkers, 3 Zealots | 3 Stalkers, 3 Zealots |
| 3s4z | 3 Stalkers, 4 Zealots | 3 Stalkers, 4 Zealots |
| 3s5z | 3 Stalkers, 5 Zealots | 3 Stalkers, 5 Zealots |
| 4s3z | 4 Stalkers, 3 Zealots | 4 Stalkers, 3 Zealots |
| 4s4z | 4 Stalkers, 4 Zealots | 4 Stalkers, 4 Zealots |
| 4s5z | 4 Stalkers, 5 Zealots | 4 Stalkers, 5 Zealots |
| 4s6z | 4 Stalkers, 6 Zealots | 4 Stalkers, 6 Zealots |

Table 5: The task composition of each scenario used in our experiments.

| Scenarios | Source Tasks | Unseen Tasks |
|-----------|--------------|--------------|
| Marine-Easy | 3m, 5m, 10m | 4m, 6m, 7m, 8m, 9m, 11m, 12m |
| Marine-Hard | 3m, 5m_vs_6m, 9m_vs_10m | 4m, 5m, 10m, 12m, 6m_vs_7m, 7m_vs_8m, 8m_vs_9m, 10m_vs_11m, 10m_vs_12m, 13m_vs_15m |
| Stalker-Zealot | 2s3z, 2s4z, 3s5z | 1s3z, 1s4z, 1s5z, 2s5z, 3s3z, 3s4z, 4s3z, 4s4z, 4s5z, 4s6z |
| Marine-Single | 3m | 5m, 8m, 10m, 12m |
| Marine-Single-Inv | 10m | 3m, 4m, 5m, 8m |

network to handle variable inputs in multi-task training. The entity decomposition method and its dimensions are the same as those used in the observation reconstruction. The specific parameters of the network structure are shown in Table 6. For Conservative Q-Learning, the sampling strategy $\mu$ we use is a uniform distribution over the action space, with 100 samples drawn to estimate the CQL regularization term. Additionally, consistent with the QMIX implementation provided by PYMARL2 (Hu et al., 2023), we use TD($\lambda$) to enhance stability when calculating the TD loss, and the $\lambda$ parameter is set to the default value from PYMARL2.

Table 6: The main network structure involved in SD-CQL.

| Network | Architecture | Activation Function |
|---|---|---|
| Transformer | Head=1, Depth=1, Embedding Dim=64 | —— |
| Fixed Individual Q Network | [96, 128, 6] | ReLU |
| Variable Individual Q Network | [96, 128, 1] | ReLU |
| HyperNet in Mixing Network | [128, 64, 32] | ReLU |
| Attention in Mixing Network | Attention Dim=8 | —— |

### D.3.3 BASELINES

For the baselines except for HiSSD, we utilize the implementations provided by ODIS. Specifically, BC-t employs a transformer capable of decomposing observations for multi-task training while optimizing the policy through behavior cloning. BC-r is similar to BC-t but incorporates return-to-go information as an additional input alongside observations. The vanilla CQL is almost the same as SD-CQL, but it removes the skill-discovery module and the local value calibration term. It is only equipped with the observation encoder, and feeds the encoding vectors $\{E_{i,k}\}_{k=0}^{K_i}$ to separated Q-networks directly. Additionally, in vanilla CQL, the gradient can propagate from the encoder to the Q-networks. The implementations of ODIS and HiSSD align with their official version.

**Reproduction Validation.** Although all baselines, when applicable, are implemented using the official codebases and the hyperparameters recommended in the original papers, we provide a side-by-side comparison on Multi-to-Multi task sets between our reproduced results and the numbers reported in the corresponding works in Tables 7, 8, and 9 for validation. Due to factors such as randomness and instability, we observe certain discrepancies. For example, the BC-best scores in our reproduction are noticeably higher than those reported by Zhang et al. (2023a). In addition, both our results and those in the ODIS paper (Zhang et al., 2023a) are based on the final policy, whereas the HiSSD paper (Liu et al., 2025) reports the best-performing checkpoint, which may further contribute to the differences.

Table 7: Comparison between the original results and our reproduced results for BC-best.

| Task | BC-best (Ours), Final | BC-best (Zhang et al., 2023a), Final |
|---|---|---|
| *marine-easy-e* | 99.38 | 82.42 |
| *marine-easy-m* | 71.81 | 70.57 |
| *marine-easy-mr* | 49.62 | 16.07 |
| *marine-easy-me* | 75.25 | 58.52 |
| *marine-hard-e* | 64.84 | 55.40 |
| *marine-hard-m* | 42.08 | 31.81 |
| *marine-hard-mr* | 46.98 | 42.55 |
| *marine-hard-me* | 51.93 | 39.51 |
| *stalker-zealot-e* | 68.61 | 67.15 |
| *stalker-zealot-m* | 23.94 | 21.84 |
| *stalker-zealot-mr* | 17.64 | 13.54 |
| *stalker-zealot-me* | 39.62 | 40.98 |

### D.4 HYPERPARAMETERS

For SD-CQL, the main hyperparameters shared across all tasks are listed in Table 10. We primarily adjust the local value calibration coefficient $\eta$, with the specific settings provided in Table 11. All other hyperparameters are kept consistent with the default configuration of QMIX in PYMARL2.

*Remark: $\eta$ is an important hyperparameter for SD-CQL to balance in-task performance and cross-task generalization. A practical tuning guideline is to start from a moderate value (e.g., $\eta = 0.5$).*

Table 8: Comparison between the original results and our reproduced results for ODIS.

| Task | ODIS (Ours), Final | ODIS (Zhang et al., 2023a), Final |
|------|------|------|
| *marine-easy-e* | 70.50 | 85.83 |
| *marine-easy-m* | 68.50 | 75.3 |
| *marine-easy-mr* | 11.06 | 11.25 |
| *marine-easy-me* | 66.00 | 71.6 |
| *marine-hard-e* | 21.09 | 55.02 |
| *marine-hard-m* | 32.55 | 41.43 |
| *marine-hard-mr* | 37.29 | 43.07 |
| *marine-hard-me* | 19.58 | 42.33 |
| *stalker-zealot-e* | 55.05 | 67.27 |
| *stalker-zealot-m* | 11.97 | 27.52 |
| *stalker-zealot-mr* | 8.17 | 11.72 |
| *stalker-zealot-me* | 30.72 | 40.27 |

Table 9: Comparison between the original results and our reproduced results for HiSSD.

| Task | HiSSD (Ours), Final | HiSSD (Liu et al., 2025), Best |
|------|------|------|
| *marine-easy-e* | 97.94 | 98.64 |
| *marine-easy-m* | 71.44 | 78.74 |
| *marine-easy-mr* | 72.88 | 79.97 |
| *marine-easy-me* | 78.31 | 74.34 |
| *marine-hard-e* | 68.02 | 68.15 |
| *marine-hard-m* | 47.60 | 49.02 |
| *marine-hard-mr* | 47.55 | 48.59 |
| *marine-hard-me* | 53.28 | 59.53 |
| *stalker-zealot-e* | 66.59 | 73.84 |
| *stalker-zealot-m* | 23.51 | 18.42 |
| *stalker-zealot-mr* | 15.00 | 12.94 |
| *stalker-zealot-me* | 40.29 | 43.91 |

*When we need to reduce Q-value estimation error, for example, with lower-quality datasets or larger agent populations, a larger $\eta$ tends to work better. Conversely, when Q-value can be estimated well, such as with higher-quality data or fewer agents, a smaller $\eta$ is generally more favorable for generalization.*

Table 10: The hyperparameters shared by all tasks for SD-CQL.

| Hyperparameter | Value |
|------|------|
| Entity Embedding Dim | 64 |
| Skill Dimension of $z$ | 32 |
| $\lambda$ for $TD(\lambda)$ | 0.6 |
| Conservative Weight $\alpha$ | 1.0 |
| $T_{\max}$ | 35,000 |
| $T_{\text{update}}$ | 80 |
| Batch Size | 32 |
| Learning Rate | 0.005 |
| Optimizer | Adam (Kingma, 2014) |

Since the *Marine-Single* and *Marine-Single-Inv* task sets designed by us, we adjust the primary hyperparameters involved in the RL losses of ODIS and HiSSD. For ODIS, we tune the conservative coefficient $\alpha$ (selected from $\{1.0, 2.5, 5.0\}$) and the distribution coefficient $\lambda$ (from $\{1.0, 2.5, 5.0\}$). For HiSSD, due to training on only one source task, the contrastive loss cannot be applied. Con-

Table 11: Local value calibration coefficient $\eta$ for each task set.

| Task set | Value |
|----------|-------|
| Marine-Easy-Expert | 0.9 |
| Marine-Easy-Medium | 0.2 |
| Marine-Easy-Medium-Replay | 0.5 |
| Marine-Easy-Medium-Expert | 0.5 |
| Marine-Hard-Expert | 0.5 |
| Marine-Hard-Medium | 0.8 |
| Marine-Hard-Medium-Replay | 0.9 |
| Marine-Hard-Medium-Expert | 0.7 |
| Stalker-Zealot-Expert | 0.3 |
| Stalker-Zealot-Medium | 0.1 |
| Stalker-Zealot-Medium-Replay | 0.5 |
| Stalker-Zealot-Medium-Expert | 0.3 |
| Marine-Single | 0.5 |
| Marine-Single-Inv | 0.3 |

sequently, we only tune the planner weight $\alpha$ (selected from $\{5.0, 10.0, 20.0\}$) and the expectile regression loss parameter $\epsilon$ (from $\{0.5, 0.7, 0.9\}$).

We select the best configuration of each for reporting. The final hyperparameter settings are provided in Table 12, while Table 13 and Table 14 present the average performance of all configurations we explored.

Table 12: Selected Hyperparameters for ODIS and HiSSD.

| Task set | ODIS | HiSSD |
|----------|------|-------|
| *Marine-Single* | $\alpha = 5.0, \lambda = 1.0$ | $\alpha = 10.0, \epsilon = 0.9$ |
| *Marine-Single-Inv* | $\alpha = 1.0, \lambda = 1.0$ | $\alpha = 5.0, \epsilon = 0.7$ |

Table 13: Average win rates of all configurations explored for ODIS. The reported results are averaged over 5 random seeds. The bold number denotes the best performance, and $\pm$ denotes one standard deviation.

| $\alpha$ | $\lambda$ | Marine-Single | Marine-Single-Inv |
|----------|-----------|---------------|-------------------|
| 1.0 | 1.0 | $27.12 \pm 7.39$ | $\mathbf{68.38} \pm 11.47$ |
| 1.0 | 2.5 | $26.88 \pm 4.56$ | $62.88 \pm 12.78$ |
| 1.0 | 5.0 | $23.38 \pm 3.30$ | $53.25 \pm 13.59$ |
| 2.5 | 1.0 | $31.38 \pm 8.55$ | $61.50 \pm 13.06$ |
| 2.5 | 2.5 | $25.25 \pm 5.99$ | $64.12 \pm 14.06$ |
| 2.5 | 5.0 | $25.25 \pm 2.11$ | $64.00 \pm 2.64$ |
| 5.0 | 1.0 | $\mathbf{33.25} \pm 16.50$ | $66.12 \pm 8.73$ |
| 5.0 | 2.5 | $22.25 \pm 1.29$ | $55.62 \pm 12.23$ |
| 5.0 | 5.0 | $24.62 \pm 4.93$ | $52.12 \pm 17.44$ |

## D.5 COMPUTATIONAL RESOURCES

We run SD-CQL on a single NVIDIA RTX A6000 GPU. Each run takes approximately 5 to 12 hours, depending on the task set (about 5 hours for the Marine-Easy-Expert task set and about 12 hours for the Stalker-Zealot-Medium-Expert task set).

Table 14: Average win rates of all configurations explored for HiSSD. The reported results are averaged over 5 random seeds. The bold number denotes the best performance, and $\pm$ denotes one standard deviation.

| $\alpha$ | $\epsilon$ | Marine-Single | Marine-Single-Inv |
|---|---|---|---|
| 5.0 | 0.5 | $38.25 \pm 13.41$ | $82.38 \pm 4.28$ |
| 5.0 | 0.7 | $43.50 \pm 14.24$ | $\mathbf{86.88} \pm 4.49$ |
| 5.0 | 0.9 | $39.00 \pm 14.09$ | $82.38 \pm 4.82$ |
| 10.0 | 0.5 | $36.75 \pm 7.78$ | $83.25 \pm 5.69$ |
| 10.0 | 0.7 | $40.37 \pm 15.25$ | $74.63 \pm 4.02$ |
| 10.0 | 0.9 | $\mathbf{44.12} \pm 18.89$ | $85.25 \pm 10.33$ |
| 20.0 | 0.5 | $39.88 \pm 12.33$ | $82.12 \pm 3.53$ |
| 20.0 | 0.7 | $37.00 \pm 12.24$ | $83.25 \pm 5.13$ |
| 20.0 | 0.9 | $37.50 \pm 12.14$ | $84.12 \pm 7.67$ |

# E    SKILL-DISCOVERY VISUALIZATION

## E.1    SKILL VISUALIZATION FOR SD-CQL

We visualize the discovered skills as follows: First, we deploy the SD-CQL agent, which has been trained on the *Marine-Single* task set, in the *3m* and *12m* tasks. Then, we run one episode in each task and record the actions and skills of each agent at each time step, while saving replay videos. Next, we collect the skill $z$ of every agent at every time step from both trajectories as samples. Finally, we apply t-SNE (Van der Maaten & Hinton, 2008) to project them into a 2D plane and normalize the two dimensions to align their scales.

In Figure 2c, the markers represent the skills of agents at corresponding time steps and tasks (Figure 2a and Figure 2b) after dimensionality reduction. Red markers represent skills from the *12m* task, green markers represent skills from the *3m* task, triangles indicate the **retreat**, circles indicate the **attack** skill, and crosses indicate dead agents.

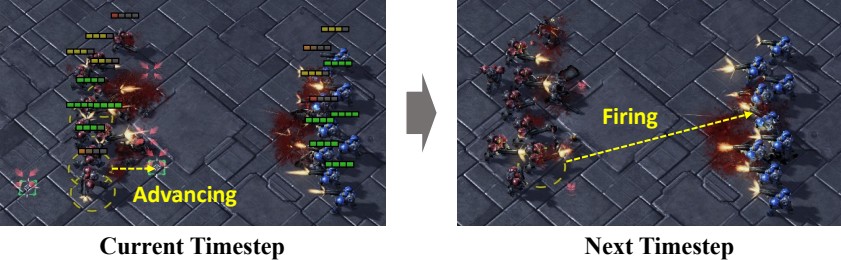

**Current Timestep**                **Next Timestep**

Figure 4: In the *12m* task, the agents first approach the enemy through the *advancing* action and then execute the *firing* action in the next timestep.

In Section 4.3, we mentioned that in the *12m* task, after extracting the **attack** skill, SD-CQL controls agents to perform two specific actions: *firing* and *advancing*. To validate this interpretation, we present in Figure 4 the next-step actions of the marine units that have just executed the *advancing* action. It can be observed that after moving forward to approach the enemies, they immediately perform a *firing* action. This supports the interpretation in Figure 2b that the *advancing* action is also part of the **attack** skill. It also indicates that SD-CQL is capable of extracting a certain general skill (e.g., **attack**) and flexibly executing different specific actions (e.g., *firing* and *advancing*) based on the situation.

## E.2    MULTI-TIMESTEP VISUALIZATION

To illustrate the consistency of SD-CQL's skill-discovery mechanism across multiple timesteps, we present in Figure 5a the game replays at different moments of the *12m* task, alongside the 2D

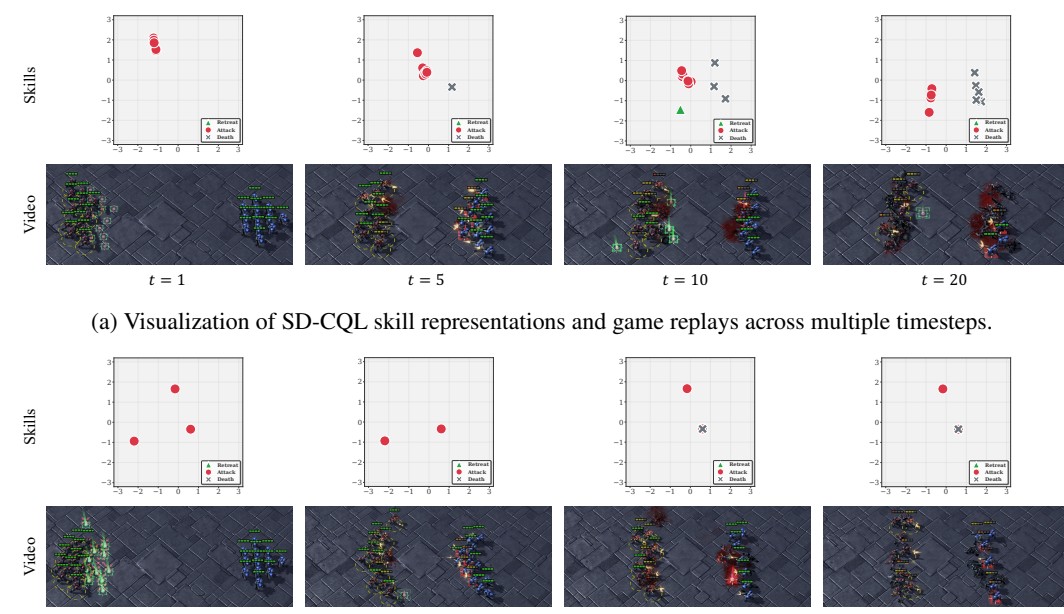

(a) Visualization of SD-CQL skill representations and game replays across multiple timesteps.

(b) Visualization of ODIS skill representations and game replays across multiple timesteps.

Figure 5: Visualization comparison of SD-CQL and ODIS across multiple timesteps in the *12m* task.

representations of the skills selected by each agent. Since HiSSD employs both common and task-specific skills, making a direct comparison with SD-CQL's skills less straightforward, we instead visualize the skills and corresponding game replays of ODIS in Figure 5b using a method similar to that described in Appendix E.1.

By comparing the visualization results of SD-CQL and ODIS in Figure 5, we observe that, due to the limited expressiveness of predefined discrete skills, ODIS fails to distinguish between dead and surviving agents. Moreover, we do not observe any agents performing retreat-like actions, which undermines the performance of ODIS. In contrast, the continuous skill vectors discovered by SD-CQL in the latent space offer much higher expressiveness, successfully differentiating agents in different states. Furthermore, by directly optimizing the skill-conditioned Q-values for specific actions, SD-CQL can flexibly execute different actions based on the situation after extracting certain skills, further enhancing its generalization performance and task efficiency.

### E.3 VISUALIZATION FOR MORE COMPLEX SKILLS

We deploy the SD-CQL policy trained on the heterogeneous *Stalker–Zealot–Expert* task set to the unseen *4s3z* scenario to illustrate the emergence of more complex behaviors. As shown in Figure 6, SD-CQL effectively extracts rich and diverse behavioral patterns from the source-task datasets, including ranged attack, melee attack, focus fire, flanking, and encirclement, and successfully transfers these skills to the unseen task, demonstrating the expressivity of SD-CQL's skill discovery module.

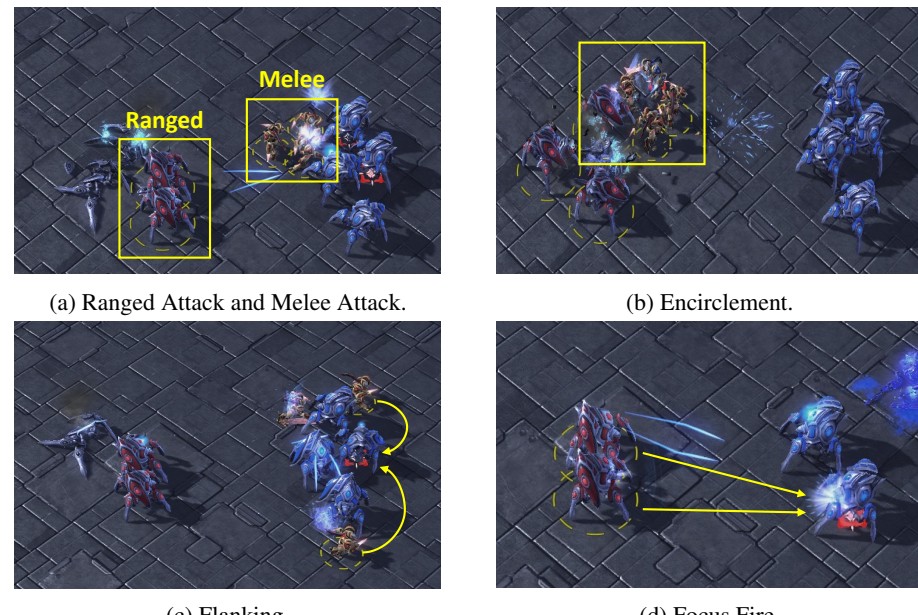

(a) Ranged Attack and Melee Attack.

(b) Encirclement.

(c) Flanking.

(d) Focus Fire.

Figure 6: Visualization of more complex skills used by SD-CQL in the *4s3z* task.

## F  ABLATION STUDY

To investigate the contribution of each component of SD-CQL to its superior performance, as well as its sensitivity to different hyperparameter values, we perform an ablation study on the One-to-Multi task sets.

### F.1  COMPONENT ABLATION

#### F.1.1  KEY COMPONENTS OF SD-CQL

In Table 15, we report the performance of three variants of SD-CQL: (i) without the local value calibration term (w/o LVC), (ii) without the skill discovery mechanism (w/o SD), and (iii) without both (w/o LVC & SD), i.e., vanilla CQL. It can be seen that these two components are indispensable for the superior performance of SD-CQL. Across both task sets, our proposed skill discovery mechanism and corresponding skill-conditioned Q-learning significantly enhance the generalization ability of SD-CQL. Meanwhile, the LVC effectively mitigates error accumulation, particularly in the *Marine-Single-Inv* task set, where the source task involves a large number of agents.

In addition, we observe some interesting phenomena. For example, the w/o SD & LVC variant achieves the second-best performance after SD-CQL on the *Marine-Single* task set, and it also outperforms the w/o LVC variant on the *Marine-Single-Inv* task set. These results may be attributed to differences in the expressiveness of the model.

Specifically, as shown in Figure 3, in w/o LVC, the encoder is updated only by $\mathcal{L}_{\text{Rec}}$ according to equation 3, while Q-values are optimized by $\mathcal{L}_{\text{Q}}$ according to equation 7. However, in w/o SD & LVC, the removal of $\mathcal{L}_{\text{Rec}}$ necessitates eliminating the gradient stop before Q-values shown in Figure 3, enabling the encoder and Q-values to be jointly optimized. As a result, the policy network possesses more trainable parameters, providing greater expressiveness. This explains why w/o SD & LVC outperforms w/o LVC. Overall, SD-CQL introduces local value calibration to bridge skill discovery and Q-learning, thereby enhancing overall cross-task performance and highlighting our key innovation.

Table 15: Win rates of different SD-CQL variants on One-to-Multi task sets. The reported results are averaged over 5 random seeds. The bold number denotes the best performance, and $\pm$ denotes one standard deviation.

| Task Set | | SD-CQL | w/o LVC | w/o SD | w/o LVC & SD |
|---|---|---|---|---|---|
| *Marine Single* | 3m $^\diamond$ | **100.00** $\pm$ 0.00 | 98.12 $\pm$ 2.80 | 99.38 $\pm$ 1.40 | **100.00** $\pm$ 0.00 |
| | 5m | **91.25** $\pm$ 10.92 | 83.75 $\pm$ 17.17 | 60.62 $\pm$ 41.79 | 88.12 $\pm$ 7.46 |
| | 8m | **49.38** $\pm$ 23.63 | 39.38 $\pm$ 17.34 | 33.75 $\pm$ 30.49 | 40.00 $\pm$ 26.00 |
| | 10m | **41.88** $\pm$ 25.83 | 27.50 $\pm$ 18.41 | 31.87 $\pm$ 36.54 | 34.38 $\pm$ 25.39 |
| | 12m | **27.50** $\pm$ 21.13 | 17.50 $\pm$ 14.76 | 10.00 $\pm$ 9.73 | 17.50 $\pm$ 10.27 |
| **Average** | | **62.00** $\pm$ 13.80 | 53.25 $\pm$ 11.47 | 47.12 $\pm$ 20.16 | 56.00 $\pm$ 11.07 |
| *Marine Single-Inv* | 10m $^\diamond$ | **100.00** $\pm$ 0.00 | 0.00 $\pm$ 0.00 | **100.00** $\pm$ 0.00 | 0.62 $\pm$ 1.4 |
| | 8m | **100.00** $\pm$ 0.00 | 0.00 $\pm$ 0.00 | 98.75 $\pm$ 2.8 | 3.75 $\pm$ 8.39 |
| | 5m | **97.50** $\pm$ 2.61 | 0.00 $\pm$ 0.00 | 26.25 $\pm$ 20.32 | 6.25 $\pm$ 13.98 |
| | 4m | **75.62** $\pm$ 21.58 | 0.00 $\pm$ 0.00 | 3.12 $\pm$ 6.99 | 0.62 $\pm$ 1.4 |
| | 3m | **64.38** $\pm$ 15.40 | 0.00 $\pm$ 0.00 | 0.0 $\pm$ 0.0 | 2.5 $\pm$ 4.07 |
| **Average** | | **87.50** $\pm$ 6.54 | 0.00 $\pm$ 0.00 | 45.62 $\pm$ 4.73 | 2.75 $\pm$ 5.19 |

$\diamond$ denotes the source task.

### F.1.2 DESIGN CHOICES IN RECONSTRUCTOR

SD-CQL extracts effective skills by reconstructing the next local observation. To investigate the role of several design choices in the reconstructor, such as the reconstruction target $o_{t+1}$, the temporal latent variable $h$, the skill vector $z$, and the stop-gradient operation, we conduct additional ablation studies. Specifically, we evaluate five variants on the marine-single and marine-single-inv task sets: (i) Reconstructing the current observation instead of the next (Rec $o_t$); (ii) Removing the temporal latent variable (w/o $h$); (iii) Not feeding the skill vector into the Q-network (No $z$ to Q); (iv) Performing reconstruction only without extracting any skill vector (Only Rec); and (v) Enabling gradient flow between the Q-network and the skill-discovery module (w/o stop-gradient).

Apart from the modifications specific to each variant, all other components, such as network architecture and hyperparameters, remain the same as in SD-CQL. We report the average performance of these variants across both task sets in Table 16.

It can be observed that the complete SD-CQL achieves the highest overall performance, and skill extraction is critical to its superior performance: all variants that incorporate the skill vector into the Q-function outperform those that do not. Even when the skill vector is excluded from the Q-function, incorporating it into the reconstruction process still helps extract more informative individual features. At the same time, predicting the next local observation and incorporating temporal latent variables indeed help SD-CQL better capture generalizable decision-making patterns. Moreover, truncating the gradients between the Q-network and the skill-discovery module further helps prevent mutual interference between them.

Table 16: The average win rates of different SD-CQL variants on One-to-Multi task sets. The reported results are averaged over 5 random seeds. The bold number denotes the best performance, and $\pm$ denotes one standard deviation.

| Task Set | SD-CQL | Rec $o_t$ | w/o $h$ | No $z$ to Q | Only Rec | w/o stop-gradient |
|---|---|---|---|---|---|---|
| *Marine-Single* | 62.00 $\pm$ 13.80 | 54.88 $\pm$ 15.27 | **62.25** $\pm$ 16.70 | 62.13 $\pm$ 15.17 | 40.88 $\pm$ 15.17 | 59.13 $\pm$ 14.42 |
| *Marine-Single-Inv* | **87.50** $\pm$ 6.54 | 87.38 $\pm$ 4.06 | 81.38 $\pm$ 8.69 | 76.25 $\pm$ 9.60 | 64.63 $\pm$ 23.69 | 84.38 $\pm$ 8.24 |
| **Average** | **74.75** $\pm$ 6.57 | 71.13 $\pm$ 8.87 | 71.94 $\pm$ 4.89 | 69.19 $\pm$ 9.77 | 52.75 $\pm$ 14.88 | 71.75$\pm$11.35 |

In addition, we validate our decoder design choice of excluding $h$ on more task sets. As shown in Table 17, excluding $h$ from the decoder tends to be more beneficial on the more complex tasks, while having only a minor impact on simpler ones, and it does not change our main performance trends.

Table 17: The average win rates of different decoder variants on 4 task sets. The reported results are averaged over 5 random seeds. The bold number denotes the best performance, and $\pm$ denotes one standard deviation.

| Task Set | Marine-Single | Marine-Single-Inv | Marine-Hard-Medium | Stalker–Zealot-Medium |
|---|---|---|---|---|
| **SD-CQL** | $62.00 \pm 13.80$ | $\textbf{87.50} \pm 6.54$ | $\textbf{48.12} \pm 5.66$ | $\textbf{40.43} \pm 7.06$ |
| **Decoder with $h$** | $\textbf{71.38} \pm 10.52$ | $84.25 \pm 8.68$ | $46.2 \pm 4.56$ | $38.03 \pm 10.12$ |

### F.2 HYPERPARAMETER SENSITIVITY

In Table 18, we separately adjusted one of the weights of the CQL regularization $\alpha$ and the LVC $\eta$ while keeping the other consistent with that in the evaluation. It can be observed that the performance of SD-CQL remains relatively stable and insensitive to the values of these two hyperparameters on the informative *Marine-Single-Inv* task set. In contrast, on the more limited *Marine-Single* task set, hyperparameter tuning requires slightly more care. However, it is not particularly challenging, as it is reasonable to adopt a fairly conservative $\alpha$ on expert datasets and apply less LVC regularization $\eta$ when dealing with a small number of agents, given the lower level of accumulated error compared to *Marine-Single-Inv*.

Table 18: Win rates of SD-CQL with different hyperparameters on One-to-Multi task sets. The reported results are averaged over 5 random seeds. The $\pm$ denotes one standard deviation.

| Hyperparameters | | $\eta = 0.7$ | $\eta = 0.5$ | $\eta = 0.3$ |
|---|---|---|---|---|
| **Average** | *Marine-Single* | $45.50 \pm 9.32$ | $62.00 \pm 13.8$ | $62.50 \pm 16.28$ |
| | *Marine-Single-Inv* | $83.50 \pm 8.67$ | $84.00 \pm 9.64$ | $87.50 \pm 6.54$ |
| **Hyperparameters** | | $\alpha = 0.5$ | $\alpha = 1.0$ | $\alpha = 2.5$ |
| **Average** | *Marine-Single* | $51.88 \pm 12.52$ | $62.00 \pm 13.8$ | $56.12 \pm 13.45$ |
| | *Marine-Single-Inv* | $81.12 \pm 4.43$ | $87.50 \pm 6.54$ | $78.88 \pm 10.69$ |

## G  ADDITIONAL RESULTS FOR CONTINUOUS-ACTION ENVIRONMENTS

Although SD-CQL is primarily designed for discrete-action settings, in this section we further investigate whether its generalization capability remains effective when extended to continuous-action environments.

### G.1  SD-CQL FOR CONTINUOUS ACTION

One intuitive approach to extending SD-CQL to the continuous-action setting is to retain the skill discovery module and replace the QMIX-based policy optimization with an actor–critic architecture similar to FACMAC (Peng et al., 2021). Specifically, both the actors and the local critics additionally take the latent skill vector $z$ as input, while the global critic is computed by passing all local critic outputs through a QMIX-style mixing network. The LVC regularization term can be adapted by adding an mean-squared error behavior regularizer between the actor's output and the actions in the offline dataset.

### G.2  EXPERIMENT

**Tasks and Datasets**  We build our experiments on the *simple-spread* environment from the Multi-Agent Particle Environment (MPE) (Lowe et al., 2017). Specifically, we consider four tasks: *SS-2*, *SS-3*, *SS-4*, and *SS-5*, where *SS-N* denotes an environment with $N$ agents and $N$ landmarks, and each agent must navigate to a unique landmark while avoiding collisions. We treat *SS-2* as the

source task and use MATD3 (Ackermann et al., 2019) to train a medium-level policy, collecting $2,000$ trajectories as the offline dataset. The remaining three tasks serve as unseen tasks for zero-shot generalization evaluation.

**Experiment Setup** We use the same baselines as in Section 4, but re-implement their continuous-action versions. For example, for BC-based methods we replace the cross-entropy loss with a mean-squared error objective, and for CQL we adopt the same actor–critic architecture used in the continuous version of SD-CQL. All algorithms are trained offline for $35,000$ steps on the source-task dataset, with ODIS receiving an additional $15,000$ pretraining steps. During evaluation, each test environment is run for 32 episodes, and we report the average return. We report the average performance of the final policy across five random seeds.

**Results** As shown in Table 19, even under continuous-action environment and with an intuitive extension, SD-CQL still achieves the best average performance across tasks. Meanwhile, although BC performs well on the source task, its generalization degrades as the number of agents increases, which matches our observations in discrete-action settings. Furthermore, we would also like to emphasize that achieving effective multi-task offline MARL in continuous-control environments remains an open and challenging future direction.

Table 19: Win rates of continuous-action environments. The results are averaged over 5 random seeds. Bold numbers indicate the best performance, and $\pm$ denotes one standard deviation.

| Task | BC-t | BC-r | CQL | ODIS | HiSSD | SD-CQL (Ours) |
|------|------|------|-----|------|-------|---------------|
| SS-2 $^\diamond$ | **-114.89** $\pm$ 2.25 | -168.94 $\pm$ 20.75 | -322.61 $\pm$ 36.93 | -122.19 $\pm$ 5.2 | -126.29 $\pm$ 5.9 | -121.07 $\pm$ 2.72 |
| SS-3 | -232.62 $\pm$ 5.72 | -356.87 $\pm$ 88.11 | -641.45 $\pm$ 89.99 | -224.62 $\pm$ 7.64 | -247.59 $\pm$ 16.98 | **-224.45** $\pm$ 12.03 |
| SS-4 | -378.52 $\pm$ 38.54 | -569.22 $\pm$ 199.68 | -1063.86 $\pm$ 157.31 | **-354.97** $\pm$ 16.05 | -408.62 $\pm$ 60.12 | -357.15 $\pm$ 24.98 |
| SS-5 | -581.95 $\pm$ 81.39 | -838.31 $\pm$ 283.33 | -1552.23 $\pm$ 248.58 | -535.84 $\pm$ 34.75 | -522.73 $\pm$ 36.07 | **-494.2** $\pm$ 27.62 |
| **Average** | -327.0 $\pm$ 25.77 | -483.34 $\pm$ 131.68 | -895.04 $\pm$ 117.72 | -309.41 $\pm$ 9.99 | -326.31 $\pm$ 16.15 | **-299.22** $\pm$ 14.37 |

$\diamond$ denotes the source task.

# H DISCUSSIONS

## H.1 ASSUMPTIONS AND SCOPE

We consider the standard multi-task MARL setting, in line with prior work (Hu et al., 2021; Zhang et al., 2023a; Chen et al., 2024; Liu et al., 2025), where tasks, whether in online or offline regimes, share a similar and well-defined observation structure that enables cross-task generalization of policy models. When this assumption is violated, such as in less structured, vision-based settings, ambiguous or incorrect entity decomposition may hinder performance for existing approaches, including SD-CQL. This challenge is shared across current multi-task MARL methods. Nevertheless, techniques such as entity segmentation and representation learning can transform noisy or unstructured inputs into structured representations that SD-CQL can exploit to learn generalizable policies. We leave this as an important direction for future work.

Even under well-defined observations, multi-task offline MARL remains highly challenging and largely open. Our experiments show that existing methods often fall short in this regime, while SD-CQL delivers substantial gains and establishes a new state of the art.

## H.2 DETAILED ARCHITECTURE COMPARISON WITH ODIS, HiSSD, AND VO-MASD

### H.2.1 COMPARISON WITH ODIS

ODIS requires a pre-specified number of skills and learns a skill classifier and action decoder by reconstructing actions from global states. During policy training, it freezes the classifier, retrains a local encoder, and uses CQL to choose skills, decoding actions through the decoder.

In contrast, SD-CQL directly discovers skills in the latent space via a local reconstructor. It then chooses actions using skill-conditioned CQL with local value calibration. This eliminates the need for predefined skill numbers or retraining, and achieves better generalization and task efficiency.

### H.2.2 COMPARISON WITH HiSSD

HiSSD comprises a high-level planner and a low-level controller. The high-level planner predicts the distribution of the next global state from the current observation and trains an IQL-style value function to weight the likelihood loss, thereby extracting common skills. The low-level controller learns task-specific skills via contrastive learning and generates actions through a BC decoder conditioned on the observation and both types of skills.

Although HiSSD also operates in a latent skill space, it exhibits several limitations compared to SD-CQL. First, like ODIS, it relies on BC imitation for action generation, limiting its ability to exceed the behavior policy. Second, its contrastive learning mechanism cannot be applied when only a single source task is available. Finally, its complex architecture inflates the parameter count to over four times that of ODIS and SD-CQL, leading to longer training times and lower efficiency.

In contrast, SD-CQL employs a simpler yet more effective design: it discovers skills through observation reconstruction and directly optimizes skill-conditioned Q-values with local value calibration, achieving better generalization and task efficiency with significantly fewer parameters and shorter training durations.

### H.2.3 COMPARISON WITH VO-MASD

VO-MASD targets multi-task MARL by discovering offline skills via an improved VQ-VAE and then training target-task high-level policies online with a hierarchical PPO. This is fundamentally different from our multi-task offline setting, where learning is completed without any online interaction and policies are deployed zero-shot.

Beyond the training protocol, the core technical designs also diverge. VO-MASD reconstructs actions from a predefined number of codebooks, implicitly assuming a fixed discrete skill inventory.

In contrast, SD-CQL reconstructs the next-step observation in a continuous latent space, enabling flexible and scalable skill discovery without fixing the number of skills, and then optimizes skill-conditioned policies entirely offline. Our results show that, when coupled with next-observation–based offline skill discovery and regularized by local value calibration, value-based method is not only viable but superior in multi-task offline MARL.

Nevertheless, exploring alternative skill-discovery modules within SD-CQL, as well as hierarchical offline-to-online extensions, is a promising direction for future work.

### H.3 "TASK LABEL" IN (OFFLINE) MULTI-TASK MARL AND CONVENTIONAL RL

The "task label" is an essential concept in multi-task RL, as it explicitly indicates the task an agent is solving and guides policy learning across diverse objectives. Although multi-task MARL (MT-MARL) typically lacks such explicit task labels, the underlying tasks can still be related to those in conventional multi-task RL by abstracting them at two levels. The first level focuses on decision patterns based on the current team state and executing corresponding actions within a specific task, and the second level corresponds to team-level objectives that differ across tasks. The core objective of MT-MARL is to discover and leverage the decision patterns that generalize across tasks to enable multi-task generalization and task efficiency.

Therefore, the concept of "task label" in MT-MARL may share some relationship with these first-level decision patterns, i.e., "skills". As illustrated in Figure 1, the agents' behaviors naturally cluster into distinct skills, and they must infer the appropriate skill from the current observation and accurately execute the corresponding actions, which is fundamentally similar to conventional single-agent multi-task RL, where different task objectives require different action policies.

However, note that the "skills" are not exactly equivalent to "task labels" in the conventional sense. Since task labels in conventional multi-task RL settings are typically discrete, pre-defined, and well-specified, whereas in MT-MARL, the skill boundaries are usually not explicitly available. As shown in Figures 1b and 5, prior methods that predefine the number of skills and discover them through classification struggle to accurately capture the characteristics of each skill, thereby hindering generalization. In contrast, leveraging continuous representations in the latent space enables more ef-

fective identification of these decision patterns and improves generalization. This also highlights that the design of SD-CQL is tailored to address the unique challenges of MT-MARL.

# I DETAILED RESULTS OF MULTI-TO-MULTI TASK SETS

## I.1 MULTI-TO-MULTI TASK SETS

### I.1.1 SEPARATED AVERAGE PERFORMANCE FOR SOURCE AND UNSEEN TASKS

In Table 1, we report averages over all tasks to assess each algorithm's overall capability across both source and unseen tasks. For completeness, Table 20 lists the averages on the unseen tasks only. As shown, SD-CQL exhibits a more pronounced advantage on 13 task sets. We also list the averages on the source tasks only in Table 21, which shows that SD-CQL achieves comparable or even superior performance on most task sets. These results further substantiate its generalization ability.

Table 20: Win rates of Multi-to-Multi task sets. The reported results are the average performance over unseen tasks, and are averaged over 5 random seeds. Bold numbers indicate the best performance, and ± denotes one standard deviation.

| Task Set | BC-t | BC-r | CQL | ODIS | HiSSD | SD-CQL (Ours) |
|---|---|---|---|---|---|---|
| marine-easy-e | **99.29** ± 0.46 | 98.48 ± 0.92 | 18.84 ± 7.54 | 64.38 ± 34.86 | 97.41 ± 1.86 | 97.86 ± 1.07 |
| marine-easy-m | 71.7 ± 6.07 | 72.5 ± 4.17 | 8.48 ± 4.12 | 67.77 ± 5.32 | 70.54 ± 3.43 | **73.66** ± 4.35 |
| marine-easy-mr | 48.04 ± 10.78 | 40.98 ± 10.39 | 22.59 ± 13.56 | 6.34 ± 9.86 | 70.27 ± 3.82 | **74.02** ± 5.31 |
| marine-easy-me | 71.88 ± 8.0 | 70.45 ± 2.73 | 24.11 ± 4.87 | 63.75 ± 11.41 | 74.29 ± 4.16 | **84.02** ± 7.24 |
| marine-hard-e | 57.85 ± 3.91 | 54.93 ± 2.3 | 22.08 ± 2.11 | 14.44 ± 12.06 | 60.0 ± 5.73 | **65.97** ± 5.9 |
| marine-hard-m | 38.06 ± 3.57 | 38.61 ± 3.51 | 2.99 ± 2.11 | 28.06 ± 4.23 | 44.65 ± 3.39 | **45.9** ± 6.99 |
| marine-hard-mr | 45.35 ± 2.32 | 46.53 ± 3.02 | 17.64 ± 3.97 | 38.75 ± 6.23 | 45.14 ± 3.59 | **50.62** ± 3.59 |
| marine-hard-me | 42.71 ± 6.81 | 49.24 ± 5.97 | 15.35 ± 2.9 | 16.53 ± 17.28 | 52.22 ± 1.02 | **56.18** ± 4.48 |
| stalker-zealot-e | 56.38 ± 3.97 | 62.12 ± 4.39 | 10.0 ± 5.36 | 49.81 ± 7.58 | 60.25 ± 9.57 | **74.19** ± 4.52 |
| stalker-zealot-m | 17.06 ± 2.74 | 20.81 ± 3.53 | 11.31 ± 4.72 | 9.94 ± 8.09 | 20.31 ± 4.74 | **40.94** ± 6.74 |
| stalker-zealot-mr | 18.19 ± 5.49 | 11.5 ± 1.43 | 11.37 ± 2.21 | 7.56 ± 5.85 | 16.31 ± 5.6 | **24.56** ± 5.06 |
| stalker-zealot-me | 32.06 ± 5.98 | 31.06 ± 6.38 | 12.5 ± 3.1 | 26.06 ± 12.29 | 34.81 ± 3.39 | **63.44** ± 7.5 |

*-e*, *-m*, *-mr*, and *-mr* represent *-expert*, *-medium*, *-medium-replay*, and *-medium-expert*, respectively.

Table 21: Win rates of Multi-to-Multi task sets. The reported results are the average performance over source tasks, and are averaged over 5 random seeds. Bold numbers indicate the best performance, and ± denotes one standard deviation.

| Task Set | BC-t | BC-r | CQL | ODIS | HiSSD | SD-CQL (Ours) |
|---|---|---|---|---|---|---|
| marine-easy-e | 99.58 ± 0.51 | **99.79** ± 0.42 | 59.79 ± 3.40 | 84.79 ± 19.98 | 99.17 ± 1.21 | 98.54 ± 1.06 |
| marine-easy-m | 71.67 ± 2.41 | 70.21 ± 6.64 | 38.12 ± 13.89 | 70.21 ± 7.29 | 73.54 ± 2.76 | **80.83** ± 5.57 |
| marine-easy-mr | 53.33 ± 9.58 | 51.46 ± 4.69 | 45.83 ± 3.49 | 22.08 ± 11.21 | 78.96 ± 5.25 | **80.00** ± 11.30 |
| marine-easy-me | 83.12 ± 6.76 | 82.71 ± 10.43 | 55.62 ± 9.52 | 71.25 ± 12.27 | 87.71 ± 12.38 | **93.54** ± 4.29 |
| marine-hard-e | 85.83 ± 5.80 | 87.08 ± 3.06 | 33.12 ± 0.78 | 41.04 ± 25.36 | **92.08** ± 1.56 | 84.79 ± 3.13 |
| marine-hard-m | 51.88 ± 6.57 | 52.50 ± 3.52 | 27.71 ± 2.68 | 46.04 ± 4.86 | **56.46** ± 0.78 | 54.79 ± 3.33 |
| marine-hard-mr | 51.88 ± 5.72 | 47.29 ± 6.10 | 33.75 ± 4.40 | 32.92 ± 4.96 | **54.79** ± 6.51 | 47.50 ± 5.21 |
| marine-hard-me | **68.96** ± 9.89 | 60.00 ± 15.29 | 35.21 ± 1.67 | 28.75 ± 20.88 | 56.46 ± 6.50 | 56.25 ± 3.09 |
| stalker-zealot-e | 87.29 ± 5.03 | **90.21** ± 2.84 | 12.71 ± 7.89 | 72.50 ± 14.86 | 87.71 ± 2.02 | 80.83 ± 4.69 |
| stalker-zealot-m | 33.33 ± 8.62 | 34.38 ± 5.74 | 28.54 ± 7.78 | 18.75 ± 6.81 | 34.17 ± 4.90 | **38.75** ± 9.80 |
| stalker-zealot-mr | 15.83 ± 1.38 | 15.83 ± 3.32 | **25.21** ± 6.70 | 10.21 ± 9.02 | 10.62 ± 3.39 | 18.12 ± 3.52 |
| stalker-zealot-me | 64.79 ± 8.77 | 61.88 ± 13.35 | 16.46 ± 3.39 | 46.25 ± 26.19 | 58.54 ± 10.06 | **78.75** ± 7.81 |

*-e*, *-m*, *-mr*, and *-mr* represent *-expert*, *-medium*, *-medium-replay*, and *-medium-expert*, respectively.

### I.1.2 TASK-WISE RESULTS

We present the detailed results of our evaluation experiments on the Multi-to-Multi task sets. The results for *Marine-Easy* are presented in Table 22, the results for *Marine-Hard* are shown in Table 23, and the results for *Stalker-Zealot* are provided in Table 24. In each table, we present the multi-task evaluation results on datasets of varying quality, with the source tasks used for training marked by "⋄". The results show that, despite being trained on a limited number of source tasks, SD-CQL exhibits strong multi-task generalization, as reflected in its exceptional average performance across all test tasks.

Table 22: Win rates of *Marine-Easy* scenario. The reported results are averaged over 5 random seeds. The bold number denotes the best performance, and ± denotes one standard deviation.

| | Task | BC-t | BC-r | CQL | ODIS | HiSSD | SD-CQL (Ours) |
|---|---|---|---|---|---|---|---|
| **Expert** | 3m ⋄ | 99.38 ± 1.40 | **100.00** ± 0.00 | 98.13 ± 2.80 | 97.50 ± 1.40 | 98.75 ± 2.80 | 98.75 ± 2.80 |
| | 4m | 90.62 ± 7.33 | 96.88 ± 3.83 | 80.63 ± 13.69 | 56.25 ± 37.69 | 92.5 ± 8.73 | **97.50** ± 2.61 |
| | 5m ⋄ | **100.00** ± 0.00 | **100.00** ± 0.00 | 81.25 ± 11.48 | 78.12 ± 31.48 | **100.00** ± 0.00 | 98.75 ± 1.71 |
| | 6m | **100.00** ± 0.00 | **100.00** ± 0.00 | 40.00 ± 30.89 | 55.62 ± 43.49 | 98.75 ± 1.71 | 96.88 ± 4.42 |
| | 7m | **100.00** ± 0.00 | 98.75 ± 1.71 | 10.63 ± 23.76 | 58.12 ± 44.28 | 98.75 ± 2.80 | 97.50 ± 4.07 |
| | 8m | 99.38 ± 1.40 | **100.00** ± 0.00 | 0.63 ± 1.40 | 63.75 ± 41.14 | **100.00** ± 0.00 | 98.12 ± 1.71 |
| | 9m | **100.00** ± 0.00 | **100.00** ± 0.00 | 0.00 ± 0.00 | 70.62 ± 42.65 | 99.38 ± 1.40 | **100.00** ± 0.00 |
| | 10m ⋄ | **100.00** ± 0.00 | 98.75 ± 1.71 | 0.00 ± 0.00 | 78.75 ± 35.24 | 98.75 ± 1.71 | 98.12 ± 2.80 |
| | 11m | 99.38 ± 1.40 | **100.00** ± 0.00 | 0.00 ± 0.00 | 80.00 ± 28.78 | 98.75 ± 1.71 | 98.12 ± 4.19 |
| | 12m | **100.00** ± 0.00 | 99.38 ± 1.40 | 0.00 ± 0.00 | 66.25 ± 42.01 | 93.75 ± 9.11 | 96.88 ± 2.21 |
| | **Average** | 98.87 ± 0.58 | **99.38** ± 0.40 | 31.13 ± 6.36 | 70.50 ± 30.14 | 97.94 ± 1.23 | 98.06 ± 1.00 |
| **Medium** | 3m⋄ | 61.25 ± 10.03 | 65.62 ± 8.84 | **70.63** ± 17.48 | 64.38 ± 10.96 | 61.88 ± 5.59 | 70.62 ± 12.62 |
| | 4m | **66.25** ± 8.09 | 64.38 ± 34.42 | 27.50 ± 30.09 | 55.00 ± 30.98 | 71.88 ± 10.13 | 52.50 ± 33.18 |
| | 5m ⋄ | 78.75 ± 8.09 | 80.00 ± 4.19 | 43.75 ± 34.59 | 76.25 ± 7.19 | 83.12 ± 9.78 | **84.38** ± 4.94 |
| | 6m | 90.00 ± 6.01 | 84.38 ± 5.85 | 24.38 ± 23.43 | **93.12** ± 8.39 | 76.88 ± 12.02 | 89.38 ± 9.27 |
| | 7m | 99.38 ± 1.40 | 90.62 ± 11.69 | 7.50 ± 9.00 | 90.62 ± 12.50 | 95.00 ± 6.48 | **100.00** ± 0.00 |
| | 8m | 86.88 ± 10.69 | **95.62** ± 6.48 | 0.00 ± 0.00 | 89.38 ± 5.23 | 93.12 ± 4.07 | 90.00 ± 6.40 |
| | 9m | **83.12** ± 8.73 | 78.75 ± 5.13 | 0.00 ± 0.00 | 72.50 ± 18.67 | 76.88 ± 7.84 | 82.50 ± 13.00 |
| | 10m ⋄ | 70.62 ± 17.34 | 69.38 ± 6.77 | 0.00 ± 0.00 | 70.00 ± 15.08 | 75.62 ± 6.01 | **87.50** ± 6.25 |
| | 11m | 43.75 ± 13.07 | 43.75 ± 5.85 | 0.00 ± 0.00 | 43.12 ± 9.73 | 46.25 ± 12.18 | **55.00** ± 10.27 |
| | 12m | 38.12 ± 7.78 | 44.38 ± 5.59 | 0.00 ± 0.00 | 30.62 ± 12.38 | 33.75 ± 5.59 | **46.25** ± 10.69 |
| | **Average** | 71.81 ± 3.74 | 71.69 ± 4.33 | 17.38 ± 7.31 | 68.50 ± 5.74 | 71.44 ± 2.77 | **75.81** ± 1.73 |
| **Medium-Replay** | 3m ⋄ | 80.00 ± 6.48 | 70.62 ± 22.81 | 71.88 ± 11.05 | 60.62 ± 35.05 | **82.50** ± 9.53 | 78.75 ± 16.45 |
| | 4m | 75.00 ± 15.15 | 70.00 ± 9.53 | 74.38 ± 9.48 | 18.75 ± 25.77 | 69.38 ± 21.13 | **79.38** ± 7.53 |
| | 5m ⋄ | 77.50 ± 31.51 | 88.75 ± 15.56 | 62.50 ± 14.82 | 73.75 ± 26.01 | 86.25 ± 7.19 | **91.25** ± 9.73 |
| | 6m | 88.75 ± 23.45 | 98.12 ± 2.80 | 40.63 ± 41.93 | 12.50 ± 27.95 | **100.00** ± 0.00 | 99.38 ± 1.40 |
| | 7m | 86.88 ± 24.15 | 90.62 ± 12.88 | 26.25 ± 36.01 | 8.12 ± 18.17 | **100.00** ± 0.00 | **100.00** ± 0.00 |
| | 8m | 19.38 ± 29.43 | 43.75 ± 35.84 | 9.38 ± 20.96 | 0.00 ± 0.00 | **96.25** ± 5.13 | 89.38 ± 10.27 |
| | 9m | 15.62 ± 18.88 | 25.00 ± 24.51 | 3.75 ± 8.39 | 5.00 ± 11.18 | 76.25 ± 4.74 | **88.75** ± 4.19 |
| | 10m ⋄ | **80.62** ± 12.18 | 76.88 ± 17.62 | 3.13 ± 6.99 | 76.88 ± 11.82 | 68.12 ± 10.92 | 70.00 ± 39.26 |
| | 11m | 1.25 ± 2.80 | 5.00 ± 8.15 | 3.13 ± 6.99 | 0.00 ± 0.00 | 24.38 ± 13.15 | **35.62** ± 19.96 |
| | 12m | 0.00 ± 0.00 | 3.75 ± 6.77 | 0.63 ± 1.40 | 0.00 ± 0.00 | 25.62 ± 23.74 | **25.62** ± 15.05 |
| | **Average** | 44.12 ± 8.19 | 49.62 ± 9.17 | 29.56 ± 11.73 | 11.06 ± 8.94 | 72.88 ± 3.16 | **75.81** ± 6.83 |
| **Medium-Expert** | 3m ⋄ | 90.00 ± 18.93 | 83.75 ± 23.94 | **96.88** ± 3.13 | 63.12 ± 27.90 | 90.62 ± 20.96 | **96.88** ± 3.12 |
| | 4m | **87.50** ± 9.11 | 78.12 ± 14.82 | 55.63 ± 24.55 | 49.38 ± 37.72 | 76.25 ± 12.62 | 72.50 ± 19.19 |
| | 5m ⋄ | 69.38 ± 9.22 | 77.50 ± 10.69 | 70.00 ± 29.03 | 5.00 ± 11.18 | 88.12 ± 14.89 | **96.25** ± 6.77 |
| | 6m | 63.75 ± 25.92 | 68.75 ± 20.73 | 80.00 ± 15.08 | 50.00 ± 9.88 | 55.00 ± 15.08 | **93.75** ± 6.25 |
| | 7m | 85.62 ± 6.85 | 86.25 ± 16.33 | 31.25 ± 17.26 | 65.62 ± 22.53 | 86.25 ± 7.84 | **90.62** ± 11.05 |
| | 8m | 56.88 ± 11.14 | 66.25 ± 24.65 | 1.88 ± 4.19 | 66.25 ± 23.84 | 76.25 ± 8.15 | **90.00** ± 9.73 |
| | 9m | 70.62 ± 12.62 | 75.00 ± 22.43 | 0.00 ± 0.00 | 73.12 ± 18.43 | **90.62** ± 5.85 | 87.50 ± 10.60 |
| | 10m ⋄ | 5.00 ± 6.85 | 11.88 ± 12.96 | 0.00 ± 0.00 | 0.62 ± 1.40 | 84.38 ± 16.83 | **87.50** ± 11.90 |
| | 11m | 68.12 ± 14.22 | 71.88 ± 8.56 | 0.00 ± 0.00 | 76.88 ± 18.83 | 75.00 ± 19.52 | **81.88** ± 17.73 |
| | 12m | 60.62 ± 12.81 | 56.88 ± 4.64 | 0.00 ± 0.00 | 65.00 ± 16.00 | 60.62 ± 12.81 | **71.88** ± 18.09 |
| | **Average** | 74.12 ± 3.21 | 75.25 ± 7.41 | 33.56 ± 6.47 | 66.00 ± 9.21 | 78.31 ± 4.99 | **86.88** ± 5.68 |

⋄ denotes the source tasks.

Table 23: Win rates of *Marine-Hard* scenario. The reported results are averaged over 5 random seeds. The bold number denotes the best performance, and ± denotes one standard deviation.

| | Task | BC-t | BC-r | CQL | ODIS | HiSSD | SD-CQL (Ours) |
|---|---|---|---|---|---|---|---|
| **Expert** | 3m ◇ | **100.00** ± 0.00 | **100.00** ± 0.00 | 98.12 ± 2.8 | 74.38 ± 34.47 | 98.12 ± 1.71 | 99.38 ± 1.40 |
| | 4m | 92.50 ± 8.44 | 95.00 ± 7.19 | 92.50 ± 7.19 | 37.50 ± 32.40 | 98.75 ± 1.71 | **99.38** ± 1.40 |
| | 5m | 97.50 ± 2.61 | 83.75 ± 12.38 | 96.25 ± 5.13 | 30.62 ± 32.73 | **100.0** ± 0.00 | 99.38 ± 1.40 |
| | 10m | 95.62 ± 3.56 | 91.25 ± 13.15 | 7.50 ± 13.55 | 18.12 ± 12.96 | **99.38** ± 1.40 | 95.00 ± 9.53 |
| | 12m | 68.75 ± 31.25 | **88.75** ± 10.73 | 1.88 ± 4.19 | 5.00 ± 11.18 | 53.75 ± 50.7 | 74.38 ± 38.04 |
| | 5m6m ◇ | 62.50 ± 8.84 | 67.50 ± 8.44 | 0.62 ± 1.40 | 25.00 ± 29.06 | **79.38** ± 6.48 | 60.00 ± 9.73 |
| | 7m8m | 30.62 ± 17.73 | 23.12 ± 19.84 | 0.62 ± 1.40 | 8.75 ± 11.14 | **37.50** ± 9.38 | **37.50** ± 16.24 |
| | 8m9m | 41.88 ± 6.85 | 43.75 ± 19.01 | 0.00 ± 0.00 | 15.00 ± 15.84 | 53.12 ± 16.09 | **87.50** ± 14.32 |
| | 9m10m ◇ | **98.75** ± 2.80 | 90.00 ± 12.38 | 0.62 ± 1.40 | 23.75 ± 36.55 | **98.75** ± 1.71 | 95.00 ± 1.71 |
| | 10m11m | 63.75 ± 7.84 | **86.25** ± 8.44 | 0.00 ± 0.00 | 12.50 ± 14.66 | 83.75 ± 16.89 | 75.62 ± 9.73 |
| | 10m12m | 3.75 ± 2.61 | 8.12 ± 5.68 | 0.00 ± 0.00 | 1.88 ± 4.19 | 10.00 ± 4.64 | **23.12** ± 16.77 |
| | 13m15m | 0.00 ± 0.00 | 0.62 ± 1.40 | 0.00 ± 0.00 | 0.62 ± 1.40 | **3.75** ± 5.13 | 1.88 ± 2.80 |
| | **Average** | 62.97 ± 2.18 | 64.84 ± 3.24 | 24.84 ± 1.53 | 21.09 ± 14.72 | 68.02 ± 4.47 | **70.68** ± 4.80 |
| **Medium** | 3m ◇ | 62.50 ± 9.38 | 52.50 ± 15.21 | **81.25** ± 6.63 | 59.38 ± 7.97 | 64.38 ± 5.23 | 75.62 ± 8.95 |
| | 4m | 41.25 ± 23.32 | 41.88 ± 25.92 | 8.12 ± 4.74 | 58.12 ± 24.37 | **76.88** ± 7.53 | 71.88 ± 7.97 |
| | 5m | 86.88 ± 6.01 | 85.62 ± 11.40 | 18.75 ± 17.26 | 68.12 ± 30.49 | 90.62 ± 12.69 | **96.25** ± 5.59 |
| | 10m | 90.00 ± 12.77 | 90.00 ± 12.38 | 0.00 ± 0.00 | 70.00 ± 32.67 | 93.75 ± 3.12 | **96.25** ± 8.39 |
| | 12m | 75.00 ± 21.31 | 64.38 ± 30.51 | 1.88 ± 2.80 | 18.12 ± 23.74 | **79.38** ± 9.53 | 72.50 ± 21.81 |
| | 5m6m ◇ | 30.62 ± 2.61 | **38.75** ± 6.48 | 0.00 ± 0.00 | 10.00 ± 11.98 | 30.63 ± 11.35 | 30.00 ± 10.50 |
| | 7m8m | 3.12 ± 3.12 | **10.00** ± 8.67 | 0.00 ± 0.00 | 4.38 ± 3.56 | 8.75 ± 10.92 | **10.00** ± 13.51 |
| | 8m9m | 6.88 ± 7.78 | 6.25 ± 7.65 | 0.00 ± 0.00 | 11.25 ± 13.18 | 13.12 ± 6.01 | **15.00** ± 18.01 |
| | 9m10m ◇ | 64.38 ± 14.76 | 64.38 ± 12.62 | 0.00 ± 0.00 | 68.75 ± 15.93 | **74.38** ± 10.22 | 58.75 ± 16.15 |
| | 10m11m | 41.88 ± 11.18 | 43.12 ± 16.30 | 0.00 ± 0.00 | 21.25 ± 9.73 | 38.12 ± 20.66 | **50.62** ± 23.11 |
| | 10m12m | 0.62 ± 1.40 | 0.62 ± 1.40 | 0.00 ± 0.00 | 0.00 ± 0.00 | **1.25** ± 2.80 | 0.00 ± 0.00 |
| | 13m15m | 1.88 ± 1.71 | **0.62** ± 1.40 | 0.00 ± 0.00 | 1.25 ± 2.80 | 0.00 ± 0.00 | **0.62** ± 1.40 |
| | **Average** | 42.08 ± 2.63 | 41.51 ± 3.53 | 9.17 ± 1.78 | 32.55 ± 4.31 | 47.60 ± 2.66 | **48.12** ± 5.66 |
| **Medium-Replay** | 3m ◇ | 75.62 ± 7.13 | 80.00 ± 17.06 | 81.88 ± 6.77 | 75.00 ± 13.62 | 84.38 ± 8.56 | **85.00** ± 10.22 |
| | 4m | 83.12 ± 10.03 | 83.12 ± 8.44 | 57.5 ± 25.64 | 55.62 ± 34.40 | 75.00 ± 8.27 | **91.88** ± 9.78 |
| | 5m | 95.62 ± 5.23 | 95.62 ± 9.78 | 89.38 ± 11.61 | 93.12 ± 8.39 | 83.75 ± 13.51 | **100.00** ± 0.00 |
| | 10m | 87.50 ± 11.27 | 90.62 ± 11.05 | 3.12 ± 6.99 | 89.38 ± 8.44 | **93.12** ± 10.22 | 86.88 ± 7.78 |
| | 12m | 85.62 ± 10.96 | 88.12 ± 4.07 | 1.25 ± 2.8 | 70.62 ± 16.33 | 88.12 ± 10.22 | **94.38** ± 4.07 |
| | 5m6m ◇ | 30.00 ± 13.55 | **33.12** ± 6.48 | 18.75 ± 9.63 | 9.38 ± 6.25 | 29.38 ± 13.37 | 13.75 ± 5.23 |
| | 7m8m | **14.38** ± 2.80 | 3.75 ± 2.61 | 6.25 ± 4.94 | 5.00 ± 7.84 | 9.38 ± 7.65 | 8.12 ± 7.19 |
| | 8m9m | 8.12 ± 3.56 | 11.88 ± 5.59 | 1.25 ± 1.71 | 1.88 ± 1.71 | 12.50 ± 4.94 | **26.88** ± 10.96 |
| | 9m10m ◇ | 36.25 ± 14.08 | 42.50 ± 14.76 | 0.62 ± 1.40 | 14.38 ± 13.18 | **50.62** ± 13.51 | 43.75 ± 12.3 |
| | 10m11m | 36.25 ± 8.15 | 29.38 ± 18.57 | 0.00 ± 0.00 | 32.50 ± 22.27 | **40.00** ± 13.69 | 38.12 ± 17.17 |
| | 10m12m | **2.50** ± 2.61 | 0.62 ± 1.40 | 0.00 ± 0.00 | 0.00 ± 0.00 | 0.62 ± 1.40 | 1.25 ± 2.80 |
| | 13m15m | 5.62 ± 2.61 | 5.00 ± 6.48 | 0.00 ± 0.00 | 0.62 ± 1.40 | 3.75 ± 4.07 | **8.12** ± 2.80 |
| | **Average** | 46.72 ± 2.35 | 46.98 ± 1.95 | 21.67 ± 2.83 | 37.29 ± 5.48 | 47.55 ± 3.56 | **49.84** ± 3.36 |
| **Medium-Expert** | 3m ◇ | 74.38 ± 14.72 | 85.00 ± 19.44 | **100.00** ± 0.00 | 53.75 ± 36.94 | 90.62 ± 19.26 | 98.75 ± 1.71 |
| | 4m | **95.00** ± 3.56 | 85.00 ± 23.11 | 57.5 ± 29.86 | 34.38 ± 45.93 | 94.38 ± 4.07 | 92.50 ± 2.80 |
| | 5m | 91.88 ± 9.78 | 58.12 ± 28.00 | 80.62 ± 13.15 | 31.25 ± 38.84 | **98.75** ± 2.80 | 93.12 ± 10.92 |
| | 10m | 93.75 ± 10.83 | 80.00 ± 19.71 | 0.00 ± 0.00 | 26.25 ± 34.13 | 95.62 ± 1.71 | **95.00** ± 5.68 |
| | 12m | 59.38 ± 43.24 | 73.12 ± 26.85 | 0.00 ± 0.00 | 3.75 ± 5.59 | 86.25 ± 13.73 | **90.00** ± 6.77 |
| | 5m6m ◇ | **43.12** ± 22.14 | 37.50 ± 18.62 | 5.62 ± 5.59 | 8.75 ± 8.09 | 42.50 ± 23.03 | 29.38 ± 17.06 |
| | 7m8m | 18.12 ± 4.64 | 16.25 ± 14.72 | 0.00 ± 0.00 | 4.38 ± 6.85 | **21.25** ± 8.67 | 19.38 ± 21.01 |
| | 8m9m | 31.25 ± 19.39 | 26.25 ± 18.96 | 0.00 ± 0.00 | 15.00 ± 21.58 | 28.12 ± 11.05 | **39.38** ± 26.57 |
| | 9m10m ◇ | 62.50 ± 24.90 | **84.38** ± 9.38 | 0.00 ± 0.00 | 23.75 ± 35.81 | 36.25 ± 20.20 | 40.62 ± 21.42 |
| | 10m11m | 51.88 ± 17.34 | 41.88 ± 19.21 | 0.00 ± 0.00 | 33.75 ± 38.87 | 42.50 ± 4.74 | **61.88** ± 20.18 |
| | 10m12m | 1.88 ± 4.19 | 2.50 ± 4.07 | 0.00 ± 0.00 | 0.00 ± 0.00 | 2.50 ± 5.59 | **4.38** ± 9.78 |
| | 13m15m | 0.00 ± 0.00 | 1.25 ± 1.71 | 0.00 ± 0.00 | 0.00 ± 0.00 | 0.62 ± 1.40 | **10.00** ± 10.69 |
| | **Average** | 51.93 ± 7.21 | 49.27 ± 5.16 | 20.31 ± 1.97 | 19.58 ± 17.63 | 53.28 ± 1.88 | **56.20** ± 3.21 |

◇ denotes the source tasks.

"$X$m$Y$m" represents "$X$m_vs_$Y$m".

Table 24: Win rates of *Stalker-Zealot* scenario. The reported results are averaged over 5 random seeds. The bold number denotes the best performance, and ± denotes one standard deviation.

| | Task | BC-t | BC-r | CQL | ODIS | HiSSD | SD-CQL (Ours) |
|---|---|---|---|---|---|---|---|
| Expert | 1s3z | 51.88 ± 28.26 | 65.00 ± 7.46 | 70.00 ± 39.57 | 41.25 ± 38.55 | 65.62 ± 19.76 | **74.38** ± 31.59 |
| | 1s4z | 34.38 ± 10.83 | **55.00** ± 31.14 | 25.62 ± 20.30 | 19.38 ± 29.18 | 52.50 ± 37.33 | 53.75 ± 28.07 |
| | 1s5z | 24.38 ± 28.42 | 19.38 ± 25.81 | 1.25 ± 1.71 | 30.62 ± 40.41 | 18.75 ± 22.10 | **60.00** ± 24.35 |
| | 2s3z ◇ | **95.62** ± 2.80 | 95.00 ± 4.74 | 36.88 ± 25.33 | 77.50 ± 25.14 | 94.38 ± 7.78 | 85.62 ± 10.50 |
| | 2s4z ◇ | **81.25** ± 7.33 | 76.88 ± 8.44 | 1.25 ± 1.71 | 53.12 ± 24.61 | 73.75 ± 5.68 | 71.88 ± 13.80 |
| | 2s5z | **66.25** ± 23.43 | 65.00 ± 18.54 | 0.00 ± 0.00 | 48.12 ± 20.44 | 61.88 ± 17.73 | 63.75 ± 10.50 |
| | 3s3z | **91.25** ± 5.59 | 76.88 ± 19.47 | 3.12 ± 5.41 | 81.88 ± 15.53 | 72.50 ± 23.53 | 90.00 ± 8.39 |
| | 3s4z | 93.12 ± 4.64 | 78.75 ± 6.01 | 0.00 ± 0.00 | 81.88 ± 8.95 | 93.12 ± 4.07 | **95.62** ± 5.23 |
| | 3s5z ◇ | 93.75 ± 5.85 | 90.00 ± 8.39 | 0.00 ± 0.00 | 86.88 ± 5.13 | **95.00** ± 4.74 | 85.00 ± 11.14 |
| | 4s3z | 77.50 ± 17.31 | 78.12 ± 16.97 | 0.00 ± 0.00 | 71.25 ± 19.06 | 83.75 ± 15.05 | **90.00** ± 9.48 |
| | 4s4z | 66.25 ± 16.89 | 52.50 ± 8.39 | 0.00 ± 0.00 | 52.50 ± 20.89 | 60.62 ± 34.28 | **74.38** ± 13.51 |
| | 4s5z | 54.38 ± 12.81 | 38.75 ± 3.56 | 0.00 ± 0.00 | 35.00 ± 27.99 | 52.50 ± 22.90 | **71.88** ± 17.95 |
| | 4s6z | 61.88 ± 10.92 | 34.38 ± 13.62 | 0.00 ± 0.00 | 36.25 ± 21.83 | 41.25 ± 21.36 | **68.12** ± 14.89 |
| | **Average** | 68.61 ± 3.93 | 63.51 ± 2.64 | 10.62 ± 5.88 | 55.05 ± 9.11 | 66.59 ± 7.77 | **75.72** ± 3.73 |
| Medium | 1s3z | 3.12 ± 3.83 | 14.38 ± 25.73 | 36.25 ± 25.25 | 10.62 ± 22.05 | 35.0 ± 22.69 | **64.38** ± 29.70 |
| | 1s4z | 18.75 ± 25.86 | 11.88 ± 24.84 | 43.75 ± 29.89 | 7.50 ± 8.15 | 13.75 ± 5.23 | **48.75** ± 37.80 |
| | 1s5z | 5.62 ± 6.77 | 3.75 ± 5.13 | 11.88 ± 13.33 | 0.62 ± 1.40 | 9.38 ± 6.25 | **51.25** ± 38.76 |
| | 2s3z ◇ | 49.38 ± 13.51 | 48.12 ± 4.74 | **62.5** ± 14.32 | 38.12 ± 11.57 | 50.62 ± 10.22 | 30.63 ± 23.94 |
| | 2s4z ◇ | 11.88 ± 8.09 | 8.75 ± 13.15 | 23.12 ± 16.03 | 8.75 ± 7.78 | 9.38 ± 7.65 | **29.38** ± 16.92 |
| | 2s5z | 13.75 ± 10.50 | 16.25 ± 8.67 | 1.25 ± 2.8 | 18.75 ± 27.15 | 14.37 ± 6.48 | **42.50** ± 23.24 |
| | 3s3z | 39.38 ± 15.56 | 26.25 ± 15.56 | 16.25 ± 22.25 | 21.88 ± 20.01 | 31.87 ± 16.30 | **45.00** ± 12.62 |
| | 3s4z | 46.25 ± 17.59 | 23.75 ± 12.22 | 3.75 ± 8.39 | 18.12 ± 17.59 | 45.62 ± 14.59 | **64.38** ± 28.35 |
| | 3s5z ◇ | 41.88 ± 9.78 | 43.12 ± 18.14 | 0.00 ± 0.00 | 9.38 ± 9.38 | 42.50 ± 16.62 | **56.25** ± 9.11 |
| | 4s3z | 37.50 ± 26.42 | 42.50 ± 18.57 | 0.00 ± 0.00 | 11.88 ± 8.67 | 33.12 ± 13.91 | **48.12** ± 16.92 |
| | 4s4z | **23.75** ± 5.23 | 16.25 ± 14.56 | 0.00 ± 0.00 | 8.12 ± 10.03 | 7.50 ± 3.56 | 18.12 ± 10.69 |
| | 4s5z | **10.62** ± 5.68 | 10.00 ± 9.73 | 0.00 ± 0.00 | 0.62 ± 1.40 | 4.38 ± 4.19 | **10.62** ± 7.84 |
| | 4s6z | 9.38 ± 4.94 | 5.62 ± 4.64 | 0.00 ± 0.00 | 1.25 ± 2.80 | 8.12 ± 4.74 | **16.25** ± 11.35 |
| | **Average** | 23.94 ± 2.72 | 20.82 ± 3.78 | 15.29 ± 4.89 | 11.97 ± 7.53 | 23.51 ± 3.76 | **40.43** ± 7.06 |
| Medium-Replay | 1s3z | 17.50 ± 11.40 | 33.12 ± 20.32 | 18.75 ± 14.66 | 3.12 ± 5.41 | **55.62** ± 26.46 | 21.25 ± 14.39 |
| | 1s4z | 7.50 ± 6.09 | 13.75 ± 13.00 | 32.5 ± 16.18 | 10.62 ± 10.73 | **15.00** ± 11.98 | 13.75 ± 24.06 |
| | 1s5z | 3.75 ± 4.07 | 3.75 ± 8.39 | **10.00** ± 9.48 | 5.00 ± 7.19 | **10.00** ± 13.87 | 8.75 ± 8.39 |
| | 2s3z ◇ | 12.50 ± 6.63 | 6.88 ± 5.59 | **53.75** ± 13.51 | 12.50 ± 16.39 | 12.50 ± 6.99 | 7.50 ± 2.80 |
| | 2s4z ◇ | 8.12 ± 9.53 | 10.62 ± 5.68 | **18.12** ± 11.98 | 12.50 ± 13.44 | 5.62 ± 4.07 | 6.25 ± 4.42 |
| | 2s5z | 11.25 ± 9.27 | **22.50** ± 24.25 | 9.38 ± 8.27 | 10.62 ± 14.59 | 9.38 ± 6.25 | 17.50 ± 15.08 |
| | 3s3z | 13.75 ± 12.62 | 12.50 ± 4.94 | 22.50 ± 7.13 | 12.50 ± 13.44 | 18.75 ± 24.11 | **57.50** ± 15.87 |
| | 3s4z | 27.50 ± 17.46 | 45.00 ± 19.09 | 13.12 ± 11.57 | 8.12 ± 9.00 | 20.62 ± 10.50 | **47.50** ± 32.43 |
| | 3s5z ◇ | 26.88 ± 13.91 | 30.00 ± 11.18 | 3.75 ± 5.13 | 5.62 ± 3.42 | 13.75 ± 7.84 | **40.62** ± 8.27 |
| | 4s3z | 11.25 ± 18.57 | **23.75** ± 19.59 | 4.38 ± 5.23 | 13.75 ± 16.18 | 15.00 ± 18.67 | **23.75** ± 21.49 |
| | 4s4z | 12.50 ± 9.11 | 18.75 ± 15.15 | 2.50 ± 3.42 | 3.75 ± 3.42 | 11.88 ± 8.09 | **28.12** ± 15.93 |
| | 4s5z | 6.25 ± 3.83 | 5.00 ± 6.48 | 0.00 ± 0.00 | 5.00 ± 8.15 | 1.88 ± 2.80 | **16.88** ± 9.00 |
| | 4s6z | 3.75 ± 5.13 | 3.75 ± 4.07 | 0.62 ± 1.40 | 3.12 ± 3.83 | 5.00 ± 1.71 | **10.62** ± 6.09 |
| | **Average** | 12.50 ± 1.75 | 17.64 ± 4.40 | 14.57 ± 2.02 | 8.17 ± 6.08 | 15.00 ± 4.02 | **23.08** ± 3.38 |
| Medium-Expert | 1s3z | 47.50 ± 39.12 | 25.00 ± 30.22 | **80.00** ± 20.20 | 49.38 ± 36.74 | 58.75 ± 35.93 | 73.75 ± 40.42 |
| | 1s4z | 5.00 ± 3.56 | 13.12 ± 9.73 | 43.12 ± 18.67 | 6.88 ± 15.37 | 10.00 ± 6.01 | **72.50** ± 27.72 |
| | 1s5z | 11.88 ± 23.22 | 10.62 ± 20.32 | 1.88 ± 1.71 | 0.62 ± 1.40 | 6.25 ± 12.30 | **72.50** ± 17.73 |
| | 2s3z ◇ | 80.62 ± 21.47 | 71.25 ± 10.22 | 49.38 ± 11.35 | 59.38 ± 35.63 | 68.75 ± 14.99 | **86.25** ± 10.50 |
| | 2s4z ◇ | 37.50 ± 32.10 | 61.25 ± 15.87 | 0.00 ± 0.00 | 47.50 ± 31.44 | 56.25 ± 24.71 | **62.50** ± 19.64 |
| | 2s5z | 25.62 ± 19.94 | 21.88 ± 11.48 | 0.00 ± 0.00 | 27.50 ± 18.14 | 23.12 ± 9.53 | **40.62** ± 13.80 |
| | 3s3z | 47.50 ± 37.85 | 56.88 ± 11.57 | 0.00 ± 0.00 | 55.00 ± 29.20 | 63.75 ± 29.37 | **71.25** ± 16.00 |
| | 3s4z | 60.00 ± 17.87 | 74.38 ± 12.77 | 0.00 ± 0.00 | 50.00 ± 30.78 | 72.50 ± 16.15 | **93.12** ± 3.42 |
| | 3s5z ◇ | 67.50 ± 17.76 | 61.88 ± 12.18 | 0.00 ± 0.00 | 31.88 ± 24.45 | 50.62 ± 18.01 | **87.50** ± 6.99 |
| | 4s3z | 62.50 ± 18.62 | **70.62** ± 7.84 | 0.00 ± 0.00 | 39.38 ± 27.12 | 55.00 ± 24.27 | 55.00 ± 28.86 |
| | 4s4z | 30.62 ± 5.59 | 35.00 ± 23.11 | 0.00 ± 0.00 | 19.38 ± 12.77 | 41.25 ± 21.36 | **61.88** ± 20.54 |
| | 4s5z | 10.00 ± 5.13 | 6.88 ± 10.22 | 0.00 ± 0.00 | 6.88 ± 2.61 | 7.50 ± 6.48 | **49.38** ± 26.28 |
| | 4s6z | 10.00 ± 7.46 | 6.25 ± 5.85 | 0.00 ± 0.00 | 5.62 ± 7.78 | 10.00 ± 5.13 | **44.38** ± 19.94 |
| | **Average** | 38.17 ± 5.38 | 39.62 ± 2.98 | 13.41 ± 2.53 | 30.72 ± 14.65 | 40.29 ± 4.64 | **66.97** ± 7.01 |

◇ denotes the source tasks.

## I.2 ONE-TO-MULTI TASK SETS

We present the detailed results of our evaluation experiments on the One-to-Multi task sets in Table 25. The source tasks used for training are marked by "◇". The results show that, despite being trained on only one source task, SD-CQL exhibits better multi-task generalization, as reflected in its exceptional average performance across all test tasks.

Table 25: Win rates of One-to-Multi task sets. The reported results are averaged over 5 random seeds. The bold number denotes the best performance, and ± denotes one standard deviation.

| Task Set | | BC-t | BC-r | CQL | ODIS | HiSSD | SD-CQL (Ours) |
|---|---|---|---|---|---|---|---|
| *Marine Single* | 3m ◇ | **100.00** ± 0.00 | 96.88 ± 4.42 | **100.00** ± 0.00 | 99.38 ± 1.4 | **100.00** ± 0.00 | **100.00** ± 0.00 |
| | 5m | 81.88 ± 9.22 | 67.5 ± 27.3 | 88.12 ± 7.46 | 37.5 ± 33.15 | 66.88 ± 32.52 | **91.25** ± 10.92 |
| | 8m | 38.75 ± 22.38 | 35.62 ± 29.37 | 40.00 ± 26.00 | 11.88 ± 21.47 | 31.87 ± 41.01 | **49.38** ± 23.63 |
| | 10m | 20.62 ± 22.38 | 16.25 ± 20.54 | 34.38 ± 25.39 | 11.25 ± 25.16 | 18.75 ± 40.20 | **41.88** ± 25.83 |
| | 12m | 9.38 ± 13.98 | 2.50 ± 5.59 | 17.50 ± 10.27 | 6.25 ± 13.98 | 3.12 ± 6.99 | **27.50** ± 21.13 |
| **Average** | | 50.12 ± 9.65 | 43.75 ± 14.42 | 56.00 ± 11.07 | 33.25 ± 16.50 | 44.12 ± 18.89 | **62.00** ± 13.80 |
| *Marine Single-Inv* | 10m ◇ | **100.00** ± 0.00 | 98.75 ± 2.80 | 0.62 ± 1.4 | 88.75 ± 12.22 | **100.00** ± 0.00 | **100.00** ± 0.00 |
| | 8m | 98.75 ± 1.71 | 83.12 ± 29.28 | 3.75 ± 8.39 | 90.62 ± 8.84 | **100.00** ± 0.00 | **100.00** ± 0.00 |
| | 5m | 3.75 ± 6.77 | 33.12 ± 33.63 | 6.25 ± 13.98 | 87.50 ± 8.84 | 91.88 ± 6.09 | **97.50** ± 2.61 |
| | 4m | 1.25 ± 2.80 | 20.62 ± 33.04 | 0.62 ± 1.4 | 62.50 ± 21.99 | 73.75 ± 15.72 | **75.62** ± 21.58 |
| | 3m | 0.00 ± 0.00 | 6.88 ± 13.69 | 2.5 ± 4.07 | 12.50 ± 18.22 | **68.75** ± 17.4 | 64.38 ± 15.4 |
| **Average** | | 40.75 ± 1.83 | 48.50 ± 17.09 | 2.75 ± 5.19 | 68.38 ± 11.47 | 86.88 ± 4.49 | **87.50** ± 6.54 |

◇ denotes the source task.

## J LEARNING CURVES

In Figures 7, we plot the learning curves of the average performance across different task sets for Multi-to-Multi. To make the figures clearer, we use abbreviations to represent the dataset quality. Specifically: -E, -M, -MR, and -ME stand for expert, medium, medium-replay, and medium-expert, respectively. We report results over 5 random seeds, where the solid line represents the mean and the shaded area represents one standard deviation. It is evident that SD-CQL sustains the highest average performance across most task sets.

In Figure 8, we plot the learning curves for the average performance (marked as "AVG") and task-specific performance (marked with the respective task name) for One-to-Multi task sets. The results are reported over 5 random seeds, with the solid line representing the mean and the shaded area indicating one standard deviation. It can be observed that even when trained on a single source task, SD-CQL exhibits the best multi-task generalization performance. Notably, in the *Marine-Single* task set, SD-CQL's peak performance on unseen tasks significantly surpasses that of the other baselines.

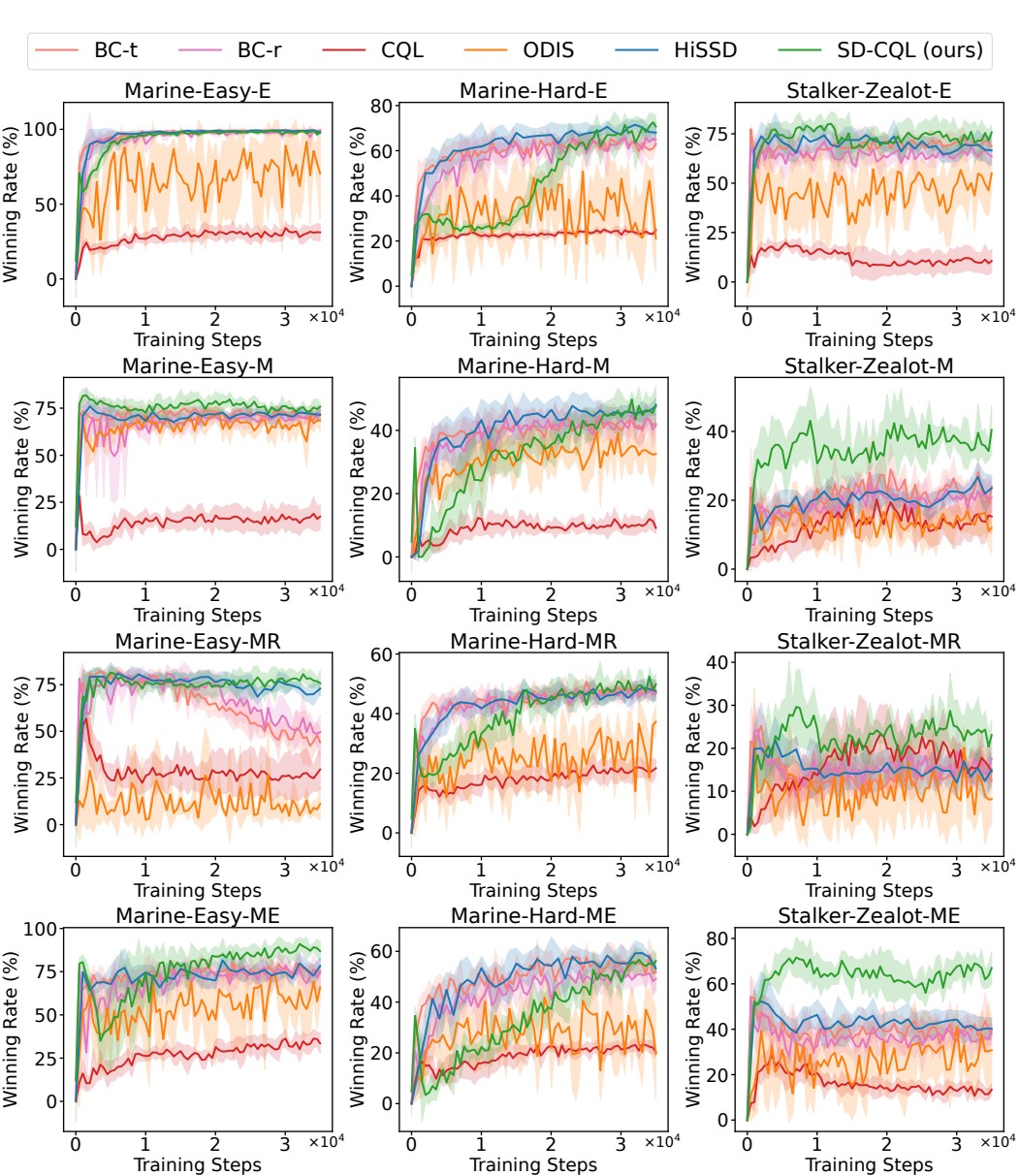

Figure 7: Average winning rates on Multi-to-Multi task sets. We report results over 5 random seeds, where the solid line represents the mean and the shaded area represents one standard deviation.

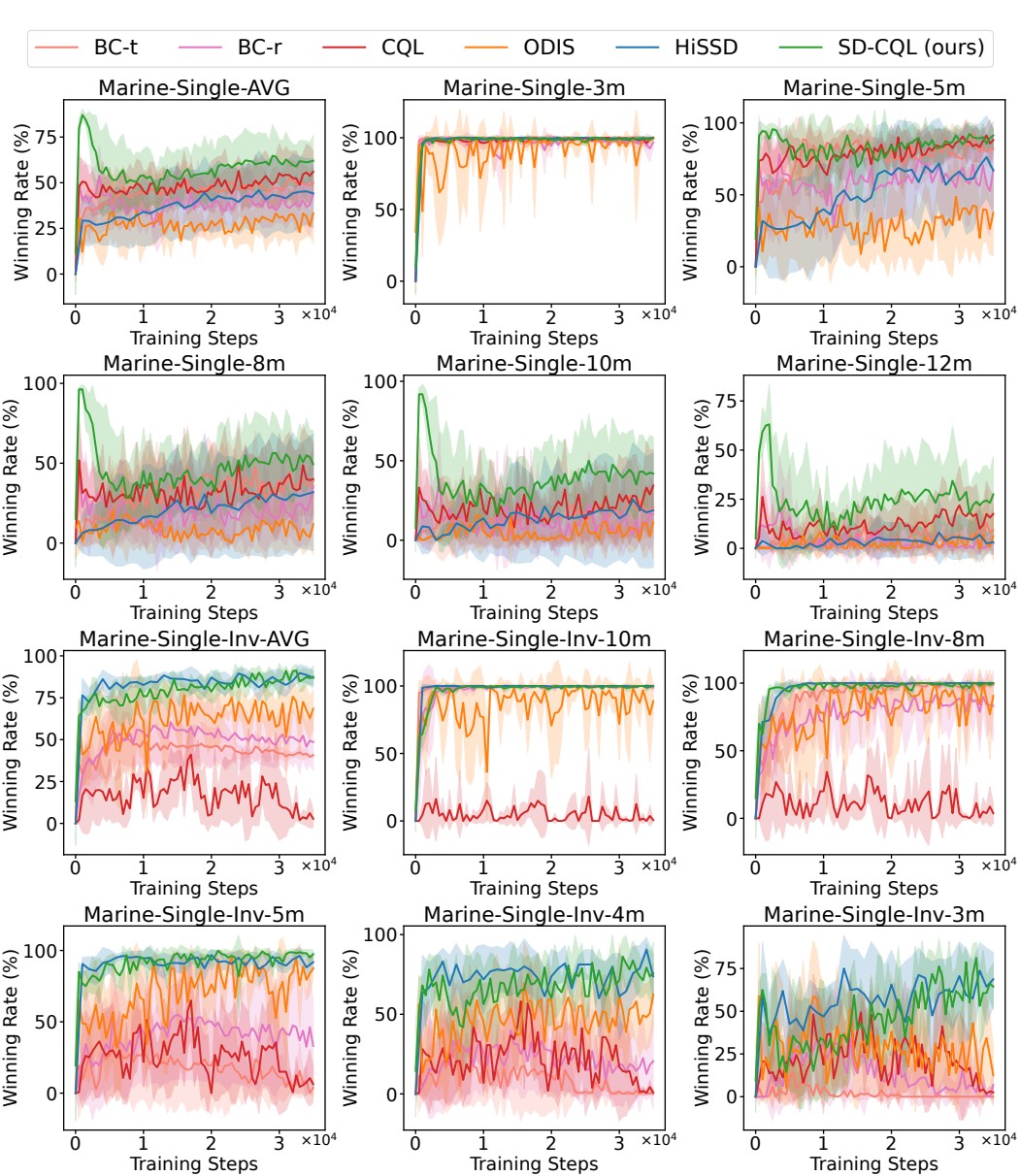

Figure 8: Average and task-specific winning rates on One-to-Multi task sets. We report results over 5 random seeds, where the solid line represents the mean and the shaded area represents one standard deviation.

