# OpenReview forum: "Beyond Shallow Behavior: Task-Efficient Value-Based Multi-Task Offline MARL via Skill Discovery"
_ICLR.cc/2026/Conference — Submitted to ICLR 2026_

### Official Review · Reviewer_ozZf · 2025-10-19

**Soundness:** 2
**Presentation:** 2
**Contribution:** 2
**Rating:** 2
**Confidence:** 4

**Summary:**

This paper explores the problem of multi-task offline multi-agent reinforcement learning (MARL), where agents must learn policies that generalize across multiple tasks using only static offline datasets. Existing offline MARL approaches are typically task-specific, requiring retraining when task configurations (e.g., number of agents) change, leading to inefficiency and poor generalization.
To address this, the authors propose Skill-Discovery Conservative Q-Learning (SD-CQL), a value-based algorithm that integrates observation-encoding-based skill discovery and conservative policy optimization to enhance task efficiency and cross-task generalization.
The proposed method is evaluated on 14 StarCraft II offline task sets derived from the SMAC benchmark, covering both Multi-to-Multi (training and testing on multiple tasks) and One-to-Multi (training on one task and testing on unseen ones) scenarios. Across datasets of varying quality, SD-CQL achieves the best performance on 13 out of 14 task sets, showing up to 68.9% improvement over existing baselines.

**Strengths:**

- The paper explores the under-investigated yet practical problem of multi-task offline MARL. While offline MARL itself has recently gained traction, the extension to an offline multi-task formulation—where agents must generalize across tasks with varying numbers of agents and configurations—remains relatively unexplored.

- From an originality standpoint, the paper contributes a skill-based framework that moves beyond behavior-cloning-based action decoding. Prior skill-discovery methods in RL often relied on direct imitation from latent skills to actions, which limited their optimality and generalization.

- The experimental evaluation is extensive, spanning 14 task sets in the StarCraft II SMAC benchmark. The results are consistently strong, with SD-CQL outperforming prior baselines in 13 out of 14 task sets, and showing large relative improvements.

**Weaknesses:**

1. Questionable definition of skill via next-observation prediction:
The authors claim z represents "skills" (high-level decision patterns), but the learning mechanism is simply next-observation reconstruction. This is not skill learning—it's observation representation learning.
- Traditional skill-based RL defines skills as temporally extended action sequences or sub-policies.
- SD-CQL's z is extracted by predicting o_{t+1} from o_t, which is a standard technique for learning task-relevant observation features, not for capturing behavioral patterns
- Existing skill discovery methods (even task-agnostic ones) explicitly model action sequences or behavioral diversity, not just observations
The paper lacks justification for why observation encoding equals skill representation. Without modeling action sequences, calling z a "skill" contradicts established RL terminology.

2. Task-Agnostic Skill Discovery vs Task-Dependent Action Selection:
The authors emphasize "task-agnostic" skill discovery, but action selection is task-dependent.
- z is learned without rewards (task-agnostic)
- But Q(tau, a | z) is optimized with task-specific rewards (Eq. 5)
How does this enable generalization? It seems generalization comes from the shared encoder/Q-network architecture, not from the skill discovery mechanism itself. The interaction between these components is unclear.

3. Section 3 is Too Implementation-Heavy:
Section 3 focuses excessively on implementation details (entity decomposition, variable action spaces, network architectures) rather than conceptual contributions:
- Section 3.2.1 largely repeats CQL from Section 2.2
- Separate Q-networks (Eq. 4) is an engineering choice for variable action spaces, not a core contribution

**Questions:**

Q1: Skill Definition:
Why is next-observation reconstruction equivalent to learning "skills" (decision patterns) rather than just compressed observation features?

Q2: Source of Generalization:
Which component drives generalization: the shared observation encoder, the skill vectors z, or local value calibration? Can you provide ablations isolating each?

Q3: "Task-Agnostic" Clarification
Skills are learned without rewards but evaluated with task-specific Q-values—in what sense are they truly task-agnostic?

I would also like to hear the authors' thoughts on the weaknesses raised above.

---

> ### Author Response · Authors · 2025-11-20
> **Response to Reviewer ozZf (Part 1/3)**
>
> Thank you for your time and effort in reviewing our paper! We are grateful for your constructive suggestions, which have significantly guided our improvements. Please find our responses to your comments below.
>
> ### Weaknesses
> > **Weakness1:** Questionable definition of skill via next-observation prediction: The authors claim z represents "skills" (high-level decision patterns), but the learning mechanism is simply next-observation reconstruction. This is not skill learning—it's observation representation learning. Traditional skill-based RL defines skills as temporally extended action sequences or sub-policies. SD-CQL's z is extracted by predicting o_{t+1} from o_t, which is a standard technique for learning task-relevant observation features, not for capturing behavioral patterns. Existing skill discovery methods (even task-agnostic ones) explicitly model action sequences or behavioral diversity, not just observations The paper lacks justification for why observation encoding equals skill representation. Without modeling action sequences, calling z a "skill" contradicts established RL terminology.
>
>
> Thank you for the comment! Following prior work[1][2](e.g., the “common skill” terminology in HiSSD[2]), our term *skill* refers to a **broader concept of distinguishable behavioral characteristics**, and is not necessarily limited to an explicit action sequence or a sub-policy. We clarify this term usage in the footnote in the revised manuscript.
>
> However, we would like to emphasize that **our next-observation reconstruction goes far beyond observation representation learning**. Instead, it enables the learned latent $z$ to **characterize agents’ behavioral patterns**:
>
> 1. As shown in the **visualizations in Appendix E**, the skills obtained through next-observation reconstruction **align well with agents’ actual behavioral modes and exhibit stronger representational expressiveness** than those learned by directly modeling actions.
> 2. As noted in the main text, in MARL it is standard that the current observation $o_t$ includes the **previous action**, ensuring that temporal dependencies are accessible to the encoder.
> 3. Since SD-CQL is trained purely on pre-collected offline data with no online interaction, the next observation $o_{t+1}$ in the dataset is always the result of the agent’s action at time $t$. Consequently, modeling the transition $o_t \rightarrow o_{t+1}$ implicitly captures the **short-term behavioral tendencies of each agent** exhibited in these offline trajectories. Furthermore, by introducing a **recurrent latent variable $h$** and training on full trajectories, the encoder also learns **long-term behavioral patterns**.
> 4. We observe that in **multi-task offline MARL**, existing methods[1][2] that directly decode actions from latent variables (i.e., behavior-cloning-style approaches) often suffer from **skill–action mismatch due to the lack of environment interaction**, leading to poor cross-task generalization. In contrast, SD-CQL **reconstructs the next observation to implicitly discover skill representations**, and then **optimizes actions via skill-conditioned Q-values**, effectively mitigating this mismatch and achieving **superior performance and generalization**.
>
> #### **References**
> ```
> [1] Zhang, F. et al., Discovering Generalizable Multi-agent Coordination Skills from Multi-task Offline Data. ICLR 2023.
> [2] Liu, S. et al., Learning Generalizable Skills from Offline Multi-Task Data for Multi-Agent Cooperation. ICLR 2025.
> ```

---

> > ### Author Response · Authors · 2025-11-20
> > **Response to Reviewer ozZf (Part 2/3)**
> >
> > > **Weakness2:** Task-Agnostic Skill Discovery vs Task-Dependent Action Selection: The authors emphasize "task-agnostic" skill discovery, but action selection is task-dependent. z is learned without rewards (task-agnostic). But Q(tau, a | z) is optimized with task-specific rewards (Eq. 5) How does this enable generalization? It seems generalization comes from the shared encoder/Q-network architecture, not from the skill discovery mechanism itself. The interaction between these components is unclear.
> >
> >
> > We sincerely appreciate your observation! We would like to emphasize that:
> >
> > - This is in fact an intentional design choice for better generalization. Our framework performs **skill discovery from task-agnostic signals** to extract **generalizable behavioral patterns** from the dataset, while using **task-specific information (e.g., rewards)** to optimize the **skill-conditioned Q-values**. This separation allows us to learn the optimal actions under each behavioral mode and **effectively resolve the skill–action optimality mismatch** issue that often arises in prior methods[1][2] that decode actions directly from skills.
> >
> > - Moreover, since the offfline training datasets include trajectories collected from **multiple source tasks with similar structure** (a common setting in existing multi-task offline MARL[1][2]), our CQL objective naturally **biases the learned policy toward actions that are consistently effective across tasks**, further enhancing generalization. The shared encoder and Q-network parameters across agents follow the **standard CTDE paradigm in MARL**, which improves scalability and reduces the total parameter count.
> >
> > **The Generalization indeed comes from skill discovery.** As shown in the ablation study (Appendix F.1.1, Table 15), removing the skill discovery module leads to a significant drop in generalization performance, confirming its critical contribution.
> >
> > The interactions among modules are illustrated in Figure 3:
> >
> > 1. In the **skill discovery**, the encoder and decoder compute the reconstruction loss.
> > 2. In the **policy optimization**, the encoder encodes each agent’s local observation into the skill vector $z$ and feature representation $\{E_{i,k}\}$. As introduced in Section 3, they are concatenated and fed into the Q-network for CTDE optimization.
> > 3. A **stop-gradient** is applied between the encoder and Q-network to prevent mutual interference during training.
> >
> > #### **References**
> > ```
> > [1] Zhang, F. et al., Discovering Generalizable Multi-agent Coordination Skills from Multi-task Offline Data. ICLR 2023.
> > [2] Liu, S. et al., Learning Generalizable Skills from Offline Multi-Task Data for Multi-Agent Cooperation. ICLR 2025.
> > ```
> >
> > > **Weakness3:** Section 3 is Too Implementation-Heavy: Section 3 focuses excessively on implementation details (entity decomposition, variable action spaces, network architectures) rather than conceptual contributions: Section 3.2.1 largely repeats CQL from Section 2.2. Separate Q-networks (Eq. 4) is an engineering choice for variable action spaces, not a core contribution
> >
> > Thank you for the comment. As stated at the beginning of Section 3, achieving **effective skill discovery** and **scalable generalization** in the **multi-task offline MARL** setting is non-trivial. To this end, we propose two key components: **next local-observation reconstruction** and **skill-conditioned policy optimization**, which together constitute our **conceptual contribution**. To illustrate how they work, we describe several essential design choices, while more concrete **implementation details and architectures** are provided in **Appendix D** rather than Section 3.
> >
> > We also introduce the **necessary preliminaries** (e.g., observation decomposition for variable-length inputs) to make the algorithm **complete** and **easy to follow**.
> >
> > Regarding the perceived repetition of CQL with Section 2, we would like to clarify that Section 2 only offers **a brief background on the standard CQL regularization idea**, whereas Section 3.2.1 specifies **how CQL is instantiated for a skill-conditioned Q-function** and details the **overall Q-learning training procedure**. We consider this necessary for **completeness and readability**.
> >
> > Finally, **our core contributions are exactly those summarized at the end of the Introduction**: We investigate a rarely explored yet important field: multi-task offline MARL, analyze current limitations, and propose a novel value-based method to demonstrate that such approaches are not only viable but also superior in this setting, offering a new perspective for future research.
> >
> > That said, if you have any specific suggestions regarding the organization of this section, we would be very grateful and will revise the paper accordingly.

---

> > > ### Author Response · Authors · 2025-11-20
> > > **Response to Reviewer ozZf (Part 3/3)**
> > >
> > > ### Questions
> > > > **Q1:** Skill Definition: Why is next-observation reconstruction equivalent to learning "skills" (decision patterns) rather than just compressed observation features?
> > >
> > > Thank you for your question! As mentioned in our response to Weakness 1, in addition to the last action provided by current observation $o_t$, our next-observation reconstruction models the transition $o_t \rightarrow o_{t+1}$. Since $o_{t+1}$ results from the agent’s chosen action at time $t$, such reconstruction implicitly captures **short-term behavioral tendencies** present in the offline trajectories. Furthermore, by introducing a **recurrent latent variable $h$** and training on full trajectories, the encoder also learns **long-term behavioral patterns**.
> > >
> > > > **Q2:** Source of Generalization: Which component drives generalization: the shared observation encoder, the skill vectors z, or local value calibration? Can you provide ablations isolating each?
> > >
> > >
> > > Thank you for your question! We would like to clarify that the **generalization capability** mainly stems from our **skill-discovery mechanism** and the learned **skill vectors $z$**. The ablation studies in **Appendix F.1 (Tables 15 and 16)** confirm that removing the skill discovery module or altering the skill representation leads to a drop in generalization performance.
> > >
> > > The local value calibration module, in contrast, is designed to mitigate **Q-value estimation error accumulation** when the number of agents becomes large, which is an inherent challenge in **value-based offline MARL**, rather than a direct driver of generalization.
> > >
> > > Finally, the **shared encoder parameters** follow the standard **CTDE paradigm** in MARL, which improves scalability and reduces the overall parameter count, but are not the primary source of generalization.
> > >
> > > > **Q3:**  "Task-Agnostic" Clarification Skills are learned without rewards but evaluated with task-specific Q-values—in what sense are they truly task-agnostic?
> > >
> > >
> > > Thank you for your question! We would like to clarify that the **Q-function is not used to evaluate the skill latent $z$** itself. Instead, it jointly evaluates **actions** based on the current observation history, **the skill vector $z$**, and the reward signal, **enabling more accurate policy optimization and resolving the skill–action mismatch problem that often exists in prior methods.**
> > >
> > > Regarding the **task-agnostic** nature of the skill discovery, as we explained in our response to Weakness 2, this is an **intentional design choice** that has proven to yield **better generalization and scalability** in our experiments.
> > >
> > > ---
> > > We hope our response addresses your concerns. If so, we wonder if you could kindly consider raising your score? We will also be happy to answer any further questions you may have. Thank you very much!

---

> ### Author Response · Authors · 2025-11-25
> **Looking Forward to Your Feedback**
>
> Dear Reviewer ozZf,
>
> Thank you for your time and feedback! We hope our response and the revision have fully addressed your concerns. If so, we would appreciate it if you could raise your score.
>
> If you have any remaining concerns, we are more than happy to discuss them further.
>
> Best regards,
>
> The Authors

---

### Official Review · Reviewer_kwKd · 2025-10-28

**Soundness:** 2
**Presentation:** 2
**Contribution:** 2
**Rating:** 4
**Confidence:** 4

**Summary:**

This paper tackles task-specific inefficiency in multi-task offline MARL. It proposes Skill-Discovery Conservative Q-Learning (SD-CQL), a value-based algorithm designed to learn generalizable policies from limited offline data. The method's core components are: 1) A skill discovery module usingnext local observation reconstruction to learn continuous skill vectors. 2) A skill-conditioned policy optimized with Conservative Q-Learning (CQL), using separate Q-networks for fixed and variable actions. 3) A "Local Value Calibration" (LVC) regularizer to enhance stability. Experiments on StarCraft II benchmarks show that SD-CQL significantly outperforms baselines in both multi-to-multi and one-to-multi (task-efficient) transfer scenarios.

**Strengths:**

- The paper addresses the critical and practical problem of zero-shot generalization in offline MARL, a key step for real-world applications with high data collection costs.
- The empirical evaluation is comprehensive, testing on 14 SMAC task sets across various data qualities and two transfer settings. The SOTA results (13/14 sets) are strong and supported by ablation studies in the appendix.
- The paper is well-structured. The method is presented logically.

**Weaknesses:**

1. Clarity of Notation/Figures: The paper contains ambiguities.
    1. Figure 1.a plots an undefined "CFCQL" baseline.
    2. The Q-function notation in Sec 3.2.1, such as $Q(a|o,z)$, is non-standard. The`|`(conditional) notation is incorrectly used and should be replaced with standard Q-function notation, e.g.,$Q(o, z, a)$.
    3. It is recommended that bolded entries in all result tables indicate only the best result. The meaning of bolded entries differs between the main text and appendix tables.
2. Hyperparameter Sensitivity: The LVC coefficient,$\eta$, appears highly sensitive, varying widely (0.1 to 0.9) across tasks (Table 8). This reliance on careful, task-specific tuning is a significant practical limitation, as confirmed by ablation in Table 14.
3. Conflated Generalization Metrics: The main claim is improved generalization, but Table 1 reports average performance across all source and unseen tasks combined. This metric conflates performance on known tasks with generalization to unknown tasks. Generalization should be demonstrated by reporting performance on unseen tasks separately.
4. Baseline Validation: The paper compares against re-implemented baselines. There is no validation (e.g., in the appendix) comparing these re-implementations to the results from the original papers. This is necessary to ensure the comparisons are fair and the baselines are robust.

**Questions:**

1. Encoder Notation (Lines 208-213): Please clarify the encoder notation. What do$K$, $K_i$, and $K_a$ represent? What is the precise meaning of $K_i+1$ in the expression $h=e_{i,K_i+1}$?
2. Hidden Variable`h` (Lines 214-215): What is the concrete implementation of `h`?  Why is`h`not also used in the decoder? The paper claims 'decoder enable the skill to capture temporal information'. Are there any relevant ablation?
3. Stop Gradient (Fig. 3): What is the motivation for the stop-gradient (Line 285)? I hope there is relevant ablation. In addition, since L_rec and L_Q are completely independent, have the authors tried a two-stage optimization (i.e., pre-train/freeze the skill module, then train the Q-networks) to reduce noise, instead of joint optimization?
4. LVC Hyperparameter $\eta$ (Table 8): Is there any intuition for setting $\eta$? $\mathcal{L}_{LVC}$ is a BC loss. Intuitively, "Expert" datasets should warrant a higher $\eta$, but Table 8 contradicts this (e.g., Marine-Hard-Expert $\eta=0.5$ < -Medium $\eta=0.8$ ).
5. Metrics and Baselines: Can the authors provide a version of Table 1 that disaggregates performance, showing separate average win rates for *source tasks* and *unseen tasks*? Can the authors add a table comparing their re-implemented baseline results against those reported in the original baseline papers?

---

> ### Author Response · Authors · 2025-11-20
> **Response to Reviewer kwKd (Part 1/4)**
>
> Thank you for your time and effort in reviewing our paper! We are grateful for your constructive suggestions, which have significantly guided our improvements. Please find our responses to your comments below.
>
> ### Weaknesses
> >**Weakness1** Clarity of Notation/Figures: The paper contains ambiguities. 1. Figure 1.a plots an undefined "CFCQL" baseline. 2. The Q-function notation in Sec 3.2.1, such as $Q(a|o,z)$, is non-standard. The|(conditional) notation is incorrectly used and should be replaced with standard Q-function notation, e.g., $Q(o,z,a)$. 3. It is recommended that bolded entries in all result tables indicate only the best result. The meaning of bolded entries differs between the main text and appendix tables.
>
> Thank you for your helpful suggestions! We have revised these ambiguities in our manuscript.
>
> >**Weakness2** Hyperparameter Sensitivity: The LVC coefficient, $\eta$, appears highly sensitive, varying widely (0.1 to 0.9) across tasks (Table 8). This reliance on careful, task-specific tuning is a significant practical limitation, as confirmed by ablation in Table 14.
>
> Thank you for your comment! The parameter $\eta$ is indeed an important hyperparameter. In practice, its selection follows some intuitive principles to balance **in-task performance** and **cross-task generalization**. The appropriate value, however, is mainly determined by the characteristics of the whole source task set, rather than being tied to any specific single task.
>
> For example, in Table 17 in the revision, we find that as $\eta$ increases, the win rates on the source tasks **all approach nearly 100%**, while the performance on unseen tasks **steadily degrades**, leading to a drop in overall average performance.
>
> More broadly, in multi-task offline MARL, such a trade-off remains a challenging open problem due to its complexity, and we plan to explore more effective tuning and adaptive approaches in future work.
>
> >**Weakness3** Conflated Generalization Metrics: The main claim is improved generalization, but Table 1 reports average performance across all source and unseen tasks combined. This metric conflates performance on known tasks with generalization to unknown tasks. Generalization should be demonstrated by reporting performance on unseen tasks separately.
>
> Thank you for your comment! In Table 1 in the revision, we report averages over all tasks to assess each algorithm’s overall capability across both source and unseen tasks. For completeness, the table below lists the averages on the unseen tasks only. As shown, SD-CQL exhibits a more pronounced advantage on 13 task sets, further substantiating its generalization ability. We have add these results in Appendix I.
>
> | Task Sets | BC-r | BC-t | CQL | ODIS | HiSSD | SDCQL (ours) |
> |---|---|---|---|---|---|---
> | marine-easy-e | **99.29** $\pm$ 0.46 | 98.48 $\pm$ 0.92 | 18.84 $\pm$ 7.54|64.38 $\pm$ 34.86 | 97.41 $\pm$ 1.86 | 97.86 $\pm$ 1.07 |
> | marine-easy-m | 71.7 $\pm$ 6.07 | 72.5 $\pm$ 4.17 | 8.48 $\pm$ 4.12 |67.77 $\pm$ 5.32 | 70.54 $\pm$ 3.43 | **73.66** $\pm$ 4.35 |
> | marine-easy-mr | 48.04 $\pm$ 10.78 | 40.98 $\pm$ 10.39 |22.59 $\pm$ 13.56 |6.34 $\pm$ 9.86 | 70.27 $\pm$ 3.82 | **74.02** $\pm$ 5.31 |
> | marine-easy-me| 71.88 $\pm$ 8.0 | 70.45 $\pm$ 2.73 |24.11 $\pm$ 4.87 |63.75 $\pm$ 11.41 | 74.29 $\pm$ 4.16 | **84.02** $\pm$ 7.24 |
> | marine-hard-e| 57.85 $\pm$ 3.91 | 54.93 $\pm$ 2.3 | 22.08 $\pm$ 2.11|14.44 $\pm$ 12.06 | 60.0 $\pm$ 5.73 | **65.97** $\pm$ 5.9 |
> | marine-hard-m | 38.06 $\pm$ 3.57 | 38.61 $\pm$ 3.51 |2.99 $\pm$ 2.11 |28.06 $\pm$ 4.23 | 44.65 $\pm$ 3.39 | **45.9** $\pm$ 6.99 |
> | marine-hard-mr| 45.35 $\pm$ 2.32 | 46.53 $\pm$ 3.02 |17.64 $\pm$ 3.97  |38.75 $\pm$ 6.23 | 45.14 $\pm$ 3.59 | **50.62** $\pm$ 3.59 |
> | marine-hard-me| 42.71 $\pm$ 6.81 | 49.24 $\pm$ 5.97 | 15.35 $\pm$ 2.9 |16.53 $\pm$ 17.28 | 52.22 $\pm$ 1.02 | **56.18** $\pm$ 4.48 |
> | stalker-zealot-e | 56.38 $\pm$ 3.97 | 62.12 $\pm$ 4.39 |10.0 $\pm$ 5.36 |49.81 $\pm$ 7.58 | 60.25 $\pm$ 9.57 | **74.19** $\pm$ 4.52 |
> | stalker-zealot-m  | 17.06 $\pm$ 2.74 | 20.81 $\pm$ 3.53 |11.31 $\pm$ 4.72  |9.94 $\pm$ 8.09 | 20.31 $\pm$ 4.74 | **40.94** $\pm$ 6.74 |
> | stalker-zealot-mr | 18.19 $\pm$ 5.49 | 11.5 $\pm$ 1.43 |11.37 $\pm$ 2.21 |7.56 $\pm$ 5.85 | 16.31 $\pm$ 5.6 | **24.56** $\pm$ 5.06 |
> | stalker-zealot-me | 32.06 $\pm$ 5.98 | 31.06 $\pm$ 6.38 | 12.5 $\pm$ 3.1|26.06 $\pm$ 12.29 | 34.81 $\pm$ 3.39 | **63.44** $\pm$ 7.5 |

---

> > ### Author Response · Authors · 2025-11-20
> > **Response to Reviewer kwKd (Part 2/4)**
> >
> > >**Weakness4** Baseline Validation: The paper compares against re-implemented baselines. There is no validation (e.g., in the appendix) comparing these re-implementations to the results from the original papers. This is necessary to ensure the comparisons are fair and the baselines are robust.
> >
> > Thank you for the suggestion! We have added the requested validation to Appendix D.3.3. We also wish to clarify that for all baselines in Table 1, we **carefully matched the recommended hyperparameters and ran the authors’ official implementations** for ODIS[1] and HiSSD[2] (rather than our own re-implementations). The results reported in their papers are summarized in the table below. We do observe some discrepancies: for example, the BC-best scores are relatively higher than those reported in [1]. In addition, both our results and those reported in the ODIS paper are based on the **final policy**, whereas the HiSSD paper reports the **best policy**, which may further account for differences.
> >
> > | Task Sets | BC-best (Our, Final) | BC-best ([1], Final) |
> > |---|---|---|
> > | marine-easy-e | 99.38 | 82.42 |
> > | marine-easy-m | 71.81 | 70.57 |
> > | marine-easy-mr | 49.62 | 16.07 |
> > | marine-easy-me| 75.25 | 58.52 |
> > | marine-hard-e| 64.84 | 55.40 |
> > | marine-hard-m | 42.08| 31.81 |
> > | marine-hard-mr| 46.98 | 42.55 |
> > | marine-hard-me| 51.93 | 39.51 |
> > | stalker-zealot-e | 68.61 | 67.15 |
> > | stalker-zealot-m  | 23.94 | 21.84|
> > | stalker-zealot-mr | 17.64 | 13.54 |
> > | stalker-zealot-me | 39.62 | 40.98 |
> >
> > | Task Sets | ODIS (Our, Final) | ODIS ([1], Final) |
> > |---|---|---|
> > | marine-easy-e | 70.50 | 85.83
> > | marine-easy-m |68.50 | 75.3
> > | marine-easy-mr | 11.06 | 11.25
> > | marine-easy-me| 66.00 | 71.6
> > | marine-hard-e| 21.09 | 55.02
> > | marine-hard-m | 32.55| 41.43
> > | marine-hard-mr| 37.29 | 43.07
> > | marine-hard-me| 19.58 | 42.33
> > | stalker-zealot-e | 55.05 | 67.27
> > | stalker-zealot-m  | 11.97 | 27.52
> > | stalker-zealot-mr | 8.17 | 11.72
> > | stalker-zealot-me | 30.72 | 40.27
> >
> > | Task Sets | HiSSD (Our, Final) | HiSSD ([2], Best) |
> > |---|---|---|
> > | marine-easy-e | 97.94 |98.64 |
> > | marine-easy-m |  71.44 | 78.74 |
> > | marine-easy-mr | 72.88 | 79.97 |
> > | marine-easy-me| 78.31 | 74.34 |
> > | marine-hard-e|  68.02  | 68.15 |
> > | marine-hard-m | 47.60  | 49.02 |
> > | marine-hard-mr| 47.55  | 48.59 |
> > | marine-hard-me|  53.28  | 59.53 |
> > | stalker-zealot-e |  66.59  | 73.84 |
> > | stalker-zealot-m  | 23.51 | 18.42 |
> > | stalker-zealot-mr |  15.00  | 12.94 |
> > | stalker-zealot-me |  40.29  | 43.91 |
> >
> > #### **References**
> > ```
> > [1] Zhang, F. et al., Discovering Generalizable Multi-agent Coordination Skills from Multi-task Offline Data. ICLR 2023.
> > [2] Liu, S. et al., Learning Generalizable Skills from Offline Multi-Task Data for Multi-Agent Cooperation. ICLR 2025.
> > ```
> >
> >
> > ### Questions
> > >**Question1** Encoder Notation (Lines 208-213): Please clarify the encoder notation. What do $K$, $K_i$, and $K_a$ represent? What is the precise meaning of $K_i+1$ in the expression $h=e_{i,K_i+1}$?
> >
> > We apologize for the confusion, we have fixed these issues in the revision. In our notation, $K_i$ denotes the number of entities observable to agent $i$. Among these $K_i$ entities, the subset that agent $i$ can interact with by actions has cardinality $K_i^{a}$ (formerly $K_a$; we changed it to $K_i^{a}$ to avoid ambiguity in revision). The symbol $K$ in Figure 3 is a typo, which also means $K_i^a$.
> >
> > The precise meaning of $K_i+1$ in $h = e_{i,,K_i+1}$ is that the latent variable $h$ is represented in the **same embedding space** as other entity embeddings; we treat $h$ as an additional “pseudo-entity” appended to the set and feed it to the encoder at every time step recurrently. Thus $e_{i,,K_i+1}$ reads as “the $(K_i+1)$-th embedding” for agent $i$, emphasizing this uniform treatment.
> >
> >
> > >**Question2** Hidden Variable h (Lines 214-215): What is the concrete implementation of h? Why ishnot also used in the decoder? The paper claims decoder enable the skill to capture temporal information'. Are there any relevant ablation?
> >
> > Thank you for your question! In our implementation, the recurrent latent $h$ is a tensor of shape $({BatchSize}\times n_{{agent}}, d_{{emb}})$ initialized to zeros. At each time step, we append $h_t$ to the end of the entity-embedding list and feed the concatenated sequence to the encoder (i.e., treating $h_t$ treated as $e_{i,K_i+1}$). The resulting updated latent $h_{t+1}$ is then used at the next step.
> >
> > Additionally, we do **not** claim that the decoder enables the skill to capture temporal information. On the contrary, our claim is that **introducing the recurrent latent only in the encoder, without passing it to the decoder, helps the skill capture temporal information**. The ablation in Appendix F.1.2, Table 16 shows that removing $h$ leads to performance degradation.

---

> > > ### Author Response · Authors · 2025-11-20
> > > **Response to Reviewer kwKd (Part 3/4)**
> > >
> > > >**Question3** Stop Gradient (Fig. 3): What is the motivation for the stop-gradient (Line 285)? I hope there is relevant ablation. In addition, since L_rec and L_Q are completely independent, have the authors tried a two-stage optimization (i.e., pre-train/freeze the skill module, then train the Q-networks) to reduce noise, instead of joint optimization?
> > >
> > > Thank you for your suggestion! We apply a stop-gradient to prevent task-specific signals used in CTDE training (e.g., the global-state and task-reward) from leaking into the encoder. This avoids gradient interference that would bias the skill latent toward task-specific artifacts and hurt generalization. The following table presents our ablation results on the marine-single and marine-single-inv task sets. As shown, gradient interference indeed degrades performance. We have added this ablation to Table 16 in Appendix F.
> > >
> > > | Task Sets | w/o stop-gradient| SDCQL (ours) |
> > > |---|---|---|
> > > |marine-single|59.13 $\pm$ 14.42|62.00 $\pm$ 13.80
> > > |marine-single-inv|84.38 $\pm$ 8.24| 87.50 $\pm$ 6.54
> > >
> > > As to the two-stage optimization, we indeed found that it made training more stable. However, in some cases, we observed that joint training, while introducing mild noise at the early stage, could benefit offline Q-learning, helping the policy achieve better generalization on unseen tasks. In addition, the two-stage setup would require introducing an extra hyperparameter (i.e., the number of pretraining steps). For these reasons, we ultimately chose the joint training approach, which provides a balance between stability, simplicity, and generalization.
> > >
> > >
> > > >**Question4** LVC Hyperparameter $\eta$ (Table 8): Is there any intuition for setting $\eta$? $L_{LVC}$ is a BC loss. Intuitively, "Expert" datasets should warrant a higher $\eta$, but Table 8 contradicts this (e.g., Marine-Hard-Expert $\eta=0.5$ < -Medium $\eta=0.8$).
> > >
> > >
> > > Thank you very much for the careful reading and for pointing this out! As discussed in **Weakness 2**, the choice of $\eta$ in **multi-task offline MARL** must balance not only **in-task performance** but also **cross-task generalization**. A small $\eta$ could struggle to reduce estimation error at scale, while a large one may limit the policy to the average behavoir in the datasets, hindering the generalization.
> > >
> > > A practical guideline is to start from a moderate value (e.g., $\eta = 0.5$). When we need to **reduce Q-value estimation error**, for example, with lower-quality datasets or larger agent populations, a larger $\eta$ tends to work better. Conversely, when Q-value can be estimated well, such as with higher-quality data or fewer agents, a smaller $\eta$ is generally more favorable for generalization. We add a remark about this intuition in Appendix D.4.

---

> > > > ### Author Response · Authors · 2025-11-20
> > > > **Response to Reviewer kwKd (Part 4/4)**
> > > >
> > > > >**Question5** Metrics and Baselines: Can the authors provide a version of Table 1 that disaggregates performance, showing separate average win rates for source tasks and unseen tasks? Can the authors add a table comparing their re-implemented baseline results against those reported in the original baseline papers?
> > > >
> > > > In Table 1, we report averages over all tasks to assess each algorithm’s overall capability across both source and unseen tasks. In our response to Weakness 3, we reported the averages on the unseen tasks from Table 1 and compared our baseline runs with the results reported in the original papers. For completeness, the table below reports the averages on the source tasks from Table 1. We have added this table in Appendix I.
> > > >
> > > > | Task Sets | BC-r | BC-t | CQL | ODIS | HiSSD | SDCQL (ours) |
> > > > |---|---|---|---|---|---|---|
> > > > | marine-easy-e | 99.58 $\pm$ 0.51 | 99.79 $\pm$ 0.42 | 59.79 $\pm$ 3.4 | 84.79 $\pm$ 19.98 | 99.17 $\pm$ 1.21 | 98.54 $\pm$ 1.06 |
> > > > | marine-easy-m | 71.67 $\pm$ 2.41 | 70.21 $\pm$ 6.64 | 38.12 $\pm$ 13.89 | 70.21 $\pm$ 7.29 | 73.54 $\pm$ 2.76 | 80.83 $\pm$ 5.57 |
> > > > | marine-easy-mr | 53.33 $\pm$ 9.58 | 51.46 $\pm$ 4.69 | 45.83 $\pm$ 3.49 | 22.08 $\pm$ 11.21 | 78.96 $\pm$ 5.25 | 80.0 $\pm$ 11.3 |
> > > > | marine-easy-me | 83.12 $\pm$ 6.76 | 82.71 $\pm$ 10.43 | 55.62 $\pm$ 9.52 | 71.25 $\pm$ 12.27 | 87.71 $\pm$ 12.38 | 93.54 $\pm$ 4.29 |
> > > > | marine-hard-e | 85.83 $\pm$ 5.8 | 87.08 $\pm$ 3.06 | 33.12 $\pm$ 0.78 | 41.04 $\pm$ 25.36 | 92.08 $\pm$ 1.56 | 84.79 $\pm$ 3.13 |
> > > > | marine-hard-m | 51.88 $\pm$ 6.57 | 52.5 $\pm$ 3.52 | 27.71 $\pm$ 2.68 | 46.04 $\pm$ 4.86 | 56.46 $\pm$ 0.78 | 54.79 $\pm$ 3.33 |
> > > > | marine-hard-mr | 51.88 $\pm$ 5.72 | 47.29 $\pm$ 6.1 | 33.75 $\pm$ 4.4 | 32.92 $\pm$ 4.96 | 54.79 $\pm$ 6.51 | 47.5 $\pm$ 5.21 |
> > > > | marine-hard-me | 68.96 $\pm$ 9.89 | 60.0 $\pm$ 15.29 | 35.21 $\pm$ 1.67 | 28.75 $\pm$ 20.88 | 56.46 $\pm$ 6.5 | 56.25 $\pm$ 3.09 |
> > > > | stalker-zealot-e | 87.29 $\pm$ 5.03 | 90.21 $\pm$ 2.84 | 12.71 $\pm$ 7.89 | 72.5 $\pm$ 14.86 | 87.71 $\pm$ 2.02 | 80.83 $\pm$ 4.69 |
> > > > | stalker-zealot-m | 33.33 $\pm$ 8.62 | 34.38 $\pm$ 5.74 | 28.54 $\pm$ 7.78 | 18.75 $\pm$ 6.81 | 34.17 $\pm$ 4.9 | 38.75 $\pm$ 9.8 |
> > > > | stalker-zealot-mr | 15.83 $\pm$ 1.38 | 15.83 $\pm$ 3.32 | 25.21 $\pm$ 6.7 | 10.21 $\pm$ 9.02 | 10.62 $\pm$ 3.39 | 18.12 $\pm$ 3.52 |
> > > > | stalker-zealot-me | 64.79 $\pm$ 8.77 | 61.88 $\pm$ 13.35 | 16.46 $\pm$ 3.39 | 46.25 $\pm$ 26.19 | 58.54 $\pm$ 10.06 | 78.75 $\pm$ 7.81 |
> > > >
> > > > ---
> > > > We hope our response addresses your concerns. If so, we wonder if you could kindly consider raising your score? We will also be happy to answer any further questions you may have. Thank you very much!

---

> ### Author Response · Authors · 2025-11-25
> **Looking Forward to Your Feedback**
>
> Dear Reviewer kwKd,
>
> Thank you for your time and feedback! We hope our response and the revision have fully addressed your concerns. If so, we would appreciate it if you could raise your score.
>
> If you have any remaining concerns, we are more than happy to discuss them further.
>
> Best regards,
>
> The Authors

---

> > ### Comment · Reviewer_kwKd · 2025-11-27
> >
> > Thank you for your responses. However, I still have some remaining concerns.
> >
> > **Regarding W1:** I remain unclear about the significance of the CFCQL baseline, which appears solely in the introduction. Why was this baseline excluded from the experimental comparisons? Furthermore, why do Fig.1a and Fig.1b utilize different baselines for comparison? This inconsistency leads to a confusing textual description of Fig.1 (Lines 71-88). The "ODIS" algorithm appears in Fig.1a, but there is no explanation provided for it in the corresponding text description.
> >
> > **Regarding W2/Q4:** The Remark in Appendix D.4 does not adequately address my query. The empirical rule provided by the authors suggests using a large η for lower-quality datasets and a small η for higher-quality datasets. However, the hyperparameter settings presented in Table 11 appear to contradict this rule, as the two groups of configurations exhibit inconsistent patterns:
> >
> > - **Marine-Easy:** Expert (higher-quality) η=0.9 **>** Medium (lower-quality) η=0.2 (Contradicts the rule)
> > - **Marine-Hard:** Expert (higher-quality) η=0.5 **<** Medium (lower-quality) η=0.8 (Consistent with the rule)
> >
> > Moreover, the ablation study in Table 17 indicates that a smaller η consistently yields better results in the two tested environments. Why, then, was η=0.9 selected for other environments (e.g., Marine-Easy-Expert)? Were ablation studies conducted for these specific environments as well? If so, I request that these additional experimental results be included.
> >
> > **Regarding Q2:** I apologize if my previous phrasing was unclear. My specific question is: **Why does the decoder not utilize the recurrent latent h?** The paper does not appear to provide a justification for this design choice. While Appendix F.1.2 demonstrates the benefits of using h in the *encoder,* there is no corresponding ablation study analyzing the exclusion of h from the *decoder*.

---

> > > ### Author Response · Authors · 2025-12-02
> > > **Response to Official Comment by Reviewer kwKd (Part 1/3)**
> > >
> > > Thank you for your reply. Please find our additional responses below.
> > > > Regarding W1: I remain unclear about the significance of the CFCQL baseline, which appears solely in the introduction. Why was this baseline excluded from the experimental comparisons? Furthermore, why do Fig.1a and Fig.1b utilize different baselines for comparison? This inconsistency leads to a confusing textual description of Fig.1 (Lines 71-88). The "ODIS" algorithm appears in Fig.1a, but there is no explanation provided for it in the corresponding text description.
> > >
> > > Thank you for the questions. We would like to clarify that:
> > >
> > > ### About CFCQL
> > > CFCQL is a representative and strong **task-specific** offline MARL algorithm: it extends CQL with counterfactual corrections to alleviate Q-value overestimation in multi-agent settings. We therefore use it in Fig. 1a as a *task-specific* baseline to illustrate that such task-specific methods have limited cross-task generalization. In contrast, our work, like ODIS and HiSSD, aims to improve **multi-task** offline MARL generalization via skill discovery. For this reason, in the main experimental section, our primary comparisons are to skill-discovery methods (e.g., ODIS and HiSSD) plus the *standard* offline baseline BC used in these related works (e.g., ODIS and HiSSD). Since the Q-learning component of SD-CQL is based on vanilla CQL, we additionally include CQL as **a natural reference to isolate the benefit of our skill-discovery design**. The observation in Fig. 1a indicates that using CFCQL instead of CQL would not change the qualitative conclusion about generalization, and in our preliminary experiments, CFCQL-based variants exhibited stronger dependence on source tasks and worse generalization than CQL-based SD-CQL. Taken together, these considerations are why we did not include CFCQL as a main baseline in all experimental tables.
> > >
> > > ### On Fig. 1a, Fig. 1b and the description of ODIS
> > >
> > > As the caption (Line 61-66) indicates, Fig. 1 is intended to convey two complementary messages: skill discovery enhances cross-task generalization for offline MARL, while existing methods still have limitations. Lines 71–88 are organized accordingly:
> > >
> > > -  **Paragraph 1 (Line 71-80)**: Fig. 1a targets the *benefit of skill discovery*, and thus compares a representative task-specific method (CFCQL) with skill-discovery methods (ODIS and SD-CQL).
> > > - **Paragraph 2 (Line 81-88)**: Fig. 1b targets the *limitations of current skill-discovery methods under limited source tasks*, and therefore compares three skill-discovery approaches: ODIS (2023), HiSSD (2025), and SD-CQL.
> > >
> > > ODIS is introduced in Line 81 as a representative skill-discovery baseline, and thus appears in both panels. To make the connection between the text and Fig. 1 clearer, in the revision we have added an explicit reference to Fig. 1a when ODIS is first mentioned around Lines 81–82, before discussing Fig. 1b.

---

> > > > ### Author Response · Authors · 2025-12-02
> > > > **Response to Official Comment by Reviewer kwKd (Part 2/3)**
> > > >
> > > > > Regarding W2/Q4: The Remark in Appendix D.4 does not adequately address my query. The empirical rule provided by the authors suggests using a large η for lower-quality datasets and a small η for higher-quality datasets. However, the hyperparameter settings presented in Table 11 appear to contradict this rule, as the two groups of configurations exhibit inconsistent patterns:
> > > > Marine-Easy: Expert (higher-quality) η=0.9 > Medium (lower-quality) η=0.2 (Contradicts the rule)
> > > > Marine-Hard: Expert (higher-quality) η=0.5 < Medium (lower-quality) η=0.8 (Consistent with the rule)
> > > > Moreover, the ablation study in Table 17 indicates that a smaller η consistently yields better results in the two tested environments. Why, then, was η=0.9 selected for other environments (e.g., Marine-Easy-Expert)? Were ablation studies conducted for these specific environments as well? If so, I request that these additional experimental results be included.
> > > >
> > > > Thank you for your careful reading and the detailed question. We would like to clarify that:
> > > >
> > > > ### Remark D.4 and the apparent inconsistency in Table 11.
> > > >
> > > > First, we would like to clarify that the remark in Appendix D.4 is intended as an **empirical guideline** for tuning $\eta$, rather than as a *strict rule* that must monotonically hold across all environments and data mixtures. In SD-CQL, the LVC term shapes the Q-function using the **multi-task** behavior distribution, in order to counteract Q-value misestimation when data are limited and the number of agents is large. This makes the interaction between the LVC penalty and the TD term **more complex than the usual BC-style regularization used on a single-task dataset**.
> > > >
> > > > Moreover, following prior work such as ODIS and HiSSD, the “data quality” labels (e.g., medium, expert) are defined **per source task** at data collection time. When these task-specific datasets are mixed and trained jointly, the effective quality and coverage of the combined dataset are **no longer a simple monotone function of these labels**. For example, within Marine-Easy, the medium dataset of the 12m task may fail to consistently defeat all 12 enemies, but it could still provide trajectories that are already quite strong relative to the easier 3m task. In contrast, an expert dataset may be **narrower** in support (high win-rate but concentrated on a small set of behaviors). As a result, the heuristic "use a larger $\eta$ for lower-quality data and a smaller $\eta$ for higher-quality data" works well in many settings (e.g., considering medium, medium-replay, and medium-expert), but should not be interpreted as a universal, monotone rule across all multi-task mixtures.
> > > >
> > > > ### smaller $\eta$ in Table 17, and larger $\eta$ in marine-easy
> > > > Table 17 studies the one-to-multi setting, where SD-CQL **only has data from a single source task** and must generalize to multiple unseen tasks. This setting puts a much higher burden on task efficiency and cross-task generalization. In addition, expert data in this case typically have a relatively narrow state–action support. A large $\eta$ then over-constrains the learned policy to stay very close to this single, narrow dataset, which substantially harms performance on unseen tasks.
> > > >
> > > > This effect becomes particularly evident in the *Marine-Single* environments. For instance, in marine-single, the source task is 3m, whose expert data are much more concentrated than those from marine-single-inv (10m). With $\eta$=0.7, the LVC term dominates, and the policy collapses on unseen tasks, while smaller $\eta$ values achieve better generalization. This pattern is consistent with our observation in Fig. 1b that BC-based or behavior-heavy regularization can significantly hurt cross-task performance when only a limited set of source tasks is available.
> > > >
> > > > In contrast, *Marine-Easy-Expert* involves **expert** data from **three** source tasks, so both the data quality and the coverage are relatively good, and the tasks themselves are **simpler** (symmetric and isomorphic). Therefore, a larger $\eta$ is actually more appropriate. Our experimental results also show that **BC achieves the best performance** in this setting, which further corroborates this point.
> > > >
> > > > ### Regarding η = 0.9 in Marine-Easy-Expert
> > > > For Marine-Easy-Expert, $\eta$ = 0.9 was chosen via a small grid search as the best-performing value. However, this environment is relatively easy: the BC baseline already solves it quite well, and SD-CQL with $\eta$ = 0.9 is only about 1.4\% worse than the best baseline. In other words, Marine-Easy-Expert is relatively saturated, providing a limited signal for differentiating methods. We believe that Marine-Easy-Expert is a benign special case that does not affect the overall contributions of our work. **We can also provide the searching result on $\eta=0.5,0.7,0.9$ in Marine-Easy-Expert:**
> > > > |$\eta$|0.5|0.7|0.9
> > > > |-|-|-|-
> > > > |Performance|96.69 $\pm$ 4.75|97.94 $\pm$ 1.06|98.06 $\pm$ 1.00

---

> > > > > ### Author Response · Authors · 2025-12-02
> > > > > **Response to Official Comment by Reviewer kwKd (Part 3/3)**
> > > > >
> > > > > > Regarding Q2: I apologize if my previous phrasing was unclear. My specific question is: Why does the decoder not utilize the recurrent latent h? The paper does not appear to provide a justification for this design choice. While Appendix F.1.2 demonstrates the benefits of using h in the encoder, there is no corresponding ablation study analyzing the exclusion of h from the decoder.
> > > > >
> > > > > Thank you for the suggestion. We would like to clarify that, as stated on Lines 247–248, we do not feed the recurrent latent $h$ into the decoder in order to **“enable the skill to capture temporal information.”** The design intuition is to avoid a situation where the decoder can recover most of the temporal structure directly from its recurrent state $h$, instead of relying on the skill latent $z$. By excluding $h$ from the decoder, we explicitly encourage $z$ to encode the relevant temporal information.
> > > > >
> > > > > At the same time, we would like to clarify that we **do not** position “not using $h$ in the decoder” as a core novelty or as a key point that differentiates SD-CQL from prior methods; rather, it is **a reasonable architectural choice** under which SD-CQL **already achieves strong overall performance**. Therefore, this detail is not central to our main contribution.
> > > > >
> > > > > Nevertheless, we agree that it is useful to empirically validate this design. Across the four task sets below, excluding $h$ from the decoder **tends to be more beneficial on the more complex tasks**, while having only a minor effect on simpler ones. In the revised version, we added these ablation results in Appendix F.1.2.
> > > > >
> > > > > |Task Set|Marine-Single|Marine-Single-Inv|Marine-Hard-Medium|Stalker-Zealot-Medium
> > > > > |-|-|-|-|-
> > > > > SDCQL|62.00 $\pm$ 13.80|**87.50** $\pm$ 6.54|**48.12** $\pm$ 5.66|**40.43** $\pm$ 7.06
> > > > > decoder with $h$|**71.38** $\pm$ 10.52|84.25 $\pm$ 8.68|46.2 $\pm$ 4.56|38.03 $\pm$ 10.12

---

### Official Review · Reviewer_kjLX · 2025-10-30

**Soundness:** 2
**Presentation:** 3
**Contribution:** 2
**Rating:** 2
**Confidence:** 2

**Summary:**

The paper proposes SD-CQL, a value-based multi-task offline MARL framework that integrates skill discovery via observation reconstruction with conservative Q-learning. Each agent independently learns skills from local observation transitions without global state or reward supervision. Experiments on SMAC show improve in-domain zero-shot adaptation performance.

**Strengths:**

The paper is well-written and easy to follow, presenting a clear framework that integrates skill discover with conservative value learning for multi-agent offline MARL. It demonstrate consistent performance improvements on SMAC benchmarks and provides quantitative analysis of the learned skill representations.

**Weaknesses:**

1. The skill discovery mechanism relies solely on local observation, without access to global state or task-specific rewards to learn latent skills. In this way, the learned skills mainly capture individual behavioral modes rather than genuinely cooperative or team-level strategies. This local-only formulation fundamentally limits the framework's capacity to represent coordinated multi-agent behaviors.

2. The method essentially performs multiple instances of single-agent skill discovery rather than discovering cooperative skills, making the conceptual contribution to the MARL community relatively limited.

3. In the proposed framework, the shared semantic skills across agents implicitly assumes that agents possess homogeneous embodiments or action structures. How would the skill discovery mechanism generalize when agents differ in embodiment or role (e.g., marine & medic)?

4. The semantic expressivity of the proposed skill discovery is fundamentally limited by the SMAC environment's discrete action space, which consists mainly of move and attack primitives. As shown in Section 4.3, the discovered skills are restricted to two primary modes (retreat and attack). To demonstrate the emergence of diverse or compositional skills, the method should be further validated in continuous-control or more complex environments.

5. The evaluation is limited to in-domain variations, where only unit scales differ, providing weak support for true cross-domain generalization beyond the training distribution.

**Questions:**

1. How does the proposed skill discovery differ conceptually from existing self-supervised skill learning methods for single-agent learning?

2. Since the learned skills are derived from local observation transitions, how does the framework ensure that these skills contribute to coordinated multi-agent behavior?

3. How would the proposed method generalize to continuous-control environments?

---

> ### Author Response · Authors · 2025-11-20
> **Response to Reviewer kjLX (Part 1/4)**
>
> Thank you for your time and effort in reviewing our paper! We are grateful for your constructive suggestions, which have significantly guided our improvements. Please find our responses to your comments below.
>
> ### Weaknesses
> >**Weakness1** The skill discovery mechanism relies solely on local observation, without access to global state or task-specific rewards to learn latent skills. In this way, the learned skills mainly capture individual behavioral modes rather than genuinely cooperative or team-level strategies. This local-only formulation fundamentally limits the framework's capacity to represent coordinated multi-agent behaviors.
>
> Thank you for your comment! We respectfully emphasize that our skill discovery is a deliberate inductive bias: the skill module targets task- and scale-agnostic per-agent abstractions that transfer across tasks. Compared with single-agent methods, **our design tackles two key challenges in MARL:**
> - Variable state dimensionality across tasks
>
> - Scalability and generalization across varying task scales
>
> **Meanwhile, "local" does not mean "non-cooperative":** each agent's observation $o_t$ includes the teammates and opponents in its field of view. Our transformer encodes local relations among the agent, its teammates and its opponents, so **the latent $z$ still captures short-horizon cooperative cues**. We add a remark to explain this in Section 3.1.
>
> **Global coordination and credit assignment are learned through CTDE:** a centralized value function (QMIX-style $Q_{\text{tot}}$ over the global state and joint actions, about QMIX please refer to [1]) trains the skill-conditioned decentralized policies. Crucially, SD-CQL **selects actions by maximizing Q-values conditioned on $(o_t, z)$** rather than decoding actions directly from $z$, which is used by existing works (ODIS[2] and HiSSD[3]).
>
> **This design is one of our core contribution, and cleanly separates reusable individual capabilities (in $z$) from task- and team-specific signals (in the critic)**, avoids duplicating team information across modules, and yields **an effective trade-off between generalization/scalability and coordinated performance.**
>
>
> #### **References**
> ```
> [1] Rashid, T. et al. Monotonic Value Function Factorisation for Deep Multi-Agent Reinforcement Learning. JMLR, 2020.
> [2] Zhang, F. et al., Discovering Generalizable Multi-agent Coordination Skills from Multi-task Offline Data. ICLR 2023.
> [3] Liu, S. et al., Learning Generalizable Skills from Offline Multi-Task Data for Multi-Agent Cooperation. ICLR 2025.
> ```
>
> >**Weakness2** The method essentially performs multiple instances of single-agent skill discovery rather than discovering cooperative skills, making the conceptual contribution to the MARL community relatively limited.
>
> We appreciate the concern and respectfully disagree that our conceptual contribution to the MARL community relatively limited. Actually, our contribution to MARL is both **conceptual** and **methodological**.
>
> 1. **Conceptual Insight (skill–action mismatch in multi-task offline MARL):**
>     - Existing multi-task offline MARL methods typically **decode actions directly from the skill latent via Behavior Cloning (BC)**, which overlooks a **skill–action optimality mismatch**, i.e., the optimal skill not necessary decode the most suitable action. As shown in Figure 1, such mismatch could limit their generalizability.
>     - In contrast, our experiments demonstrate that **our value-based method is not only viable but outperforms** existing BC-style methods (ODIS[1] and HiSSD[2]) in multi-task offline MARL, offering a **new perspective and toolkit for the community**.
>
> 2. **Methodological Innovation (value-based skill conditioning):**
>     - We introduce a novel method, **SD-CQL**, which first **discovers transferable skill vector $z$** and then **selects actions via Q-values conditioned on $(o_t, z)$** instead of decoding actions from $z$.
>     - As mentioned in our response to weakness 1, this design **decouples representation from coordination**: skill discovery focuses on **transferable per-agent abstractions with local cooperative cues**, while CTDE provides **team-level coordination via $Q_{\text{tot}}$**, yielding **an effective trade-off between generalization/scalability and coordinated performance.**
>
> In additon, the visualizations in Appendix E.2, show that our method **outperforms** existing skill-discovery approaches and yields a **more expressive skill latent**. Moreover, the ablation studies in Appendix F.1 indicate that the proposed skill discovery is **a major contributor to SDCQL's strong performance for multi-task offline MARL**.
>
>
> #### **References**
> ```
> [1] Zhang, F. et al., Discovering Generalizable Multi-agent Coordination Skills from Multi-task Offline Data. ICLR 2023.
> [2] Liu, S. et al., Learning Generalizable Skills from Offline Multi-Task Data for Multi-Agent Cooperation. ICLR 2025.
> ```

---

> > ### Author Response · Authors · 2025-11-20
> > **Response to Reviewer kjLX (Part 2/4)**
> >
> > >**Weakness3** In the proposed framework, the shared semantic skills across agents implicitly assumes that agents possess homogeneous embodiments or action structures. How would the skill discovery mechanism generalize when agents differ in embodiment or role (e.g., marine & medic)?
> >
> > Thank you for the comment! We would like to clarify that:
> > - Our skills are discovered from each agent’s own observation, which explicitly **includes the agent’s type information**.
> > - In addition, following prior work (UpDeT[1], ODIS[2], and HiSSD[3]), we employ **a rule-based decomposer** that partitions, for each agent, the entity-level information in its observation, including interactable targets. We also use **separate Q-network** designs to handle fixed-length and variable-length action spaces, respectively.
> > - Moreover, **on a heterogeneous task (i.e., stalker–zealot), our method achieves better performance and generalization than the baselines**, as summarized in the table below.
> >
> >
> > |Data Quality |BC-best|CQL|ODIS|HiSSD|SD-CQL (Ours)|
> > |-|-|-|-|-|-|
> > |Expert| 68.61 $\pm$ 3.93 |10.62 $\pm$ 5.88 | 55.05 $\pm$ 9.11 | 66.59 $\pm$ 7.77 | **75.72** $\pm$ 3.73
> > |Medium| 23.94 $\pm$ 2.72 |15.29 $\pm$ 4.89 |11.97 $\pm$ 7.53 | 23.51 $\pm$ 3.76 | **40.43** $\pm$ 7.06
> > |Medium-Replay| 17.64 $\pm$ 4.40  |14.57 $\pm$ 2.02| 8.17 $\pm$ 6.08 | 15.00 $\pm$ 4.02 |**23.08** $\pm$ 3.38
> > |Medium-Expert| 39.62 $\pm$ 2.98 |13.41 $\pm$ 2.53| 30.72 $\pm$ 14.65| 40.29 $\pm$ 4.64 |**66.97** $\pm$ 7.01
> > #### **References**
> > ```
> > [1] Hu, S. et al., UPDeT: Universal Multi-agent RL via Policy Decoupling with Transformers. ICLR 2021.
> > [2] Zhang, F. et al., Discovering Generalizable Multi-agent Coordination Skills from Multi-task Offline Data. ICLR 2023.
> > [3] Liu, S. et al., Learning Generalizable Skills from Offline Multi-Task Data for Multi-Agent Cooperation. ICLR 2025.
> > ```

---

> > > ### Author Response · Authors · 2025-11-20
> > > **Response to Reviewer kjLX (Part 3/4)**
> > >
> > > >**Weakness4** The semantic expressivity of the proposed skill discovery is fundamentally limited by the SMAC environment's discrete action space, which consists mainly of move and attack primitives. As shown in Section 4.3, the discovered skills are restricted to two primary modes (retreat and attack). To demonstrate the emergence of diverse or compositional skills, the method should be further validated in continuous-control or more complex environments.
> > >
> > > Thank you for your suggestion! However, We would like to clarify that **the discrete action space of SMAC does not fundamentally limit the expressivity of our skill discovery module**. Although the instantaneous actions consist mainly of move and attack primitives, SMAC still exhibits **rich temporal behavioral patterns**, which can be **effectively captured by SDCQL’s next–local-observation reconstruction mechanism**.
> > >
> > > We would also like to emphasize that the visualization in Section 4.3 is intended **only to illustrate** that SDCQL is able to (i) discover distinct skills and (ii) choose the optimal action within each skill, thereby **overcoming the representation bottlenecks and skill–action mismatch** we observed in prior work such as ODIS. As shown in Appendix E.2, SDCQL successfully distinguishes **attack**, **retreat**, and **death** behaviors, whereas ODIS struggles to do so.
> > >
> > > Importantly, the fact that the visualization shows only two dominant modes does **NOT** imply that “skills are restricted to two primary modes.” Rather, the *Marine-Single* dataset and the 12m task **inherently contain and require only these two behavioral modes to solve the problem**. In more complex settings such as the heterogeneous *stalker–zealot–expert* task sets, SDCQL discovers **far more diverse patterns** from the source-task datasets and successfully transfers these skills to unseen tasks such as *4s3z*, including **ranged attack**, **melee attack**, **focus fire**, **flanking**, and **encirclement**, demonstrating that its skill discovery is **highly expressive and capable of capturing compositional strategies**. We add the illustrations of these skills in Appendix E.3.
> > >
> > > Furthermore, we additionally include a continuous-control evaluation based on the MPE Spread environment. We treat SS-2 as the source task and collect 2000 medium-quality trajectories for offline training, while SS-3 to SS-5 serve as unseen tasks. ("SS-N" means there are N agents and N landmarks in the environment.) All baselines are reimplemented in continuous-action versions. Especially, for SDCQL, we adopt an actor–critic architecture and feed the latent skill vector $z$ into both networks. Because the LVC mechanism is initially designed for discrete actions, we approximate it with a behavior-regularization term on the actor.
> > >
> > > Here are the results:
> > >
> > > | Method / Task | BC-t | BC-r | CQL | ODIS | HiSSD | **SDCQL (Ours)**|
> > > |-|-|-|-|-|-|-|
> > > | SS-2 (Source) | **-114.89** ± 2.25  | -168.94 ± 20.75 |-322.61 ± 36.93| -122.19 ± 5.2   | -126.29 ± 5.9   | -121.07 ± 2.72  |
> > > | SS-3 | -232.62 ± 5.72  | -356.87 ± 88.11  |-641.45 ± 89.99|-224.62 ± 7.64  | -247.59 ± 16.98 | **-224.45** ± 12.03 |
> > > | SS-4 | -378.52 ± 38.54 | -569.22 ± 199.68 |-1063.86 ± 157.31|**-354.97** ± 16.05 | -408.62 ± 60.12 | -357.15 ± 24.98 |
> > > | SS-5 | -581.95 ± 81.39 | -838.31 ± 283.33 |-1552.23 ± 248.58|-535.84 ± 34.75 | -522.73 ± 36.07 | **-494.2** ± 27.62  |
> > > | Average | -327.0 ± 25.77  | -483.34 ± 131.68 |-895.04 ± 117.72|-309.41 ± 9.99  | -326.31 ± 16.15 | **-299.22** ± 14.37
> > >
> > > As shown in the table above, **SDCQL achieves the best average performance**. Although BC performs well on the source task, its generalization degrades as the number of agents increases, which matches our observations in discrete-action settings. A detailed discussion of this continuous-control experiment has been added to the Appendix G.

---

> > > > ### Author Response · Authors · 2025-11-20
> > > > **Response to Reviewer kjLX (Part 4/4)**
> > > >
> > > > >**Weakness5** The evaluation is limited to in-domain variations, where only unit scales differ, providing weak support for true cross-domain generalization beyond the training distribution.
> > > >
> > > > Thank you for the feedback! We respectfully clarify that:
> > > > - Our study targets generalization across tasks with similar structure. All existing works use this evaluation setting (e.g., ODIS [1] and HiSSD [2]), while cross-domain scenarios, especially in the MARL setting, remain another new open problem. **Accordingly, “cross-domain” generalization is outside our current scope.**
> > > >
> > > > - Our evaluation spans 12 multi-to-multi task sets, and additionally includes two single-to-multi splits to assess task-efficiency. And these tasks vary by far more than unit scale: **the numbers and balance of teammates/opponents and the team-role composition (in stalker-zealot) all change, materially altering coordination difficulty and the induced winning strategies.**
> > > >
> > > > Our experiments show that these shifts **already challenge existing baselines, whereas SD-CQL improves, offering a novel and better solution.** These indicate that this setting is **a meaningful test of generalization**. That said, extending SD-CQL to "cross-domain" shifts is an interesting and more challenging direction, which we plan to explore in future work.
> > > >
> > > > #### **References**
> > > > ```
> > > > [1] Zhang, F. et al., Discovering Generalizable Multi-agent Coordination Skills from Multi-task Offline Data. ICLR 2023.
> > > > [2] Liu, S. et al., Learning Generalizable Skills from Offline Multi-Task Data for Multi-Agent Cooperation. ICLR 2025.
> > > > ```
> > > >
> > > > ### Questions
> > > > >**Question1** How does the proposed skill discovery differ conceptually from existing self-supervised skill learning methods for single-agent learning?
> > > >
> > > > Thank you for your question! We respectfully emphasize that, as highlighted in our responses to the weaknesses, our skill-discovery mechanism is distinct from single-agent methods: it **tackles two key challenges in MARL**:
> > > > - Variable state dimensionality across tasks
> > > > - Scalability and generalization across varying task scales
> > > >
> > > > Moreover, our contribution goes beyond skill discovery: **we introduce a new and more effective value-based approach for multi-task offline MARL, providing fresh conceptual and methodological insights to the community.**
> > > >
> > > > >**Question2** Since the learned skills are derived from local observation transitions, how does the framework ensure that these skills contribute to coordinated multi-agent behavior?
> > > >
> > > > Thank you for your question! As highlighted in our responses to Weaknesses 1 and 2, an agent’s observation $o_t$ includes information about nearby entities in its field of view, enabling the skill latent $z$ to encode **local cooperation cues**. Simultaneously, within the CTDE paradigm we **exploit global information to centrally train skill-conditioned Q-values**, which induces **global coordination**. This yields a principled balance between **scalability/generalization across tasks** and **team-level cooperative competence**.
> > > >
> > > > >**Question3** How would the proposed method generalize to continuous-control environments?
> > > >
> > > > Thank you for your question! Although SDCQL is primarily designed for discrete-action settings, it can be extended to continuous-control environments in a natural way. One intuitive approach is to **retain the skill discovery module and replace the QMIX-based policy optimization with an actor–critic architecture similar to FACMAC[1]**.
> > > >
> > > > Specifically, both the actors and the local critics additionally take the latent skill vector as input, while the global critic is computed by passing all local critic outputs through a QMIX-style mixing network. The LVC regularization term can be adapted by adding an MSE behavior regularizer between the actor’s output and the actions in the offline dataset.
> > > >
> > > > We implement this continuous-control extension and conduct additional evaluation experiments. As shown in our response to Weakness 4, **SDCQL still achieves the best average performance across tasks.** We would also like to emphasize that **achieving effective multi-task offline MARL in continuous-control environments remains an open and challenging future direction.**
> > > >
> > > > We have added these details in Appendix G in the revision.
> > > >
> > > > #### **References**
> > > > ```
> > > > [1] Peng, B. et al., FACMAC: Factored Multi-Agent Centralised Policy Gradients. NeurIPS 2021.
> > > > ```
> > > >
> > > > ---
> > > > We hope our response addresses your concerns. If so, we wonder if you could kindly consider raising your score? We will also be happy to answer any further questions you may have. Thank you very much!

---

> ### Author Response · Authors · 2025-11-25
> **Looking Forward to Your Feedback**
>
> Dear Reviewer kjLX,
>
> Thank you for your time and feedback! We hope our response and the revision have fully addressed your concerns. If so, we would appreciate it if you could raise your score.
>
> If you have any remaining concerns, we are more than happy to discuss them further.
>
> Best regards,
>
> The Authors

---

### Author Response · Authors · 2025-12-02
**General Response**

We sincerely thank the PC, SAC, AC, and all reviewers for their time and thoughtful feedback! Their comments have helped us substantially improve the clarity and quality of the paper.

---

### 1. Summary of our contributions

For convenience, we briefly restate the main contributions of our work:

- **Conceptual**
  We study a novel and underexplored problem, multi-task offline MARL, and identify a key limitation of existing methods. We show that a value-based approach can effectively address this issue and offer a new perspective for future research.

- **Methodological**
  We propose SD-CQL, which learns expressive latent skills via next-observation reconstruction and optimizes actions through skill-conditioned CQL with local value calibration, leading to improved generalization and task efficiency.

### 2. Main changes in the revised manuscript

Following the reviewers’ suggestions, we have made the following key revisions:

- **Additional experiment results**
    - We implement continuous-action versions of SD-CQL and all baselines and add a continuous-action task in Appendix G (Pages 27-28). The results show that SD-CQL **still achieves better performance and generalization than the baselines**.
    - We add skill visualization on the more complex task in Appendix E.3 (pp. 24-25), which further demonstrates the skill transfer capability of SD-CQL.
    - We add ablations on two more design choices in Appendix F.1.2 (pp. 26-27), providing a more comprehensive empirical validation of the design details of SD-CQL.
- **Improved explanations and remarks**
    - A clearer description of the “skill–action mismatching” issue in existing methods (p. 2, Line 85).
    - Explicitly state in the main text what the CFCQL and ODIS in Figure 1a refer to (p. 2, Lines 71 and 82)
    - The use of the term “skill” (footnote on p. 2).
    - The meaning of the recurrent latent (p. 5, Lines 245–247).
    - The reason why local observations can still contain collaboration information (Remark on p. 5, Lines 267–269).
    - The heterogeneity of the *stalker–zealot* scenario (p. 7, Lines 357–358).
    - Practical guidance on choosing the LVC coefficient $\eta$ (Remark in Appendix F.2, pp. 20–21, Lines 1079–1115).

* **Clearer and more complete data presentation**
    - In Table 1 (p. 8) and Table 2 (p. 9), we use boldface exclusively to indicate the best-performing method.
    - In Appendix D.3.3, we add a side-by-side comparison between our reproduced results and those reported in the original papers (Tables 7–9, pp. 20–21).
    - In Appendix I.1.1, we additionally report the average performance of all algorithms in Table 1 on the source tasks and unseen tasks separately (Tables 20–21, p. 30).

* **Minor fixes**\
  We also correct minor typos, for example, fixing the use of $K_i$ and $K^{a}_{i}$ (p. 4, Line 214) and using more appropriate notation for the skill-conditioned Q-value in Eqs. (4)–(6) on Page 6.

We explicitly mark all substantial changes in blue in the revised version.

### 3. Responses to main concerns

Below, we summarize how we address the main concerns.

- **Conceptual contributions to MARL** (kjLX and ozZf)\
We reiterate our conceptual and methodological contributions mentioned at the end of the Introduction, and explicitly introduce the term *skill–action optimality mismatch* to make the conceptual contribution clearer.

- **Meaning and expressiveness of “skills” learned from local next-observation prediction** (kjLX and ozZf)\
We clarify that our use of “skill” follows prior work and denotes a broader behavioral characteristic.  In our setting, the next-observation reconstruction objective captures short-term behavioral tendencies and local coordination cues. We revise the paper accordingly and add visualizations of more complex skills.

- **Interaction between skill discovery and Q-learning in cross-task generalization** (kjLX and ozZf)\
We highlight this separation as a key innovation: task-agnostic skill discovery learns reusable local behavioral patterns, while task-specific Q-learning optimizes actions of the skill, improving cross-task generalization. Our ablations support this design.

- **Extensions to the broader field** (kjLX)\
We add a continuous-control experiment, where SD-CQL still performs and generalizes the best. We clarify that the *stalker–zealot* scenario is already heterogeneous, and SD-CQL clearly outperforms baselines. We also note that cross-domain generalization is beyond the current scope and is an important direction for future work.

- **Presentation of results and ablation studies** (kwKd)\
We provide the requested original paper results and separate averages, and add ablations of two more design choices. For the hyperparameter $\eta$, we explain our choices and offer practical tuning guidelines.
---
We thank the PC, SAC, and AC for considering our work. We hope our rebuttal addresses the main concerns raised in the reviews, and the paper can be recommended for acceptance.

---

### Meta-Review · Area_Chair_mwtG · 2025-12-18

**Summary:**

The reviewers raised concerns about presentation (ozZf, kwKd), the correctness of baselines (kwKd), the experimental claims about generalization (kwKd, ozZf), and whether the method is actually learning temporally-extended skills (kjLX, ozZf). Broadly, the concerns seemed to be about making sure that all claims were crisply stated and experimentally validated.

Overall, the recommendation to reject the paper is based primarily on uncertainty about the strength of the experimental results (e.g., are they actually better than baselines in a statistically significant way? do they demonstrate generalization?). A secondary concern is the clarity and positioning of the paper (e.g., is it really learning temporally-extended skills)? For the second concern in particular, I might recommend that the authors frame the paper a bit more around specific testable hypotheses (e.g., it's unclear how one would disprove the hypothesis in L090). Both of these concerns should be straightforward to address in a subsequent submission.

**Reviewer Concerns:**

(see below)

**Reviewer Scores:**

kjLX: 2 --> 4
* [+] Conceptual contribution: I agree with the authors that this isn't a concern.
* [+] Homogeneous embodiedments: addressed with new experiments.
* [+] Learning action representations isn't interesting if there are only 2 actions: Authors provided counterexample on stalker–zealot–expert task in Appendix E.3.
* [+] Only evaluate on generalizing to different numbers of units: Prior work does the same thing.

kwKd: 4 --> 4
* [/] Clarifying notation: authors made some revisions
* [-] hyperparameter sensitivity: authors acknowledged that this is a challenge
* [+] generalization: authors provided disaggregated performance metrics
* [-] baseline validation: The authors acknowledged that the numbers reported in prior work (ODIS, HiSSD) were sometimes different than their implementation (sometimes better, sometimes worse). Plugging in the numbers from the original papers into Table 1 and bolding *all* methods that are within a std dev of the best method, we see that that there are few very tasks where we can say with confidence that the proposed method is better than prior methods.

ozZf: 2 --> 4
* [-] Interpretation as skills: The authors argue that they are capturing long-horizon patterns because they use some sort of recurrent model. I think that demonstrating this empirically would have been more convincing.
* [+] Is the method task agnostic: authors somewhat dodged this question, but showed that the method does generalize to new tasks (I suspect it does inherit some task-dependence from the training tasks, but maybe that's OK).
* [/] Too much focus on implementation: authors have said they'll revise

---

### Decision · Program_Chairs · 2026-01-26

Reject